# 🧑‍🏫 Pedagogically-Inspired Data Synthesis for Language Model Knowledge Distillation

**Bowei He**[1,2], **Yankai Chen**[1,2,*], **Xiaokun Zhang**[3], **Linghe Kong**[4], **Philip S. Yu**[5],
**Xue Liu**[1,2], **Chen Ma**[3,*]
[1] MBZUAI, [2] McGill, [3] CityUHK, [4] SJTU, [5] UIC

## Abstract

Knowledge distillation from Large Language Models (LLMs) to smaller models has emerged as a critical technique for deploying efficient AI systems. However, current methods for distillation via synthetic data lack pedagogical awareness, treating knowledge transfer as a one-off data synthesis and training task rather than a systematic learning process. In this paper, we propose a novel pedagogically-inspired framework for LLM knowledge distillation that draws from fundamental educational principles. Our approach introduces a three-stage pipeline—**Knowledge Identifier**, **Organizer**, and **Adapter** (**IOA**)—that systematically identifies knowledge deficiencies in student models, organizes knowledge delivery through progressive curricula, and adapts representations to match the cognitive capacity of student models. We integrate Bloom's Mastery Learning Principles and Vygotsky's Zone of Proximal Development to create a dynamic distillation process where student models approach teacher model's performance on prerequisite knowledge before advancing, and new knowledge is introduced with controlled, gradual difficulty increments. Extensive experiments using LLaMA-3.1/3.2 and Qwen2.5 as student models demonstrate that IOA achieves significant improvements over baseline distillation methods, with student models retaining 94.7% of teacher performance on DollyEval while using less than 1/10th of the parameters. Our framework particularly excels in complex reasoning tasks, showing 19.2% improvement on MATH and 22.3% on HumanEval compared with state-of-the-art baselines.

## 1 Introduction

Knowledge distillation (KD) has been a widely adopted approach to compress a large teacher model into a smaller student model by transferring knowledge (Hinton et al., 2015). Early-stage language model distillation methods mostly rely on minimizing the discrepancy of logit (output distribution) (Gu et al., 2024; Zhang et al., 2023) or intermediate layer representations (Jiao et al., 2020; Sun et al., 2020) between the teacher model and student model. However, due to the tokenizer mismatching and access limitation of proprietary large language model (LLM) parameters like OpenAI o1 (Jaech et al., 2024) and GPT-4 (Achiam et al., 2023), many recent distillation methods instead train the smaller LLM (SLM) using synthetic data generated by the teacher model. In fact, there have been many successful applications of LLM distillation with synthetic data, like Stanford's Alpaca model (Taori et al., 2023), which fine-tuned the open-sourced LLaMA-7B model (Touvron et al., 2023) with synthetic data from OpenAI's text-davinci-003 (i.e. the original ChatGPT model), improving its instruction-following capability to a comparable level. For the most recent reasoning models, DeepSeek (Guo et al., 2025) distilled DeepSeek-R1's outputs with simple fine-tuning, enabling the efficient DeepSeek-R1-Distill-Qwen-7B trained from Qwen2.5-7B (Qwen, 2024) to outperform specially designed reasoning model QwQ-32B-Preview (Qwen, 2025). The distilled model DeepSeek-R1-Distill-Qwen-32B even achieved higher performance than OpenAI-o1-mini on complex reasoning tasks like math, code, and science question answering (QA).

Some works have proposed different data synthesis strategies to distill LLM's knowledge and reasoning abilities into SLMs. One line of work (Yu et al., 2024) focuses on bootstrapping seed

---

*Corresponding authors: `yankaichen@acm.org`, `chenma@cityu.edu.hk`

questions by rewriting them from multiple perspectives, thereby enhancing distillation generalizability. Another line (Hsieh et al., 2023; Chen et al., 2024b; Dai et al., 2024) emphasizes eliciting and distilling the teacher LLM's Chain-of-Thought (CoT) reasoning traces to improve the student's step-by-step reasoning ability. Adversarial distillation frameworks such as Lion (Jiang et al., 2023) leverage feedback by identifying hard instructions and generating new ones to iteratively strengthen the student model's reasoning ability. Beyond these, counterfactual data synthesis (Feng et al., 2024a) has been proposed to encourage more robust reasoning by exposing the student to alternative problem-solving perspectives. To further safeguard the quality of the synthesized data, heuristic filtering strategies and rejection sampling mechanisms (Zhao et al., 2025) have also been adopted.

Although these methods achieve acceptable performance, many of them primarily focus on improving the quality or diversity of synthetic data either in a single-shot or iterative manner. Considering the analogy of LLMs as teachers and SLMs as students (as shown in Figure 1), these works did not explicitly model KD as a systematic knowledge teaching process, in which teaching contents and strategies should be dynamically and selectively adapted according to the student model's prior knowledge and learning progress. In fact, there are several fundamental principles in pedagogy (Bloom, 1968; Vygotsky & Cole, 1978) which can inspire better distillation design: (1) **What to teach**: *identifying and targeting critical knowledge deficiencies*. In the absence of specific knowledge units, even strong reasoning abilities may not be sufficient to obtain correct answers. For instance, a student with strong logical reasoning skills but no prior exposure to logarithms would be unable

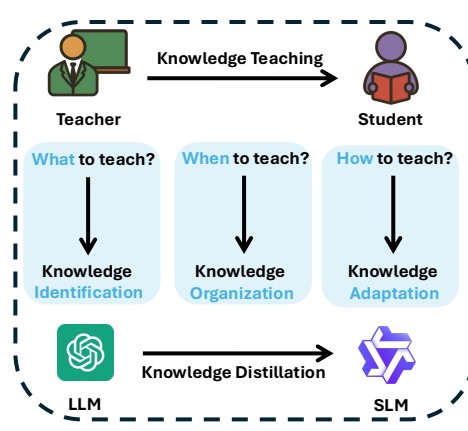

Figure 1: Analogy between real education and language model knowledge distillation.

to solve an equation such as $\log_2(x) = 3$. (2) **When to teach**: *organizing knowledge delivery through progressively complex curricula*. Designing a curriculum in which the knowledge components are sequentially connected and gradually increase in difficulty, while ensuring that students have a solid grasp of prior knowledge before progressing to the next stage-often leads to more effective learning outcomes. (3) **How to teach**: *adapting knowledge representation to the learner's cognitive level*. For example, in mathematics education, complex concepts like derivatives are often first introduced using intuitive, visual representations such as slope diagrams or motion-based analogies (e.g., the speed of a moving car), before transitioning to formal symbolic notation.

Based on such inspirations, we analyze the drawbacks of existing methods as follows: 1) **Knowledge Identification**: Current synthetic data generation lacks knowledge-aware targeting and fails to identify the specific knowledge that SLMs require to develop certain capabilities. The deficiency of such targeted knowledge remains prominent in SLMs. 2) **Knowledge Organization**: Existing synthetic data is generated without pedagogical organization, missing critical opportunities to optimize knowledge learning trajectories for SLMs. Ignoring the sequence of knowledge and failing to maintain an appropriate teaching pace make it difficult for SLMs to develop deep understanding and mastery of concepts. As a result, SLMs often rely on mechanical memorization, which undermines their ability to generalize in real-world scenarios. 3) **Knowledge Adaptation**: Most synthesis methods overlook SLM-adaptive knowledge representation, resulting in suboptimal knowledge absorption even with high-quality data. In fact, the cognitive level often cannot accommodate the knowledge involving abstract concepts or complex derivation process; such knowledge should be adapted using figurative metaphors and broken down into sub-steps to facilitate the knowledge distillation.

To effectively tackle these challenges, we propose a knowledge-aware three-stage data synthesis framework for distillation, **Identifier-Organizer-Adapter (IOA)**, that explicitly answers what to teach, when to teach, and how to teach through (i) deficiency diagnosis and dependency-aware targeting, (ii) curriculum sequencing with mastery gating, and (iii) cognitively aligned representation adaptation. We operationalize Bloom's Mastery Learning(Bloom, 1968) and Vygotsky's Zone of Proximal Development (ZPD) (Vygotsky & Cole, 1978) into concrete criteria: topological curricula, bounded difficulty increments, and stage-wise advancement rules, together with prompting templates for data synthesis. Empirically, across instruction-following and reasoning benchmarks, our IOA

framework achieves obvious gains over distillation baselines and consumes relately moderate time when taking LLaMA-3.1/3.2 and Qwen2.5 as student models. This underscores the effectiveness and efficiency of pedagogically inspired data synthesis for language model knowledge distillation.

## 2 RELATED WORKS

**Synthetic Data for LLMs** Synthetic data has been widely employed in the LLM scenario, because such LLM-generated text can significantly reduce the human annotation costs, overcome the data scarcity in specific low-resources scenarios (Nadas et al., 2025), and enhance the diversity of exiting web corpus (Chen et al., 2024a), thus further facilitating the extension of scaling laws. Previous works have explored how to augment the corpus formats and styles with synthesis data in pretraining phase (Hao et al., 2025). Besides, some researchers investigated the "model collapse" phenomenon when utilizing the synthetic data to recursively pretrain new language models (Feng et al., 2024b; Seddik et al., 2024; Zhu et al., 2024; Gerstgrasser et al., 2024). In the post-training phase, synthetic data has also been utilized to distill LLM knowledge and capabilities into smaller models (Taori et al., 2023; Guo et al., 2025). Recently, synthetic interaction trajectories have even been employed to strengthen LLM's agentic capabilities (Prabhakar et al., 2025). In addition to performance benefits, another advantage of synthetic data lies on protecting the privacy in case of the sensitive information leakage from the original data (Flemings & Annavaram, 2024). In this paper, our focused scenario is that SLMs have exhibited base knowledge obtained from pretraining, but needs effective distillation via synthetic data to gain more difficult knowledge and more complex task-completion capabilities.

**Language Model Distillation** Knowledge distillation has been successfully applied to language models to enrich model internal knowledge and enhance various capabilities like instruction following and reasoning. Previous representative distillation approaches include logit/output distribution distillation (Gu et al., 2024; Zhang et al., 2023; Ko et al., 2024; Wang et al., 2025b; Ko et al., 2025) and intermediate layer representation distillation (Jiao et al., 2020; Liang et al., 2023; Sy et al., 2024). Especially, on-policy distillation (Agarwal et al., 2024) is an effective approach to tackle the distribution mismatch issue between output sequences seen during training and those generated by the student during inference. Some previous works have also explored integrating curriculum learning into model distillation by ordering "easy-to-hard" training samples according to the teacher-student output distribution divergence (Zhu et al., 2021) or learning from successive intermediate checkpoints of the teacher progressively (Panigrahi et al., 2025; Gupta & Karmalkar, 2025). However, these types of white-box schemes are often unpractical when distilling knowledge from most advanced LLMs, because many of them are closed-sourced commercial LLMs, like OpenAI o1 (Jaech et al., 2024) and Gemini 2.5 (Comanici et al., 2025). In contrast, the black-box distillation with LLM-generated synthetic data has become a more common and realistic way. Some recent works (Zhou et al., 2024; Wang et al., 2025a) has noticed the significance of such data synthesis and proposed corresponding approaches, like augmenting the original seed mathematical questions and answers with LLMs (Yu et al., 2024), eliciting the teacher LLM's Chain-of-Thought reasoning process (Hsieh et al., 2023; Chen et al., 2024b; Dai et al., 2024; Yang et al., 2025), and introducing counterfactual data generation (Feng et al., 2024a). Notably, Ocra series works (Mukherjee et al., 2023; Mitra et al., 2023) highlight that adapting knowledge representations in synthetic data to the student's cognitive level can produce substantial gains. Different from previous works, our this work mainly explores how to teach a student model with a large black-box teacher via pedagogically guided data synthesis.

## 3 METHODOLOGY

### 3.1 PROBLEM FORMULATION

Given a teacher LLM $T$ with parameters $\theta_T$ and a student SLM $S$ with parameters $\theta_S$ (where $|\theta_S| \ll |\theta_T|$), the goal is to generate a synthetic dataset $\mathcal{D}_{\text{syn}}$ that maximizes the knowledge transfer from $T$ to $S$. By trained on distillation data $\mathcal{D}_{\text{syn}}$, the parameters of student model $S$ are updated to $\theta'_S$, whose performance is further evaluated on the specific targeted downstream tasks. Conventional approaches generate $\mathcal{D}_{\text{syn}}$ by prompting $T$ with seed data $\mathcal{D}_{\text{seed}} = \{d_1, d_2, ..., d_n\}$:

$$\mathcal{D}_{\text{syn}} = \bigcup_{i=1}^{n} \{\hat{d}_{i,j} : \hat{d}_{i,j} = T(d_i; \theta_T), j = 1, 2, \ldots, J_i\}, \tag{1}$$

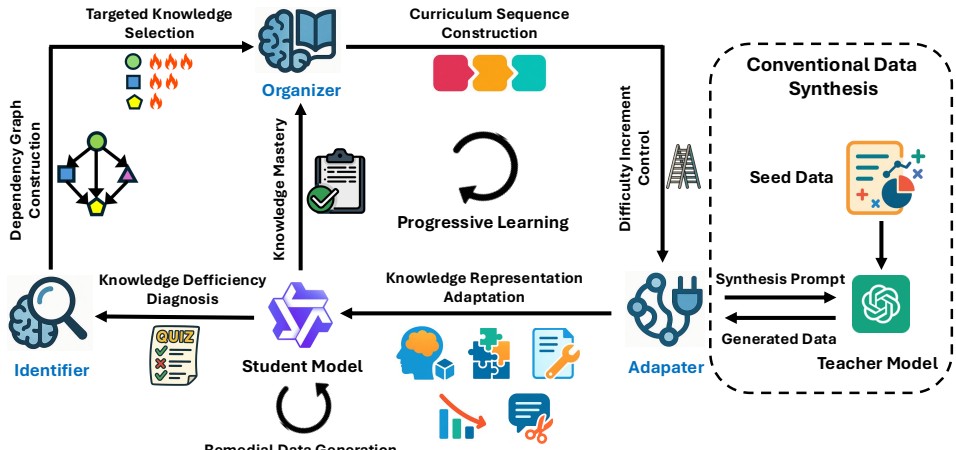

Figure 2: Pedagogically-inspired data synthesis framework for language model knowledge distillation.

where $J_i$ represents the number of synthetic samples generated from the $i$-th seed data $d_i$. Here, we assume access to a compact seed dataset that provides minimal yet sufficient coverage of the target domain, and is partitioned into training and validation splits. The training portion serves as the initial contexts for synthesis, while the validation split later enables diagnostic evaluation.

### 3.2 IDENTIFIER: KNOWLEDGE DEFICIENCY DIAGNOSIS AND TARGETING

The knowledge Identifier module first diagnoses knowledge deficiencies by systematically evaluating performance disparities between teacher and student models across fine-grained knowledge components, and then prioritizes most critical knowledge gaps by analyzing knowledge dependency.

**Knowledge Deficiency Diagnosis** Rather than treating capabilities as monolithic entities, our approach first decomposes complex domains (e.g., mathematical reasoning) into constituent knowledge modules, enabling identification of specific deficiencies that impede student model performance. For a given target capability domain $\mathcal{D}$ (e.g., mathematical problem-solving), we employ a hierarchical decomposition strategy that organizes knowledge across multiple granular levels. Taking mathematics as an example, we query the teacher LLM to structure knowledge units as: $\mathcal{D} = \{K_1, K_2, ..., K_m\}$ where $K_i = \{k_{i,1}, k_{i,2}, ..., k_{i,n_i}\}$. Here, $K_i$ represents major knowledge categories (e.g., algebra, geometry, calculus) and $k_{i,j}$ denotes specific knowledge modules within each category (e.g., linear equations, quadratic functions, trigonometric identities). This structure facilitates both comprehensive coverage and targeted intervention. Note this knowledge hierarchy keeps fixed once generated for each domain. Based on this, given a teacher model $T$ and a student model $S$, we evaluate their respective capabilities on carefully constructed probe tasks $\mathcal{P}_k$ for each knowledge module $k$. The data of $\mathcal{P}_k$ comes from the validation part of seed data $\mathcal{D}_{\text{seed}}$ (see Appendix B for details). The performance gap is quantified as:

$$\Delta(k) = \frac{P_T(k) - P_S(k)}{P_T(k)} = \frac{\sum_{p \in \mathcal{P}_k} \left[ \mathbb{I}[T(p) = y_p^*] - \mathbb{I}[S(p) = y_p^*] \right]}{\sum_{p \in \mathcal{P}_k} \mathbb{I}[T(p) = y_p^*]}, \tag{2}$$

where $P_T(k)$ and $P_S(k)$ represent the score of teacher and student models on knowledge module $k$ (accuracy or ROUGE-L depending on the probe task), respectively. Knowledge modules are classified as deficient when $\Delta(k) > \tau_{\text{gap}}$, typically set to 0.3 to ensure meaningful performance disparities warrant targeted intervention.

**Targeted Knowledge Selection** Understanding prerequisite relationships is crucial for establishing pedagogically sound learning trajectories. We construct a directed acyclic graph $G = (V, E)$ where vertices $V$ represent knowledge modules and edges $E$ encode prerequisite dependencies. The dependency strength between modules $k_i$ and $k_j$ is empirically determined through conditional performance analysis:

$$\text{Dependency}(k_i \to k_j) = \frac{P_S(k_j | \frac{P_S(k_i)}{P_T(k_i)} \geq \tau_{\text{high}}) - P_S(k_j | \frac{P_S(k_i)}{P_T(k_i)} < \tau_{\text{low}})}{P_S(k_j | \frac{P_S(k_i)}{P_T(k_i)} \geq \tau_{\text{high}}) + \epsilon}, \tag{3}$$

where $\tau_{\text{high}}$ and $\tau_{\text{low}}$ are two thresholds to indicate if student model has mastered the knowledge module $k_i$ enough during the learning process, which are set as 0.9 and 0.7, respectively. This metric captures how mastery of prerequisite knowledge $k_i$ influences performance on dependent knowledge $k_j$. Dependencies with strength above threshold $\tau_{\text{dep}} = 0.3$ are incorporated into the graph structure. Here, to compute $P_S(k_j)$ under different conditions in Eq. 3, we simply record the student's performance on $k_j$ at the moments when $\frac{P_S(k_i)}{P_T(k_i)}$ first exceeds $\tau_{\text{high}}$ or falls below $\tau_{\text{low}}$. Beside, to handle rare cyclic knowledge dependencies in this graph, we detect cycles and break them by removing the weakest edge (lowest dependency score), ensuring that the resulting structure remains a valid DAG for curriculum construction. To prioritize distillation efforts, we establish a severity ranking mechanism that considers both absolute performance gaps and relative importance within the knowledge hierarchy. The deficiency severity score is computed as:

$$\text{Severity}(k) = \alpha \cdot \Delta(k) + (1 - \alpha) \cdot \frac{1}{|\mathcal{N}(k)|} \sum_{k' \in \mathcal{N}(k)} \text{Dependency}(k \to k') \tag{4}$$

where the first term captures the absolute performance gap between teacher and student models. The second term measures the knowledge module's connectivity within the dependency graph $G$, where $\mathcal{N}(k)$ represents neighboring modules and $\text{Dependency}(k \to k')$ denotes dependence strength. The weighting parameters $\alpha$ is empirically set to 0.7, prioritizing performance gaps while accounting for structural importance. The identifier outputs a prioritized list of deficient knowledge modules $\mathcal{K}_{\text{target}} = \{k_1, k_2, ..., k_t\}$ ranked by severity scores, along with their associated dependency structures. This targeted selection ensures that subsequent synthetic data generation focuses on the most critical knowledge gaps, maximizing distillation efficiency. Specifically, modules are selected such that:

$$\mathcal{K}_{\text{target}} = \text{Top-}m(\mathcal{K}_{\text{deficient}}, \text{Severity}(\cdot)), \tag{5}$$
$$\text{where } \mathcal{K}_{\text{deficient}} = \{k : \Delta(k) > \tau_{\text{gap}}\}. \tag{6}$$

The threshold $m$ is dynamically adjusted based on available computational resources and desired distillation scope, typically set to cover approximately 20–30% of difficient modules $\mathcal{K}_{\text{deficient}}$ to balance comprehensiveness with tractability.

### 3.3 Organizer: Progressive Curriculum Design with Mastery Learning

The knowledge Organizer module transforms the deficient knowledge modules prioritized by the Identifier into a pedagogically structured learning sequence. Drawing from Bloom's Mastery Learning Principles (Bloom, 1968) and Vygotsky's Zone of Proximal Development (Vygotsky & Cole, 1978) (see Appendix A for details), this module ensures that knowledge acquisition follows optimal learning trajectories while maintaining an appropriate progression of difficulty.

**Curriculum Sequence Construction** Given the targeted knowledge modules $\mathcal{K}_{\text{target}}$ and their dependency graph $G$ from the Identifier, we construct a learning sequence $\mathcal{S} = \langle s_1, s_2, ..., s_n \rangle$ where each stage $s_i$ contains a subset of semantically similar knowledge modules to be learned simultaneously (e.g., linear equations and systems of linear equations). The sequence construction follows topological ordering of the dependency graph while respecting pedagogical constraints:

$$s_i = \{k \in \mathcal{K}_{\text{target}} : \forall k' \in \text{Prerequisites}(k), k' \in \bigcup_{j < i} s_j\}. \tag{7}$$

Here, the Prerequisites$(\cdot)$ function returns the set of prerequisite knowledge modules identified from the dependency graph in the Identifier stage, formally defined as Prerequisites$(k) = \{ k' \mid \text{Dependency}(k' \to k) > \tau_{\text{dep}} \}$. This ensures that all prerequisite knowledge is mastered before introducing dependent knowledge. Notably, to create gentle learning curves and reduce cognitive overload, we apply Vygotsky's ZPD principles by controlling difficulty increments between consecutive learning stages. The difficulty difference between stages is constrained as:

$$\frac{1}{|s_{i+1}|} \sum_{k \in s_{i+1}} P_S(k) - \frac{1}{|s_i|} \sum_{k \in s_i} P_S(k) \leq \tau_{\text{ZPD}} \cdot \frac{1}{|s_i|} \sum_{k \in s_i} P_S(k) \tag{8}$$

where $\frac{1}{|s_i|} \sum_{k \in s_i} P_S(k)$ represents the average difficulty level of knowledge modules in stage $s_i$, and $\tau_{\text{ZPD}} = 0.15$ ensures that difficulty increases remain within the student model's learning capacity.

---

**Algorithm 1** Pedagogically-Inspired Data Synthesis for Language Model Knowledge Distillation

---

**Require:** Teacher model $T$, Student model $S$, Target domain $\mathcal{D}$, Seed data $\mathcal{D}_{\text{seed}}$
**Ensure:** Fine-tuned student model $S'$
 1: **// Identifier: Deficiency Diagnosis and Targeting**
 2: $\mathcal{K}_{\text{modules}} \leftarrow \text{DecomposeKnowledge}(\mathcal{D})$; $\Delta \leftarrow \text{EvaluateGaps}(T, S, \mathcal{K}_{\text{modules}})$
 3: $G \leftarrow \text{ConstructDependencyGraph}(\mathcal{K}_{\text{target}})$; $\mathcal{K}_{\text{target}} \leftarrow \text{PrioritizeCriticalGaps}(\Delta, G)$
 4: **// Organizer: Curriculum Sequence Construction**
 5: $\mathcal{S} \leftarrow \text{OrganizeLearningSequence}(\mathcal{K}_{\text{target}}, G, S)$
 6: **// Organizer: Mastery-Based Progressive Learning**
 7: **for** $i = 1$ to $|\mathcal{S}|$ **do**
 8:    **// Adapter: Cognitive Level Alignment**
 9:    $\mathcal{D}_{\text{syn}} \leftarrow \text{AdaptKnowledgeRepresentation}(T, s_i, \mathcal{D}_{\text{seed}})$
10:    $S \leftarrow \text{FineTune}(S, \mathcal{D}_{\text{syn}})$; $P_S \leftarrow \text{EvaluateStage}(S, s_i)$; $P_T \leftarrow \text{EvaluateStage}(T, s_i)$
11:    **while** $\min_{k \in s_i} \frac{P_S(k)}{P_T(k)} < \tau_{\text{mastery}}$ **do**
12:       $S \leftarrow \text{FineTune}(S, \text{GenerateRemedialData}(T, s_i, \mathcal{D}_{\text{seed}}))$
13:    **end while**
14: **end for**
15: **return** $S'$

---

**Mastery-Based Progressive Learning** Following Bloom's criterion-referenced evaluation, we implement strict mastery requirements between learning stages. For each knowledge module $k$ in stage $s_i$, the SLM must achieve relative performance before progressing to learning next stage $s_{i+1}$:

$$\text{Progress}(s_i \rightarrow s_{i+1}) = \begin{cases} \text{True} & \text{if } \min_{k \in s_i} \frac{P_S(k)}{P_T(k)} \geq \tau_{\text{mastery}} \\ \text{False} & \text{otherwise.} \end{cases} \tag{9}$$

where $\tau_{\text{mastery}} = 0.9$ requires the student model to achieve 90% of the teacher model's performance before advancement. Otherwise, the remedial data will be synthesized to continue learning knowledge in this stage untill mastery. This prevents knowledge gaps from accumulating and ensures that the student model has sufficiently mastered each prerequisite concept relative to the teacher's capability.

### 3.4 Adapter: Knowledge Representation Adaptation for Cognitive Alignment

The knowledge Adapter module transforms the structured curriculum from the Organizer into cognitively appropriate representations that align with the student model's learning capacity. Instead of querying the teacher model $T$ with $\mathcal{D}_{\text{seed}}$ and then directly utilizing the obtained synthesized data $\mathcal{D}_{\text{syn}}$, the Adapter systematically modifies content delivery to match the cognitive constraints of student models, ensuring genuine understanding rather than mechanical memorization. The adaptation is performed on the subset of $\mathcal{D}_{\text{seed}}$ corresponding to the knowledge units scheduled for current stage $s_t$. In detail, the knowledge adaptation is conducted from following five perspectives:

**Abstract Concept Concretization** The Adapter transforms abstract mathematical concepts into concrete, intuitive representations using analogical reasoning. For instance, derivatives are initially introduced as "measuring how fast something changes, like a car's speedometer" before presenting the formal limit definition. Complex logarithmic relationships begin with concrete scenarios: "If bacteria double every hour, how many doublings reach 256 from 1?" This grounds abstract concepts in familiar experiences, thus facilitating the understanding and absorbing of knowledge in SLM .

**Complex Reasoning Decomposition** Multi-step reasoning processes are systematically disaggregated into atomic cognitive operations. For example, for math problems, the process is broken into sequential sub-procedures: information extraction $\Rightarrow$ relationship identification $\Rightarrow$ equation formation $\Rightarrow$ solution execution $\Rightarrow$ verification. Each component is mastered independently before integration, while building comprehensive reasoning capabilities. Note that if obtained results cannot pass the verification, the corresponding reasoning process will be filtered out.

**Cognitive Load Management** Information density and complexity are carefully regulated to match SLMs' cognitive constraints. Systems of equations begin with $2\times2$ integer coefficient cases before progressing to larger or fractional systems. Complex problems are divided into manageable segments with explicit intermediate verification, ensuring sustainable learning progression.

📝 **Representation Format Optimization** Knowledge encoding is restructured using templated frameworks and structured pedagogical formats. Standardized solution templates provide consistent scaffolding, for example: "Step 1: Identify coefficients. Step 2: Calculate discriminant. Step 3: Apply formula." Multiple worked examples following the same templates, while varying surface features, enable pattern extraction and facilitate robust knowledge generalization.

💬 **Linguistic Complexity Reduction** Content undergoes systematic simplification across vocabulary, syntax, and discourse structure. Mathematical terminology is replaced with simpler equivalents when possible ("reciprocal" for "multiplicative inverse"). Complex sentences with subordinate clauses are decomposed into direct statements. Besides, clear signaling words ("first," "therefore," "next") are integrated into texts to provide explicit logical structure guidance through reasoning processes.

### 3.5 Overall Knowledge Distillation Framework with Data Synthesis

Our pedagogically-inspired framework integrates the Identifier, Organizer, and Adapter modules into a coherent distillation pipeline that systematically transfers knowledge from teacher to student models through structured synthetic data generation. As shown in Algorithm 1, the framework operates in iterative cycles, continuously adapting the training data at each stage based on student model progress. The distillation process begins with a deficiency diagnosis and targeting phase where the Identifier evaluates performance gaps across knowledge modules and prioritizes the most critical gaps. The Organizer then constructs a progressive learning curriculum respecting prerequisite relationships, the zone of proximal development, and mastery requirements. In each curriculum stage, the Adapter generates cognitively appropriate data, which is used to train student models until achieving mastery.

## 4 Experiments

### 4.1 Experiment Setup

**Teacher and Student Models**: We utilize two most representative large reasoning LLMs as teacher models: closed-sourced OpenAI o1 (Jaech et al., 2024) and open-sourced DeepSeek-R1 (Guo et al., 2025). It is conservatively estimated that both models at least have over 100B parameters (Abacha et al., 2024). As for student models, we select recent small language models from Qwen family (Qwen2.5-3B, Qwen2.5-7B, Qwen2.5-14B (Qwen, 2024)) and LLaMA family (LLaMA3.1-8B (Meta AI, 2024a), LLaMA3.2-3B (Meta AI, 2024b)). Due to page limitation, we only provide results on two 3B models in the main text, leaving others in Appendix F.

**Evaluation Benchmarks** We employ two main types of benchmarks to comprehensively evaluate the effectiveness of different methods: *Instruction Following*: Dolly Evaluation (Conover et al., 2023) and Vicuna Evaluation (Chiang et al., 2023); *Reasoning*: math problem solving (GSM8K (Cobbe et al., 2021), MATH (Lewkowycz et al., 2022), AIME2024 (Mathematical Association of America, 2024)), code generation (HumanEval (Chen et al., 2021), MBPP (Austin et al., 2021), LiveCodeBench (Jain et al., 2025)), academic knowledge reasoning (GPQA-Diamond, abbreviated as GPQA-D (Rein et al., 2024)). As for the evaluation metrics, we adopt ROUGE-L and Pass@$k(k = 1)$ for instruction following and reasoning benchmarks respectively, referring previous literature (He et al., 2025; Guo et al., 2025).

**Baselines** We primarily compare our framework with following several types of data synthesis-based language model distillation approaches: *Vanilla Synthesis:* Self-Instruct (Wang et al., 2023), LaMini (Wu et al., 2024); *Adversarial Synthesis:* Lion (Jiang et al., 2023); *CoT Synthesis:* Distilling Step-by Step (DSS) (Hsieh et al., 2023), CasCoD (Dai et al., 2024); *Counterfactual Synthesis:* CounterDistill (Feng et al., 2024a); *Multi-Agent Synthesis:* Star-Agents (Zhou et al., 2024), MADA (Wang et al., 2025a).

Besides, we have provided the detailed seed data construction in Appendix B, detailed introductions to baselines and evaluation settings in Appendix C and D, implementation details and hyperparameter settings in Appendix E. Supplementary experiments on agentic tasks can been in Appendix G.

### 4.2 Main Results and Analysis

**Overall Performance Comparison** Tables 1 and 2 report results with OpenAI o1 and DeepSeek-R1 as teachers. IOA consistently achieves the best performance across both Qwen2.5-3B and LLaMA3.2-

Table 1: Comparison among different data synthesis-based distillation approaches on various evaluation benchmarks for LLaMA and Qwen models. OpenAI o1 is taken as the teacher LLM. **Bold** and underline indicate the best and second-best results, respectively. ∗ indicates statistically significant.

| Model | Method | DollyEval | VicunaEval | GSM8K | MATH | AIME2024 | HumanEval | MBPP | LiveCodeBench | GPQA-D |
|---|---|---|---|---|---|---|---|---|---|---|
| | Undistilled | 25.37 | 23.82 | 37.24 | 5.79 | 2.13 | 22.46 | 31.58 | 14.72 | 7.95 |
| | Self-Instruct | 32.18 | 29.36 | 43.69 | 7.12 | 2.97 | 25.63 | 36.27 | 16.84 | 9.28 |
| | LaMini | 33.94 | 31.15 | 45.28 | 7.96 | 3.05 | 26.83 | 37.49 | 17.42 | 9.65 |
| | Lion | 34.73 | 32.62 | 48.36 | 9.43 | 3.63 | 28.95 | 39.14 | 19.74 | 10.46 |
| Qwen2.5-3B | DSS | 35.28 | 33.45 | 49.57 | 10.64 | 4.08 | 30.57 | 40.03 | 20.85 | 10.94 |
| | CasCoD | 35.65 | 33.91 | 50.46 | 11.71 | 4.62 | 31.46 | 40.87 | 21.64 | 11.19 |
| | CounterDistill | 34.26 | 33.08 | 49.84 | 11.19 | 4.77 | 31.15 | 40.35 | 21.29 | 11.05 |
| | Star-Agents | 36.08 | 34.69 | 51.25 | 12.47 | 5.23 | 32.74 | 41.63 | 22.53 | 11.67 |
| | MADA | 36.42 | 35.09 | 52.04 | 13.15 | 5.54 | 33.39 | 42.18 | 23.06 | 11.93 |
| | IOA (Ours) | **38.16*** | **36.83*** | **55.79*** | **15.53*** | **6.29*** | **40.64*** | **47.86*** | **26.94*** | **13.74*** |
| | Undistilled | 23.68 | 21.97 | 34.13 | 5.12 | 1.81 | 20.35 | 29.63 | 13.24 | 7.14 |
| | Self-Instruct | 30.42 | 27.84 | 40.57 | 6.31 | 2.48 | 23.94 | 33.82 | 15.25 | 8.54 |
| | LaMini | 31.77 | 29.65 | 42.25 | 7.03 | 2.73 | 25.15 | 34.94 | 15.92 | 8.91 |
| | Lion | 32.91 | 30.86 | 45.63 | 8.59 | 3.26 | 27.14 | 36.53 | 18.16 | 9.72 |
| LLaMA3.2-3B | DSS | 33.63 | 31.57 | 46.75 | 9.68 | 3.74 | 28.52 | 37.35 | 19.26 | 10.08 |
| | CasCoD | 34.05 | 31.93 | 47.48 | 10.88 | 4.13 | 29.41 | 38.02 | 19.94 | 10.43 |
| | CounterDistill | 33.19 | 31.12 | 47.06 | 10.43 | 4.35 | 29.05 | 37.71 | 19.63 | 10.28 |
| | Star-Agents | 34.69 | 32.63 | 48.14 | 11.31 | 4.52 | 30.26 | 38.64 | 20.61 | 10.64 |
| | MADA | 35.32 | 33.21 | 48.82 | 11.76 | 4.58 | 31.18 | 39.13 | 21.02 | 10.87 |
| | IOA (Ours) | **36.88*** | **34.81*** | **52.07*** | **14.02*** | **5.47*** | **37.98*** | **44.25*** | **24.08*** | **12.15*** |

Table 2: Comparison among different data synthesis-based distillation approaches on various evaluation benchmarks for LLaMA and Qwen models. DeepSeek-R1 is taken as the teacher LLM. **Bold** and underline indicate the best and second-best results, respectively. ∗ indicates statistically significant.

| Model | Method | DollyEval | VicunaEval | GSM8K | MATH | AIME2024 | HumanEval | MBPP | LiveCodeBench | GPQA-D |
|---|---|---|---|---|---|---|---|---|---|---|
| | Undistilled | 25.39 | 23.84 | 37.23 | 5.82 | 2.12 | 22.45 | 31.59 | 14.73 | 7.94 |
| | Self-Instruct | 32.14 | 29.35 | 43.74 | 7.13 | 2.95 | 25.65 | 36.23 | 16.86 | 9.26 |
| | LaMini | 33.92 | 31.13 | 45.27 | 7.94 | 3.02 | 26.86 | 37.53 | 17.43 | 9.63 |
| | Lion | 35.01 | 32.93 | 49.12 | 9.79 | 3.82 | 29.73 | 39.63 | 20.23 | 10.62 |
| Qwen2.5-3B | DSS | 35.58 | 33.81 | 50.32 | 11.14 | 4.39 | 31.63 | 40.71 | 21.41 | 11.18 |
| | CasCoD | 36.13 | 34.25 | 51.18 | 12.29 | 4.91 | 32.63 | 41.52 | 22.25 | 11.54 |
| | CounterDistill | 34.72 | 33.37 | 50.74 | 11.82 | 5.08 | 32.17 | 41.03 | 21.82 | 11.29 |
| | Star-Agents | 36.62 | 35.07 | 52.15 | 13.21 | 5.58 | 33.69 | 42.37 | 23.14 | 11.92 |
| | MADA | 37.01 | 35.42 | 52.81 | 13.48 | 5.83 | 34.71 | 42.91 | 23.84 | 12.27 |
| | IOA (Ours) | **37.77*** | **36.29*** | **56.68*** | **16.02*** | **6.57*** | **42.08*** | **49.32*** | **27.73*** | **14.43*** |
| | Undistilled | 23.69 | 22.01 | 34.12 | 5.09 | 1.82 | 20.34 | 29.61 | 13.21 | 7.12 |
| | Self-Instruct | 30.41 | 27.83 | 40.53 | 6.29 | 2.53 | 23.92 | 33.81 | 15.24 | 8.52 |
| | LaMini | 31.82 | 29.58 | 42.24 | 7.02 | 2.71 | 25.13 | 34.92 | 15.91 | 8.94 |
| | Lion | 33.12 | 31.02 | 46.29 | 8.87 | 3.37 | 27.81 | 36.94 | 18.54 | 9.93 |
| LLaMA3.2-3B | DSS | 33.91 | 31.93 | 47.63 | 10.27 | 3.88 | 29.31 | 37.87 | 19.64 | 10.37 |
| | CasCoD | 34.46 | 32.28 | 48.47 | 11.46 | 4.32 | 30.28 | 38.72 | 20.27 | 10.72 |
| | CounterDistill | 33.63 | 31.47 | 48.02 | 11.03 | 4.48 | 29.94 | 38.25 | 20.03 | 10.51 |
| | Star-Agents | 34.97 | 33.01 | 49.09 | 11.91 | 4.74 | 31.03 | 39.19 | 20.92 | 10.94 |
| | MADA | 35.58 | 33.63 | 49.72 | 12.28 | 4.95 | 31.63 | 39.82 | 21.28 | 11.13 |
| | IOA (Ours) | **36.73*** | **34.72*** | **53.31*** | **14.81*** | **5.87*** | **39.37*** | **44.97*** | **24.88*** | **12.47*** |

3B, covering instruction-following, reasoning, and coding tasks. On instruction-following tasks (DollyEval, VicunaEval), IOA outperforms strong baselines like MADA by approximately 1.5–2.0 points. On mathematical reasoning, IOA reaches 15.53/14.02 (Qwen/LLaMA with o1), and further 16.02/14.81 with DeepSeek-R1, showing the added benefit of a reasoning-oriented teacher. For code benchmarks, the gains are most pronounced: HumanEval improves from 33–34 (MADA) to over 40 with IOA, a relative gain exceeding 20%, with MBPP and LiveCodeBench showing similar trends. On knowledge-intensive QA (GPQA-D), IOA also provides consistent 1–2 point improvements. In sum, IOA delivers robust gains across tasks and student SLMs, with especially strong advantages on math and coding when distilled from DeepSeek-R1. These results highlight the effectiveness of IOA's pedagogical design in transferring both instruction-following and reasoning abilities.

**Ablation Study** Table 3 presents the ablation results when removing the identifier, organizer, or adapter modules from IOA. Specifically, the "-Identifier" removes the deficiency-identification step, causing all knowledge modules to be treated uniformly and preventing the framework from focusing on the student's actual weaknesses; "- Organizer" disables the dependency-driven curriculum and presents identified deficient modules in a random order, removing prerequisite-aware sequencing; "- Adapter" eliminates the adaptive synthesis mechanisms in Section 3.4, prompting the teacher to generate data without difficulty or representation adjustments and thereby producing samples that

Table 3: Ablation study on various evaluation benchmarks for LLaMA and Qwen models. OpenAI o1 is taken as the teacher LLM. ∗ indicates statistically significant.

| Model | Method | DollyEval | VicunaEval | GSM8K | MATH | AIME2024 | HumanEval | MBPP | LiveCodeBench | GPQA-D |
|---|---|---|---|---|---|---|---|---|---|---|
| **Qwen2.5-3B** | Full IOA | 38.16* | 36.83* | 55.79* | 15.53* | 6.29* | 40.64* | 47.86* | 26.94* | 13.74* |
| | - Identifier | 36.72 | 34.91 | 54.23 | 14.92 | 6.01 | 39.87 | 46.95 | 26.13 | 13.21 |
| | - Organizer | 37.04 | 35.22 | 51.68 | 13.77 | 5.42 | 40.21 | 47.03 | 26.34 | 13.08 |
| | - Adapter | 37.61 | 36.05 | 54.72 | 15.01 | 6.12 | 36.48 | 42.67 | 23.18 | 13.25 |
| **LLaMA3.2-3B** | Full IOA | 36.88* | 34.81* | 52.07* | 14.02* | 5.47* | 37.98* | 44.25* | 24.08* | 12.15* |
| | - Identifier | 35.47 | 33.26 | 50.73 | 13.61 | 5.23 | 37.25 | 43.32 | 23.47 | 11.83 |
| | - Organizer | 35.69 | 33.74 | 48.92 | 12.48 | 4.81 | 37.51 | 43.61 | 23.64 | 11.72 |
| | - Adapter | 36.11 | 34.09 | 51.26 | 13.73 | 5.32 | 34.42 | 39.58 | 21.11 | 11.79 |

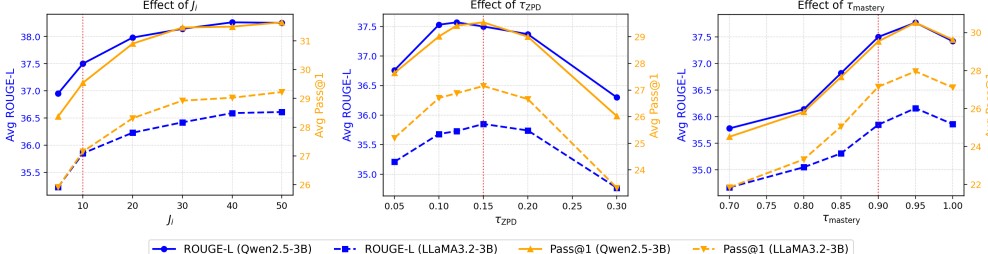

Figure 3: The hyperparameter robustness analysis for three critical hyperparameters $J_i$, $\tau_{\text{ZPD}}$, $\tau_{\text{mastery}}$.

may exceed the student's cognitive capacity. For Qwen2.5-3B and LLaMA3.2-3B, the complete IOA consistently achieves the highest scores, while excluding any component leads to noticeable degradation. Removing the identifier primarily hurts instruction-following benchmarks (DollyEval, VicunaEval), since errors can no longer be effectively localized. Removing the organizer impacts reasoning tasks such as GSM8K and MATH most severely, reflecting the importance of structured curriculum in learning problem-solving skills. Finally, removing the adapter causes the largest drops on coding tasks (HumanEval, MBPP, LiveCodeBench), highlighting the necessity of representation adaptation for generalizing in program synthesis. The consistent performance decline across all tasks when any component is removed demonstrates that each stage of IOA contributes uniquely and positively to the overall distillation performance.

**Hyperparameter Robustness** Figure 3 reports the effect of three hyperparameters: synthesis data amount $J_i$ for each seed data, proximal development zone threshold $\tau_{\text{ZPD}}$, and progressive learning mastery threshold $\tau_{\text{mastery}}$, on averaged ROUGE-L over instruction-tuning tasks (blue, left $y$-axis) and averaged Pass@1 over reasoning tasks (orange, right $y$-axis). The vertical red lines indicate our standard setting ($J_i$=10, $\tau_{\text{ZPD}}$=0.15, $\tau_{\text{mastery}}$=0.90). As $J_i$ increases from 5 to 50, both metrics improve and then plateau around $J_i \in [30, 40]$, showing diminishing returns; Qwen2.5-3B benefits slightly more than LLaMA3.2-3B. Varying $\tau_{\text{ZPD}}$ yields a clear non-monotonic trend with a peak near 0.15: too small under-challenges the learner, while larger thresholds overshoot the step size and hurt transfer. For $\tau_{\text{mastery}}$, performance rises as the requirement becomes stricter and peaks around 0.90–0.95, after which excessive strictness leads to a mild drop, likely due to overfitting and slower progression. Across all sweeps, Qwen2.5-3B maintains a consistent advantage yet follows the same trends as LLaMA3.2-3B, indicating that IOA is robust and easy to tune.

**Time-Consumption Analysis** To address concerns about end-to-end efficiency, Figure 4 compares the averaged wall-clock time of IOA with representative data-synthesis baselines under our standard setting. IOA (Ours) requires ∼11.0 h on Qwen2.5-3B and ∼12.2 h on LLaMA3.2-3B, outperforming heavier pipelines such as Lion (12.1/13.0 h; ≈9.1% / 6.2% faster) and CasCoD (11.6/12.7 h; ≈5.2% / 3.9% faster), and remaining competitive with MADA (11.8/12.6 h; ≈6.8% / 3.2% faster). While Self-Instruct is the fastest (9.8/10.7 h), it lags notably in quality, so its time is not directly comparable to IOA's stronger results (Tables 1–2). We attribute this efficiency to three key design choices in IOA. First, the Identifier narrows the synthesis scope by diagnosing knowledge gaps and selecting only

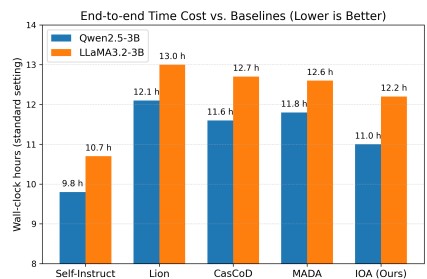

Figure 4: Time consumption comparison between our IOA and baselines.

Table 4: Comparison between IOA and other recent distillation methods. **Bold** and underline indicate the best and second-best results, respectively. ∗ indicates statistically significant.

| Model | Method | DollyEval | VicunaEval | GSM8K | MATH | AIME2024 | HumanEval | MBPP | LiveCodeBench | GPQA-D |
|-------|--------|-----------|------------|-------|------|----------|-----------|------|---------------|--------|
| | ABKD | 37.85 | 36.01 | 55.56 | 14.31 | 5.62 | 36.86 | 44.87 | 25.04 | 12.99 |
| | DistiLLM-2 | 38.28 | 36.80 | 56.35 | 15.07 | 6.23 | 38.14 | 45.63 | 28.93 | 13.47 |
| Qwen2.5-7B | GKD | 38.62 | 37.20 | 57.14 | 15.75 | 6.54 | 39.19 | 48.16 | 26.46 | 13.73 |
| | SuperCorrect | 38.46 | 36.98 | 56.24 | 16.51 | 7.97 | 35.95 | 45.05 | 24.99 | 15.05 |
| | POCL | 39.55 | 37.61 | 57.76 | 14.49 | 6.12 | 38.46 | 46.57 | 26.14 | 13.79 |
| | IOA (Ours) | **41.64*** | **39.97*** | **63.95*** | **19.64*** | **10.89*** | **47.67*** | **54.94*** | **32.20*** | **17.08*** |

Table 5: Averaged performance across 9 tasks under the moderately strong teacher model setting.

| Model | Self-Instruct | LaMini | Lion | DSS | CasCoD | CounterDistill | Star-Agents | MADA | IOA (Ours) |
|-------|---------------|--------|------|-----|--------|----------------|-------------|------|------------|
| **Qwen2.5-3B** | 20.25 | 22.18 | 25.02 | 25.45 | 27.11 | 26.07 | 27.65 | 28.32 | 31.19 |

the most essential missing components, thereby reducing teacher calls and training volume. Second, the Organizer enforces a dependency-aware curriculum with controlled difficulty progression, preventing wasted effort on overly challenging samples before prerequisites are mastered. Finally, the Adapter promotes structured and simplified representations that reduce unnecessary verbosity, resulting in more compact supervision and improved token efficiency during training. Together, these mechanisms enable IOA to achieve stronger performance while maintaining competitive or even reduced wall-clock time compared to more complex multi-round or adversarial pipelines.

### 4.3 SUPPLEMENTARY RESULTS AND ANALYSIS

**Comparison with Other Distillation Methods** To enhance comparison completeness, we also compare with several recent distillation methods beyond those in main experiments: *Vanilla White-box Distillation:* ABKD (Wang et al., 2025b) and DistiLLM-2 (Ko et al., 2025); *Only-Policy Logit Distillation:* GKD (Agarwal et al., 2024); *RL-based Distillation:* SuperCorrect (Yang et al., 2025); *Curriculum-based Distillation:* POCL (Liu & Zhang, 2025). To ensure tokenizer matching necessary for white-box distillation, we take DeepSeek-R1-Distill-Qwen-14B and Qwen2.5-7B as the teacher model and student model, respectively. According to corresponding results in Table 4, IOA stably outperforms various recent baselines, demonstrating the effectiveness of IOA even against white-box distillation and on-policy/curriculum-based methods.

**Distillation on Moderately Strong Teacher Models** To evaluate whether our IOA frameworks still performs well under moderately strong teacher model setting, we employ DeepSeek-R1-Distill-Qwen-7B as the teacher and Qwen2.5-3B as the student to conduct the experiments. From the averaged results on Table 5, we can observe that IOA continues to provide consistent improvements over baselines, although the absolute gain is naturally smaller compared to large-gap settings.

**Seed Data Analysis** To explore the influence of seed data quality and quantity on performance, we conduct experiments under two setting: a) We first filter and downsample the high-quality OpenThoughts3-1.2M dataset (Guha et al., 2025) to 3K examples, then use this set as the seed; b) We take samples in our currated $D_{seed}$ as base templates and synthesize 9K samples as the seed. Note in our default setting of main experiments, we directly employ $D_{seed}$ with 3K samples as the seed. Especially on the challenging AIME2024 dataset, improving see data quality (setting a) achieves 27.52, much higher than default setting of 6.29; increasing seed data quantity (setting b) achieves 11.83, moderately higher that default setting. Thus, we conclude that improving seed quality (coverage and distribution) can bring more obvious performance gain than solely increasing amounts.

## 5 CONCLUSIONS AND FUTURE WORKS

In this work, we introduce a pedagogically inspired framework IOA, for data synthesis, in which the prioritized critical knowledge is progressively adapted and delivered to student language models through curriculum structuring and knowledge representation adaptation. Drawing analogies from human learning theories, our approach demonstrates that carefully designed synthetic data can substantially improve the effectiveness and efficiency of language model distillation, as validated by the extensive experiments across multiple instruction following and reasoning benchmarks. In the future, we plan to further integrate insights from cognitive science with language models to establish more systematic principles for curriculum design in LLM distillation.

## ETHICS STATEMENT

This work focuses on improving the effectiveness of language model knowledge distillation through pedagogically inspired data synthesis. All experiments were conducted using publicly available large language models (e.g., LLaMA, Qwen) and openly released benchmarks for instruction-following, reasoning, coding, science QA tasks. To mitigate risks of misuse, our framework emphasizes synthetic data generation without reliance on sensitive or personally identifiable information. We acknowledge that more capable student models distilled from powerful teacher models could potentially be misapplied in harmful contexts; however, we believe that our contributions primarily advance research toward more resource-efficient and accessible AI, reducing costs and environmental footprint.

## REPRODUCIBILITY STATEMENT

We provide detailed descriptions of our methodology, including the Identifier–Organizer–Adapter framework, experimental setups, baselines, and evaluation protocols in the main text and appendices. Hyperparameters, prompting templates, and seed data construction processes are documented to enable replication. All models used are publicly available (e.g., LLaMA, Qwen families), and the benchmarks employed (e.g., DollyEval, VicunaEval, GSM8K, MATH, HumanEval) are open-sourced. To further support reproducibility, we have provided code in the Github repository `https://github.com/BokwaiHo/IOA`.

## ACKNOWLEDGMENTS

This work was supported by the Early Career Scheme (No. CityU 21219323) and the General Research Fund (No. CityU 11220324) of the University Grants Committee (UGC), the NSFC Young Scientists Fund (No. 9240127), and the Donation for Research Projects (No. 9229164 and No. 9220187).

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

CONTENTS

## A    PEDAGOGICAL FOUNDATIONS AND HOW THEY SHAPE IOA

### A.1    THEORETICAL BACKGROUND

**Bloom's Mastery Learning (Bloom, 1968).**  Mastery learning emphasizes criterion-referenced progress: learners advance only after demonstrating sufficient mastery of prerequisite material; otherwise they receive targeted remediation. In our framework this principle appears as stage-wise advancement gates that require the student model to reach a high performance ratio relative to the teacher before moving on, i.e., the mastery gate with threshold $\tau_{\text{mastery}}$.

**Vygotsky's Zone of Proximal Development (ZPD) (Vygotsky & Cole, 1978).**  ZPD emphasizes introducing tasks just beyond the learner's current independent capability while providing scaffolds, thereby controlling the "step size" of difficulty. In IOA this appears as bounded difficulty increments between adjacent curriculum stages regulated by the ZPD threshold $\tau_{\text{ZPD}}$.

Together, these principles motivate our design criteria: topological curricula, bounded difficulty increments, and stage-wise advancement rules, operationalized throughout the pipeline.

### A.2    OPERATIONALIZATION IN IOA

**Identifier → "What to teach":** We diagnose fine-grained knowledge gaps and build a dependency graph over modules so that remediation targets concepts whose mastery most strongly enables downstream progress. This includes thresholds for recognizing mastery of prerequisites ($\tau_{high} = 0.9$, $\tau_{low} = 0.7$) and a dependency inclusion threshold ($\tau_{dep} = 0.3$).

**Organizer → "When to teach":** We construct a topological curriculum over the dependency graph so that prerequisite knowledge is learned before dependents, forming stages $s_1, s_2, \ldots$. Drawing on ZPD, we bound difficulty increments between consecutive stages using:

$$\Delta_{\text{difficulty}}(s_i \to s_{i+1}) \leq \tau_{\text{ZPD}} \cdot \text{avg\_difficulty}(s_i),$$

with $\tau_{\text{ZPD}} = 0.15$, ensuring each step stays within the student's learnable zone.

**Organizer → Mastery gating:** Following Bloom, we require criterion-referenced mastery before progression:

$$\min_{k \in s_i} \frac{P_S(k)}{P_T(k)} \geq \tau_{\text{mastery}},$$

with $\tau_{\text{mastery}} = 0.9$, triggering remedial synthesis if unmet.

**Adapter → "How to teach":** To keep each stage within the ZPD and to facilitate mastery, we *scaffold* representations via: (i) abstract-to-concrete concretization, (ii) reasoning decomposition into atomic steps with verification, (iii) cognitive-load management via controlled instance complexity, (iv) template-based solution formats for stable scaffolds, and (v) linguistic simplification with explicit discourse markers. Full prompt templates for these adaptations appear in Appendix N and O.

### A.3    WHY THESE PRINCIPLES IMPROVE DISTILLATION

In this part, we clarify why pedagogical principles in IOA can benefit language model distillation, even though human learning and neural optimization differ substantially.

**1. Curriculum principles reshape the optimization landscape through training-data distribution.** LLMs are trained by gradient descent, and the gradients they receive are determined entirely by the distribution of training examples. Mastery Learning and ZPD act precisely on this distribution: they regulate which examples appear at each point in training and how difficulty progresses. Although these ideas originate from human education, in IOA they function as *data-level inductive biases* that smooth the optimization trajectory, reduce gradient variance, and prevent the student from being exposed to examples far beyond its current capacity. These effects are known to improve convergence stability in neural models.

**2. Controlling difficulty prevents gradient domination by unsolvable examples.** If a student model is repeatedly trained on examples that are too difficult, the loss surface is dominated by high-error regions that produce large gradients but little useful signal. ZPD's bounded difficulty

step (Eq. 8 prevents this mismatch by ensuring that each new batch of synthesized data remains within a solvable range relative to the student's competence. This avoids gradient explosion, reduces ineffective updates, and keeps the optimization process in a learnable region.

**3. Mastery thresholds align the training sequence with knowledge dependencies.** Advancing to new knowledge only when the student has achieved sufficient mastery on prerequisite modules (Eq. 9 prevents underfitting of foundational skills. For neural networks, this mechanism reduces interference from partially learned concepts and promotes *representation consolidation* before new content is introduced. Such staging is a known driver of sample efficiency in curriculum and self-paced learning frameworks.

**4. Structured ordering reduces destructive interference between tasks.** The Identifier's prerequisite graph ensures that modules that support one another are learned in an order consistent with their dependency relations. For neural architectures, this reduces conflicting gradients between related sub-tasks and enables shared internal representations to develop gradually. This effect parallels observations in multi-task and progressive learning that ordering tasks by dependency improves transfer and reduces forgetting.

**5. Adaptive representation scaffolding improves gradient informativeness.** The Adapter modifies the representation format of synthesized data to better match the student's expressive capacity (Sec. 3.4). This increases the alignment between the student's hypothesis class and the target signals it receives, which reduces loss ambiguity and yields more informative gradients. In gradient-descent terms, this provides a better-conditioned optimization problem.

Overall, these pedagogical principles improve distillation because they induce a *structured, capacity-aware training distribution* that reduces gradient variance, prevents exposure to unsolvable examples, respects knowledge dependencies, and aligns data representations with the student's capabilities. Their benefit arises not from human-like cognition but from the way they shape the optimization landscape through curriculum design.

### A.4 PRACTICAL HYPERPARAMETER GUIDANCE

In our standard configuration we set $\tau_{ZPD} = 0.15$ and $\tau_{mastery} = 0.90$. Hyperparameter sweeps show a non-monotonic trend for $\tau_{ZPD}$ with a peak near 0.15 (too small under-challenges; too large overshoots), and performance for $\tau_{mastery}$ peaks around 0.90–0.95 before very strict values slow progress.

### A.5 SUMMARY: FROM PRINCIPLES TO MECHANISMS

- **Bloom's Mastery Learning** $\rightarrow$ Stage gating + remedial loops (advance only after demonstrating criterion-level competence).

- **Vygotsky's ZPD** $\rightarrow$ Bounded difficulty increments + scaffolding (keep steps learnable; adapt representations and reduce load).

- **Curriculum theory + Prerequisites** $\rightarrow$ Topological sequencing over a dependency graph to respect knowledge structure.

These mechanisms instantiate educational theory in concrete synthesis and training rules, yielding the IOA pipeline that explicitly answers **what** to teach (Identifier), **when** to teach (Organizer with ZPD-bounded progression and Bloom's mastery gates), and **how** to teach (Adapter with scaffolds and cognitive alignment).

## B  SEED DATA CONSTRUCTION

To anchor the synthesis pipeline, we curate a compact seed dataset $D_{seed}$ specifically aligned with four core domains: instruction following, mathematical problem solving, code generation, and academic knowledge reasoning. The design emphasizes *coverage, difficulty stratification, and cleanliness*.

**Instruction Following** We collect ∼800 prompts from open instruction datasets (e.g., Dolly [1], OpenAssistant [2] under permissive licenses) and complement them with author-authored tasks targeting underrepresented skills such as multi-turn dialogue, style transfer, and long-form summarization. All items are filtered to avoid overlap with instruction-tuning evaluation benchmarks such as Dolly Evaluation, Vicuna Evaluation, MT-Bench and AlpacaEval.

**Mathematical Problem Solving** We construct ∼900 problems spanning multiple difficulty levels:

- *Elementary to undergraduate-level:* sampled from openly licensed K–12 and college math repositories (arithmetic, algebra, calculus).

- *Graduate-level and Olympiad-style problems:* re-authored from public-domain collections (e.g., International Mathematical Olympiad archives) and graduate problem sets from open courseware (e.g., MIT OCW [3]). All problems are paraphrased and re-contextualized to avoid leakage into evaluation sets such as GSM8K or MATH.

- *Synthetic templates:* ∼200 programmatically generated algebra and number theory problems with parameterized difficulty.

**Code Generation** We curate ∼700 tasks, combining both toy and realistic programming challenges:

- *Algorithmic and snippet-level tasks:* author-written, inspired by programming textbooks (e.g. "Introduction to algorithms" (Cormen et al., 2022)) and tutorials.

- *Project-level code excerpts:* drawn from permissively licensed open-source repositories on GitHub [4]. We select self-contained functions, modules, or scripts that perform meaningful tasks (e.g., data parsers, configuration utilities, numerical algorithms), avoiding benchmark-style tasks from HumanEval or MBPP.

**Academic Knowledge Reasoning** We compile ∼600 items across science, engineering, and social sciences:

- *Undergraduate-level factual and reasoning questions:* curated from open educational resources (e.g., Wikibooks [5], OpenStax [6] textbooks).

- *Graduate-level or applied science/engineering problems:* adapted from open-access lecture notes and course materials (e.g., control systems, thermodynamics, molecular biology). Problems are paraphrased and checked to avoid overlap with GPQA subject pools.

**Cleaning & Partitioning** Across domains, we enforce strict de-duplication and contamination checks: items with n-gram or semantic similarity to evaluation benchmarks (e.g., Dolly Evaluation, MATH, HumanEval) are excluded. Additional filtering removes toxic or privacy-sensitive content. The dataset is split into train/validation at an 8:2 ratio, with validation reserved for probe tasks. This splitting operation is conducted on the granularity of knowledge modules in the knowledge hierarchy.

**Statistics** The final $D_{\text{seed}}$ consists of ∼3000 items: ∼800 instruction, ∼900 math (including graduate/Olympiad-level), ∼700 code (including project-level snippets), and ∼600 academic reasoning (with university-level science/engineering). Despite its compact scale, the dataset provides broad, stratified coverage that anchors synthesis while enabling fine-grained evaluation.

## C  DETAILED INTRODUCTION TO BASELINES

To help better understand the baseline methods used in our experiments, we provide the corresponding detailed introductions as follows:

- **Self-Instruct** (Wang et al., 2023): Self-Instruct introduces a semi-automated pipeline for creating instruction–response data without heavy human annotation. Starting from a small seed set of

---

[1] https://huggingface.co/datasets/databricks/databricks-dolly-15k

[2] https://huggingface.co/datasets/OpenAssistant/oasst1

[3] https://ocw.mit.edu/

[4] https://github.com/

[5] https://www.wikibooks.org/

[6] https://openstax.org/

manually written tasks, the framework iteratively prompts a pretrained language model to generate new instructions and corresponding input–output instances. Low-quality or redundant samples are filtered using heuristic rules, and the resulting synthetic dataset is then used to finetune the same model. This bootstrapping process enables large-scale data synthesis that aligns pretrained models to follow instructions more effectively through supervised distillation from their own generations.

- **LaMini** (Wu et al., 2024): LaMini proposes a large-scale data distillation framework that creates instruction–response pairs by first collecting diverse queries from publicly available instruction datasets and community sources, then unifying and normalizing them into a standardized format. A strong teacher model is prompted to generate high-quality responses for these queries, yielding a massive synthetic corpus covering varied domains and task types. Smaller student models are then finetuned on this distilled dataset, allowing them to inherit instruction-following ability from the teacher while operating with significantly fewer parameters.

- **Lion** (Jiang et al., 2023): Lion introduces an adversarial distillation framework that synthesizes training data by querying a proprietary teacher model with carefully designed prompts. Instead of relying on static seed instructions, Lion iteratively generates diverse and challenging instruction–response pairs, where the student's weaknesses are identified and exploited to craft adversarial prompts. The teacher model provides outputs for these prompts, forming a synthetic dataset that directly targets areas where the student underperforms. The student is then finetuned on this evolving dataset, enabling knowledge distillation from the teacher through a closed-loop, adversarial data generation process.

- **Distilling Step-by-Step (DSS)** (Hsieh et al., 2023): DSS leverages the reasoning ability of large language models to produce not only task labels but also intermediate natural language rationales via chain-of-thought prompting. These rationales, paired with the predicted labels, are treated as synthetic supervision and used to train smaller student models under a multi-task framework, where the student learns both to predict final answers and to generate rationales. By enriching the distilled dataset with step-by-step explanations, this approach provides denser task knowledge than label-only distillation, allowing compact models to inherit reasoning skills from larger teachers without requiring the teacher model at inference time

- **CasCoD** (Dai et al., 2024): CasCoD proposes a curriculum-based data synthesis framework for distilling large language models. It first generates synthetic instruction–response pairs using a teacher model, but instead of treating all data equally, the method organizes these pairs into a structured curriculum. The curriculum is built by ranking samples according to difficulty and causal relevance, starting from simpler instances and gradually moving toward harder ones. The synthetic dataset is then used to train student models in a staged manner, ensuring that knowledge distillation from the teacher is both effective and stable.

- **CounterDistill** (Feng et al., 2024a): CounterDistill introduces a counterfactual distillation framework that augments training data by systematically editing task instances with the help of a large teacher model. The method first applies topic word extraction and syntactic analysis to identify causal features in the input text, which are then masked and replaced to generate counterfactual examples that closely resemble the original inputs but yield different labels. To further enrich supervision, the teacher model produces multi-view chain-of-thought rationales: positive rationales explaining why the correct answer is valid, and negative rationales refuting each incorrect option. By combining factual and counterfactual data with diverse reasoning paths, the approach synthesizes a rich distillation dataset that helps student models learn causal reasoning patterns and avoid spurious correlations

- **Star-Agents** (Zhou et al., 2024): Star-Agents formulates data synthesis for distillation in a multi-agent framework. A set of specialized teacher agents, each with distinct roles such as planner, executor, or critic, are orchestrated to collaboratively generate high-quality instruction–response data. The interaction among agents allows them to refine prompts, critique outputs, and iteratively improve the synthetic dataset. This diverse and self-improving corpus is then used to distill knowledge into a student model, enabling it to acquire richer capabilities than what could be achieved from single-agent data generation alone.

- **MADA** (Wang et al., 2025a): MADA introduces a multi-perspective distillation framework that synthesizes training data by prompting a teacher model to generate responses under diverse reasoning styles or perspectives. For each input instruction, the teacher produces multiple complementary answers—such as step-by-step explanations, concise summaries, or alternative solution strategies—which are then aggregated to form a richer supervision signal. The student model is distilled

on this multi-view synthetic dataset, allowing it to capture broader reasoning patterns and avoid overfitting to a single response style.

## D    DETAILED INTRODUCTION TO EVALUATION BENCHMARKS AND METRICS

As mentioned in the Section 4.1, we evaluate the language model distillation performance from two perspectives: *instruction following* and *reasoning*, considering they are most attractive LLM capabilities currently. Here, we supplement detailed descriptions of these benchmarks:

- **Dolly Evaluation** (Conover et al., 2023): A benchmark derived from the Dolly dataset, covering diverse instruction-following tasks including open-ended question answering, brainstorming, and text generation. It measures how well a model follows natural instructions without additional context.
- **Vicuna Evaluation** (Chiang et al., 2023): A widely used benchmark for instruction-following models, consisting of 80 multi-turn conversations. Model outputs are compared with reference responses from strong open-source models, making it a proxy for alignment and usability.
- **GSM8K** (Cobbe et al., 2021): A grade-school math word problem dataset with 8.5k questions. It evaluates arithmetic reasoning and step-by-step problem-solving, often requiring multi-hop logical inference.
- **MATH** (Lewkowycz et al., 2022): A large-scale benchmark of high-school to competition-level math problems across algebra, geometry, calculus, and number theory. It tests deeper mathematical reasoning beyond GSM8K.
- **AIME2024** (Mathematical Association of America, 2024): The 2024 American Invitational Mathematics Examination dataset, consisting of challenging competition-style math problems. It pushes models toward precise reasoning under high difficulty, closer to Olympiad-level tasks.
- **HumanEval** (Chen et al., 2021): A code generation benchmark of 164 hand-crafted Python programming problems. Models are required to generate correct, executable solutions given only natural language descriptions.
- **MBPP** (Austin et al., 2021): The "Mostly Basic Programming Problems" dataset of 974 crowd-sourced Python programming tasks. It evaluates code synthesis ability on short, self-contained tasks that resemble beginner-to-intermediate coding exercises.
- **LiveCodeBench** (Jain et al., 2025): A dynamic, automatically updated benchmark for code generation, pulling from real-world programming questions. It is designed to prevent test-set contamination and provides a more realistic measure of coding capability.
- **GPQA-Diamond** (Rein et al., 2024): A subset of the GPQA benchmark that focuses on high-quality, difficult graduate-level questions across science and engineering. It evaluates deep domain knowledge and expert-level reasoning.

Furthermore, we provide the explanations to the evaluation metrics: ROUGE-L and Pass@$k$.

- **ROUGE-L**: A text-overlap metric that measures the longest common subsequence (LCS) between generated and reference answers. It captures fluency and content overlap, and is commonly used for summarization and instruction-following evaluation.
- **Pass@$k$**: A probabilistic evaluation metric that measures the likelihood of obtaining a correct answer within $k$ independent attempts. Formally, given multiple sampled generations, Pass@$k$ is defined as the probability that at least one of the $k$ outputs matches the ground truth solution. Originally popularized in code generation tasks, this metric has been adopted more broadly as a robust measure of reasoning performance, as it captures both the accuracy of individual attempts and the benefits of diverse solution exploration. A higher Pass@$k$ indicates that a model is more capable of eventually producing a correct reasoning path, even if not on the first try.

## E    IMPLEMENTATION DETAILS AND HYPERPARAMETER SETTINGS

All experiments are conducted on multiple clusters, each equipped with $8\times$ Nvidia HGX H20 GPUs (96GB memory), Intel Xeon Platinum 8358 CPUs, and 1.5 TiB RAM. We implement our

Table 6: Comparison among different data synthesis-based distillation approaches on various evaluation benchmarks for remaining student models. OpenAI o1 is taken as the teacher LLM. **Bold** and underline indicate the best and second-best results, respectively. ∗ indicates statistically significant.

| Model | Method | DollyEval | VicunaEval | GSM8K | MATH | AIME2024 | HumanEval | MBPP | LiveCodeBench | GPQA-D |
|---|---|---|---|---|---|---|---|---|---|---|
| Qwen2.5-7B | Undistilled | 27.87 | 26.12 | 41.44 | 7.89 | 2.83 | 25.66 | 34.58 | 17.32 | 9.15 |
| | Self-Instruct | 34.68 | 31.66 | 47.89 | 9.22 | 3.67 | 28.83 | 39.27 | 19.44 | 10.48 |
| | LaMini | 36.44 | 33.45 | 49.48 | 10.06 | 3.75 | 30.03 | 40.49 | 20.02 | 10.85 |
| | Lion | 37.23 | 34.92 | 52.56 | 11.53 | 4.33 | 32.15 | 42.14 | 22.34 | 11.66 |
| | DSS | 37.78 | 35.75 | 53.77 | 12.74 | 4.78 | 33.77 | 43.03 | 23.45 | 12.14 |
| | CasCoD | 38.15 | 36.21 | 54.66 | 13.81 | 5.32 | 34.66 | 43.87 | 24.24 | 12.39 |
| | CounterDistill | 36.76 | 35.38 | 54.04 | 13.29 | 5.47 | 34.35 | 43.35 | 23.89 | 12.25 |
| | Star-Agents | 38.58 | 37.00 | 55.45 | 14.57 | 5.93 | 35.94 | 44.63 | 25.13 | 12.87 |
| | MADA | 38.92 | 37.40 | 56.24 | 15.25 | 6.24 | 36.59 | 45.18 | 25.66 | 13.13 |
| | IOA (Ours) | **40.66*** | **39.13*** | **59.99*** | **17.63*** | **6.99*** | **43.84*** | **50.86*** | **29.54*** | **14.94*** |
| Qwen2.5-14B | Undistilled | 29.57 | 27.72 | 43.64 | 9.09 | 3.33 | 27.26 | 36.28 | 18.42 | 9.95 |
| | Self-Instruct | 36.38 | 33.26 | 50.09 | 10.42 | 4.17 | 30.43 | 40.97 | 20.54 | 11.28 |
| | LaMini | 38.14 | 35.05 | 51.68 | 11.26 | 4.25 | 31.63 | 42.19 | 21.12 | 11.65 |
| | Lion | 38.93 | 36.52 | 54.76 | 12.73 | 4.83 | 33.75 | 43.84 | 23.44 | 12.46 |
| | DSS | 39.48 | 37.35 | 55.97 | 13.94 | 5.28 | 35.37 | 44.73 | 24.55 | 12.94 |
| | CasCoD | 39.85 | 37.81 | 56.86 | 15.01 | 5.82 | 36.26 | 45.57 | 25.34 | 13.19 |
| | CounterDistill | 38.46 | 36.98 | 56.24 | 14.49 | 5.97 | 35.95 | 45.05 | 24.99 | 13.05 |
| | Star-Agents | 40.28 | 38.59 | 57.65 | 15.77 | 6.43 | 37.54 | 46.33 | 26.23 | 13.67 |
| | MADA | 40.62 | 38.99 | 58.44 | 16.45 | 6.74 | 38.19 | 46.88 | 26.76 | 13.93 |
| | IOA (Ours) | **42.39*** | **40.75*** | **62.16*** | **18.89*** | **7.45*** | **45.47*** | **52.56*** | **30.63*** | **15.78*** |
| LLaMA3.1-8B | Undistilled | 25.48 | 23.57 | 37.13 | 6.62 | 2.31 | 22.85 | 32.03 | 15.24 | 8.14 |
| | Self-Instruct | 32.22 | 29.44 | 43.57 | 7.81 | 2.98 | 26.44 | 36.22 | 17.25 | 9.54 |
| | LaMini | 33.57 | 31.25 | 45.25 | 8.53 | 3.23 | 27.65 | 37.34 | 17.92 | 9.91 |
| | Lion | 34.71 | 32.46 | 48.63 | 10.09 | 3.76 | 29.64 | 38.93 | 20.16 | 10.72 |
| | DSS | 35.43 | 33.17 | 49.75 | 11.18 | 4.24 | 31.02 | 39.75 | 21.26 | 11.08 |
| | CasCoD | 35.85 | 33.53 | 50.48 | 12.38 | 4.63 | 31.91 | 40.42 | 21.94 | 11.43 |
| | CounterDistill | 34.99 | 32.72 | 50.06 | 11.93 | 4.85 | 31.55 | 40.11 | 21.63 | 11.28 |
| | Star-Agents | 36.49 | 34.23 | 51.14 | 12.81 | 5.02 | 32.76 | 41.04 | 22.61 | 11.64 |
| | MADA | 37.12 | 34.81 | 51.82 | 13.26 | 5.08 | 33.68 | 41.53 | 23.02 | 11.87 |
| | IOA (Ours) | **38.68*** | **36.41*** | **55.07*** | **15.52*** | **5.97*** | **40.48*** | **46.65*** | **26.08*** | **13.15*** |

method in PyTorch with HuggingFace Transformers, enabling mixed precision training and gradient checkpointing for efficiency. For data synthesis, we set $J_i$ as the uniform value 10 for main experiments. During the training phase, we use the AdamW optimizer with $\beta_1 = 0.9$, $\beta_2 = 0.95$, weight decay $= 0.01$, and gradient clipping at 1.0. A cosine learning rate schedule with 3% warm-up is applied, setting the peak learning rate to 2e−5 for full-parameter fine-tuning (3B models) and 1e−4 for LoRA-based tuning (7/8/14B models). Training is performed with an effective global batch size of 128 using gradient accumulation, and a maximum context length of up to 4096 tokens, depending on the task (4096 for reasoning, 2048 for instruction-following). We train up to 3 epochs over the stage's synthesized data (or until $\min_{k \in s_i} \frac{P_S(k)}{P_T(k)} \geq 0.9$), then advance to next unit. All reported results are averaged over five runs with different random seeds. We further conduct paired t-tests between our method and the best-performing baseline, confirming statistical significance at the 99% confidence level ($p < 0.01$). The code has been provided in the Github repository https://github.com/BokwaiHo/IOA.

## F  EXTENSIVE EXPERIMENTS ON MORE STUDENT MODELS

In addition to the experiment results on Qwen2.5-3B and LLaMA3.2-3B as provided in the main text, we conduct more extensive experiments on student models of Qwen2.5-7B, Qwen2.5-14B, and LLaMA3.1-8B. The corresponding results with OpenAI o1 and DeepSeek-R1 as teacher models are provided in Table 6 and 7, respectively. Across all settings, IOA consistently achieves the best performance, outperforming strong multi-agent, cascade, and adversarial data synthesis baselines by a clear margin. For example, with o1 as teacher, IOA boosts Qwen2.5-7B on GSM8K from 41.44 (undistilled) to 59.99 and HumanEval from 25.66 to 43.84, while similar improvements hold for larger Qwen2.5-14B and LLaMA3.1-8B students. Scaling the student further amplifies the benefits, with IOA gains increasing steadily from 7B to 14B (e.g., GSM8K: 59.99 → 62.16; HumanEval: 43.84 → 45.47). Moreover, teacher choice shapes the performance profile: R1-trained students are

Table 7: Comparison among different data synthesis-based distillation approaches on various evaluation benchmarks for remaining student models. DeepSeek-R1 is taken as the teacher LLM. **Bold** and underline indicate the best and second-best results, respectively. ∗ indicates statistically significant.

| Model | Method | DollyEval | VicunaEval | GSM8K | MATH | AIME2024 | HumanEval | MBPP | LiveCodeBench | GPQA-D |
|---|---|---|---|---|---|---|---|---|---|---|
| **Qwen2.5-7B** | Undistilled | 27.57 | 25.92 | 42.34 | 8.39 | 3.13 | 26.86 | 35.58 | 18.12 | 9.75 |
| | Self-Instruct | 34.38 | 31.46 | 48.79 | 9.72 | 3.97 | 30.03 | 40.27 | 20.24 | 11.08 |
| | LaMini | 36.14 | 33.25 | 50.38 | 10.56 | 4.05 | 31.23 | 41.49 | 20.82 | 11.45 |
| | Lion | 36.93 | 34.72 | 53.46 | 12.03 | 4.63 | 34.35 | 43.14 | 23.14 | 12.26 |
| | DSS | 37.48 | 35.55 | 54.67 | 13.24 | 5.08 | 35.97 | 44.03 | 24.25 | 12.74 |
| | CasCoD | 37.85 | 36.01 | 55.56 | 14.31 | 5.62 | 36.86 | 44.87 | 25.04 | 12.99 |
| | CounterDistill | 36.46 | 35.18 | 54.94 | 13.79 | 5.77 | 36.55 | 44.35 | 24.69 | 12.85 |
| | Star-Agents | 38.28 | 36.80 | 56.35 | 15.07 | 6.23 | 38.14 | 45.63 | 25.93 | 13.47 |
| | MADA | 38.62 | 37.20 | 57.14 | 15.75 | 6.54 | 39.19 | 46.18 | 26.46 | 13.73 |
| | IOA (Ours) | **39.97\*** | **38.39\*** | **61.78\*** | **18.62\*** | **7.57\*** | **46.48\*** | **53.32\*** | **31.13\*** | **16.23\*** |
| **Qwen2.5-14B** | Undistilled | 29.27 | 27.52 | 44.54 | 9.59 | 3.63 | 28.46 | 37.28 | 19.22 | 10.55 |
| | Self-Instruct | 36.08 | 33.06 | 51.00 | 10.92 | 4.47 | 31.63 | 41.97 | 21.34 | 11.88 |
| | LaMini | 37.84 | 34.85 | 52.58 | 11.76 | 4.55 | 32.83 | 43.19 | 21.92 | 12.25 |
| | Lion | 38.63 | 36.32 | 55.66 | 13.23 | 5.13 | 35.95 | 44.84 | 24.24 | 13.06 |
| | DSS | 39.18 | 37.15 | 56.87 | 14.44 | 5.58 | 37.57 | 45.73 | 25.35 | 13.54 |
| | CasCoD | 39.55 | 37.61 | 57.76 | 15.51 | 6.12 | 38.46 | 46.57 | 26.14 | 13.79 |
| | CounterDistill | 38.16 | 36.78 | 57.14 | 14.99 | 6.27 | 38.15 | 46.05 | 25.79 | 13.65 |
| | Star-Agents | 39.98 | 38.40 | 58.55 | 16.27 | 6.73 | 39.74 | 47.33 | 27.03 | 14.27 |
| | MADA | 40.32 | 38.80 | 59.34 | 16.95 | 7.04 | 40.79 | 47.88 | 27.56 | 14.53 |
| | IOA (Ours) | **41.64\*** | **39.97\*** | **63.95\*** | **19.64\*** | **7.89\*** | **47.67\*** | **54.94\*** | **32.20\*** | **17.08\*** |
| **LLaMA3.1-8B** | Undistilled | 25.18 | 23.37 | 38.03 | 7.12 | 2.61 | 24.05 | 33.03 | 16.04 | 8.74 |
| | Self-Instruct | 31.92 | 29.24 | 44.47 | 8.31 | 3.28 | 27.64 | 37.22 | 18.05 | 10.14 |
| | LaMini | 33.27 | 31.05 | 46.15 | 9.03 | 3.53 | 28.85 | 38.34 | 18.72 | 10.51 |
| | Lion | 34.41 | 32.26 | 49.53 | 10.59 | 4.06 | 30.84 | 39.93 | 20.96 | 11.32 |
| | DSS | 35.13 | 32.97 | 50.65 | 11.68 | 4.54 | 32.22 | 40.75 | 22.06 | 11.68 |
| | CasCoD | 35.55 | 33.33 | 51.38 | 12.88 | 4.93 | 33.11 | 41.42 | 22.74 | 12.03 |
| | CounterDistill | 34.69 | 32.52 | 50.96 | 12.43 | 5.15 | 32.75 | 41.11 | 22.43 | 11.88 |
| | Star-Agents | 36.19 | 34.03 | 52.04 | 13.31 | 5.32 | 33.96 | 42.04 | 23.41 | 12.24 |
| | MADA | 36.82 | 34.61 | 52.72 | 13.76 | 5.38 | 34.88 | 42.53 | 23.82 | 12.47 |
| | IOA (Ours) | **38.23\*** | **36.12\*** | **57.21\*** | **16.81\*** | **6.67\*** | **43.07\*** | **48.37\*** | **27.68\*** | **14.07\*** |

particularly strong on reasoning and code (e.g., Qwen2.5-7B GSM8K: 61.78, HumanEval: 46.48), while o1 yields slightly higher instruction-following scores, showing that IOA effectively transfers the strengths of distinct teachers. Finally, the observed improvements generalize across model backbones (Qwen vs. LLaMA) and task categories (instruction following and reasoning including competition math, code, and science QA). Taken together, these results demonstrate that IOA is not only effective but also scalable and broadly generalizable across different student model sizes and structures.

## G  EVALUATION ON AGENTIC TASKS

In addition to experiments on instruction following and reasoning benchmarks employed in the main text, we further evaluate the performance of our proposed IOA framework on agentic tasks, which has attracted more and more attention recently. Specifically, we select $\tau$-Bench (Yao et al., 2024) and BFCL (Patil et al., 2025) benchmarks. $\tau$-Bench evaluates agents' ability to interact with simulated human users and domain-specific APIs under realistic rules (e.g., retail and airline customer service), emphasizing consistency and reliability via the pass@$k$ metric. BFCL benchmarks LLMs' function-calling skills across single-turn, crowd-sourced, multi-turn, and agentic tasks, with a distinctive abstract syntax tree-based validation method that enables scalable and deterministic correctness checking. Especially, for agentic capability distillation, we additionally augment the seed synthesis pool with agentic-oriented exemplars (tool-use/function-calling, multi-turn user–API interactions, and task decomposition patterns) so that the distilled data explicitly covers the behaviors targeted by $\tau$-Bench and BFCL. As shown in Table 8, IOA consistently achieves the highest performance across all student models and both teacher LLMs, while MADA is the strongest among prior baselines. Concretely, withn OpenAI o1 as teacher, IOA improves Qwen2.5-7B from 44.83 (undistilled) to 59.02 on $\tau$-Bench and from 42.55 to 56.48 on BFCL, and similar margins hold for larger Qwen2.5-14B and LLaMA3.1-8B students. With DeepSeek-R1 as teacher, IOA yields further gains on agentic skills, particularly on BFCL which emphasizes function-calling correctness (e.g., Qwen2.5-14B

Table 8: Comparison among different data synthesis-based distillation approaches on agentic benchmarks $\tau$-Bench and BFCL. OpenAI o1 and DeepSeek-R1 are taken as teacher LLMs. **Bold** and underline indicate the best and second-best results, respectively. For clear illustration, Qwen2.5 and LLaMA3.1/3.2 are abbreviated as Q and L, respectively. $*$ indicates statistically significant.

| LLM | SLM | Task | Undistilled | Self-Instruct | LaMini | Lion | DSS | CasCoD | CounterDistill | Star-Agents | MADA | IOA(Ours) |
|---|---|---|---|---|---|---|---|---|---|---|---|---|
| o1 | Q-3B | $\tau$-Bench | 41.22 | 47.36 | 48.15 | 49.28 | 50.14 | 50.66 | 49.87 | 51.72 | 52.48 | 55.63* |
| | | BFCL | 38.97 | 44.28 | 45.16 | 46.24 | 47.05 | 47.66 | 46.93 | 48.42 | 49.33 | 52.11* |
| | Q-7B | $\tau$-Bench | 44.83 | 50.27 | 51.36 | 52.57 | 53.41 | 53.96 | 53.27 | 54.85 | 55.77 | 59.02* |
| | | BFCL | 42.55 | 47.68 | 48.72 | 49.91 | 50.68 | 51.19 | 50.53 | 52.36 | 53.14 | 56.48* |
| | Q-14B | $\tau$-Bench | 46.72 | 52.31 | 53.42 | 54.56 | 55.38 | 55.94 | 55.11 | 56.92 | 57.86 | 61.14* |
| | | BFCL | 44.37 | 49.63 | 50.58 | 51.76 | 52.53 | 53.09 | 52.41 | 54.28 | 55.16 | 58.52* |
| | L-8B | $\tau$-Bench | 40.08 | 45.72 | 46.55 | 47.63 | 48.42 | 49.01 | 48.16 | 49.88 | 50.67 | 53.84* |
| | | BFCL | 37.26 | 42.31 | 43.18 | 44.26 | 45.09 | 45.72 | 44.85 | 46.38 | 47.24 | 50.16* |
| | L-3B | $\tau$-Bench | 36.47 | 41.25 | 42.03 | 43.19 | 44.07 | 44.63 | 43.78 | 45.52 | 46.34 | 49.28* |
| | | BFCL | 34.12 | 39.27 | 40.15 | 41.23 | 42.06 | 42.64 | 41.77 | 43.26 | 44.08 | 46.87* |
| R1 | Q-3B | $\tau$-Bench | 42.35 | 48.46 | 49.27 | 50.42 | 51.26 | 51.84 | 51.05 | 52.96 | 53.81 | 57.09* |
| | | BFCL | 39.78 | 45.12 | 46.08 | 47.28 | 48.09 | 48.68 | 47.93 | 49.52 | 50.46 | 53.77* |
| | Q-7B | $\tau$-Bench | 45.94 | 51.42 | 52.37 | 53.58 | 54.41 | 55.03 | 54.28 | 56.11 | 57.03 | 60.44* |
| | | BFCL | 43.67 | 48.75 | 49.73 | 50.92 | 51.74 | 52.33 | 51.57 | 53.42 | 54.36 | 57.62* |
| | Q-14B | $\tau$-Bench | 47.88 | 53.41 | 54.52 | 55.73 | 56.54 | 57.16 | 56.42 | 58.37 | 59.22 | 62.58* |
| | | BFCL | 45.51 | 50.86 | 51.81 | 53.02 | 53.84 | 54.48 | 53.73 | 55.58 | 56.44 | 59.83* |
| | L-8B | $\tau$-Bench | 41.23 | 46.91 | 47.72 | 48.93 | 49.78 | 50.35 | 49.61 | 51.42 | 52.31 | 55.66* |
| | | BFCL | 38.42 | 43.63 | 44.57 | 45.78 | 46.59 | 47.18 | 46.42 | 48.24 | 49.17 | 52.54* |
| | L-3B | $\tau$-Bench | 37.69 | 42.43 | 43.26 | 44.48 | 45.36 | 45.94 | 45.18 | 46.92 | 47.76 | 50.98* |
| | | BFCL | 35.27 | 40.44 | 41.36 | 42.57 | 43.39 | 43.98 | 43.23 | 44.87 | 45.73 | 48.92* |

Table 9: Supplementary Ablation study on various evaluation benchmarks for LLaMA and Qwen models. OpenAI o1 is taken as the teacher LLM. $*$ indicates statistically significant.

| Model | Method | DollyEval | VicunaEval | GSM8K | MATH | AIME2024 | HumanEval | MBPP | LiveCodeBench | GPQA-D |
|---|---|---|---|---|---|---|---|---|---|---|
| Qwen2.5-3B | Full IOA | 38.16* | 36.83* | 55.79* | 15.53* | 6.29* | 40.64* | 47.86* | 26.94* | 13.74* |
| | - GenerateRemedialData | 37.42 | 35.68 | 53.25 | 14.35 | 5.84 | 39.92 | 46.74 | 26.18 | 13.28 |
| LLaMA3.2-3B | Full IOA | 36.88* | 34.81* | 52.07* | 14.02* | 5.47* | 37.98* | 44.25* | 24.08* | 12.15* |
| | - GenerateRemedialData | 36.15 | 33.92 | 50.42 | 13.25 | 5.08 | 37.31 | 43.28 | 23.41 | 11.76 |

improves from 45.51 to 59.83). These results highlight three consistent trends: (i) **effectiveness** — IOA provides large and robust improvements over both undistilled models and prior distillation methods; (ii) **scalability** — larger students (7B $\rightarrow$ 14B) consistently benefit more; (iii) **generalization** — IOA adapts to different teachers and backbones while excelling on new task types beyond standard instruction following and reasoning. Taken together, these findings confirm that IOA not only distills basic knowledge and reasoning ability effectively, but also enhances students' agentic capabilities in realistic interactive and function-calling scenarios.

## H ABLATION AND ANALYSIS OF GENERATEREMEDIALDATA

**Ablation on GenerateRemedialData.** To isolate the contribution of Mastery-Based Progressive Learning, we ablate the GenerateRemedialData component while keeping the dificient knowledge identification, dependency-driven curriculum and the adaptation mechanisms unchanged. As shown in Table 9, removing remedial data consistently lowers performance across all benchmarks and across both Qwen2.5-3B and LLaMA3.2-3B student models. The full IOA pipeline yields gains of $+2.54$ and $+1.65$ points on GSM8K, $+1.18$ and $+0.77$ points on MATH, and $+0.45$ and $+0.39$ points on AIME2024, respectively. Similar improvements are observed for code-generation tasks such as HumanEval, MBPP, and LiveCodeBench. These results indicate that a fixed curriculum alone is insufficient; targeted remedial examples help the student close the residual gaps that remain after the first pass through a module.

**How often does GENERATEREMEDIALDATA occur in practice?** In our implementation, remedial data generation is invoked only when the student fails to meet the mastery criterion for a module (Eq. 9. Empirically, this condition is triggered infrequently. Across all experiments reported in the main paper,

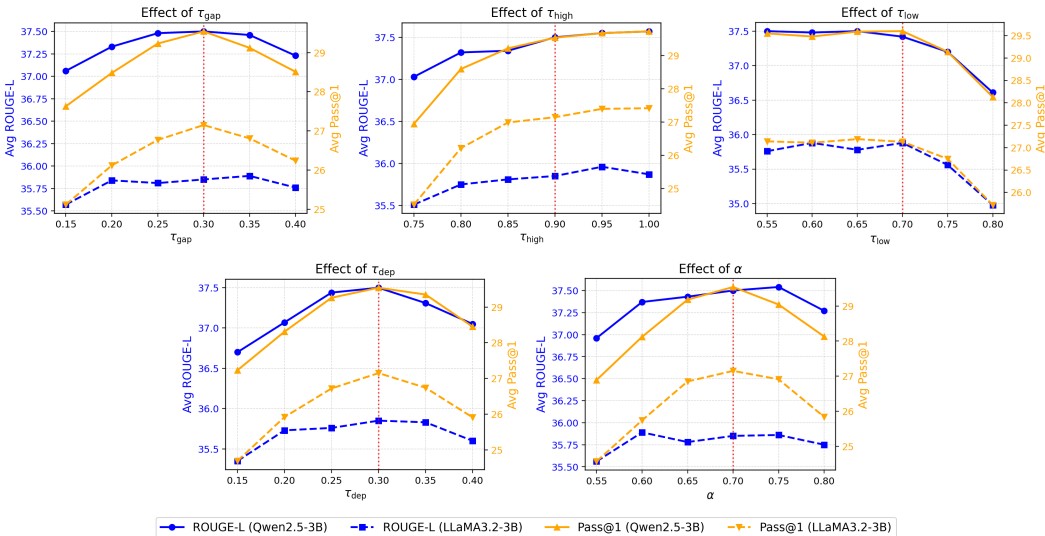

Figure 5: The hyperparameter robustness analysis for other five critical hyperparameters: $\tau_{\text{gap}}$, $\tau_{\text{high}}$, $\tau_{\text{low}}$, $\tau_{\text{dep}}$, $\alpha$ in the Identifier component of our IOA framework.

we observe that the majority (86–92% depending on the student model) of knowledge modules are mastered in a single pass without requiring any remedial iteration. Among the remaining cases, most require exactly one remedial generation step, and only a small fraction ($< 3\%$ of modules) invoke it more than once. This behavior arises naturally from the structure of IOA. The dependency-driven curriculum ensures that modules are presented in an order aligned with prerequisite relations, so by the time a module is introduced, the student has already consolidated the foundational knowledge needed to learn it efficiently. Moreover, the bounded difficulty increment enforced by ZPD (Eq. 8) prevents the student from being exposed to tasks that are too far beyond its current competence, reducing the likelihood that mastery fails on the first attempt.

**Overall impact.** Together with the ablation results, these observations suggest that GenerateRemedialData plays a targeted rather than pervasive role in IOA: it is invoked only when necessary and typically only once per module, yet it yields consistent improvements on both reasoning and coding benchmarks. This demonstrates that Mastery-Based Progressive Learning provides an effective and computationally efficient mechanism for correcting residual weaknesses that would otherwise remain unaddressed in a single-pass curriculum.

## I  ADDITIONAL HYPERPARAMETER ROBUSTNESS ANALYSIS

To further explore the robustness of remaining hyperparameters $\tau_{\text{gap}}$, $\tau_{\text{high}}$, $\tau_{\text{low}}$, $\tau_{\text{dep}}$, and $\alpha$ in the Identifier component, we conduct additional robustness experiments by varying each hyperparameter value while keeping all others fixed. Figure 5 reports the results for both Qwen2.5-3B and LLaMA3.2-3B student models.

Across all five hyperparameters, we observe a consistent pattern: IOA maintains stable performance over a broad range of values, with degradation occurring only at extreme settings. For $\tau_{\text{gap}}$ and $\tau_{\text{dep}}$, performance improves as the dependency threshold increases from very small values and plateaus around 0.25–0.35, indicating that overly permissive thresholds introduce noisy or spurious dependencies while overly strict thresholds remove useful prerequisite structure. Similarly, $\tau_{\text{high}}$ exhibits a broad optimum between 0.85 and 0.95, reinforcing that IOA benefits from a strict but not extreme high-mastery criterion. Here, a stricter threshold (e.g., increasing $\tau_{\text{high}}$ from 0.9 to 0.95) does lead to slightly more remedial iterations, but the additional computational overhead is modest, typically within 3–5% of the total training time, and does not obviously affect the practicality of IOA.

For $\tau_{\text{low}}$, both models show declining performance when the threshold is too high (e.g., 0.8), as this causes insufficient separation between well-mastered and poorly-mastered modules. Performance stabilizes near the default value of 0.7 and remains robust $\leq 0.75$. Finally, the severity parameter

Table 10: Exploration on the combination of IOA and on-policy ditillation method: GKD. DeepSeek-R1-Distill-Qwen-14B and Qwen2.5-7B are taken as the teacher and student model, respectively. **Bold** indicates the best results. ∗ indicates statistically significant.

| Model | Method | DollyEval | VicunaEval | GSM8K | MATH | AIME2024 | HumanEval | MBPP | LiveCodeBench | GPQA-D |
|---|---|---|---|---|---|---|---|---|---|---|
| | IOA | 41.64 | **39.97*** | 63.95 | 19.64 | 10.89 | 47.67 | 54.94 | 32.20 | 17.08 |
| **Qwen2.5-7B** | GKD | 38.62 | 37.20 | 57.14 | 15.75 | 6.54 | 39.19 | 48.16 | 26.46 | 13.73 |
| | IOA+GKD | **42.85*** | 38.92 | **65.43*** | **20.78*** | **11.62*** | **49.25*** | **56.38*** | **34.51*** | **17.86*** |

$\alpha$ demonstrates a smooth unimodal trend with an optimum near 0.7; lower values under-emphasize deficient modules, while overly large values may suppress important but less-challenging knowledge.

Overall, these results indicate that IOA is not sensitive to precise hyperparameter choices: each hyperparameter has a wide region where performance is stable, and the defaults used in the main paper lie near the center of these regions. This further confirms that the Identifier component is controlled by coarse stability regulators rather than fine-grained, task-specific tuning knobs.

## J ROBUSTNESS TO HIERARCHICAL KNOWLEDGE DECOMPOSITION

To examine how sensitive IOA is to the quality of the hierarchical knowledge decomposition extracted from the teacher model, we conduct a set of controlled perturbation experiments. Specifically, we introduce structural noise by (i) randomly merging a fraction of neighboring knowledge modules into coarser units, and (ii) randomly splitting some modules into finer modules. These perturbations directly disturb the dependency graph and the scope of each module, thereby testing whether IOA critically relies on a perfectly decomposed hierarchy.

Across both Qwen2.5-3B and LLaMA3.2-3B students, we observe that IOA's performance degrades only mildly under moderate levels of perturbation. Random merging causes a slight reduction in reasoning tasks due to less granular prerequisite structure, while random splitting occasionally increases the number of small modules requiring mastery. However, in all settings, the relative performance drop remains small (typically $< 2\%$ across benchmarks), and IOA continues to outperform baselines that do not use hierarchical decomposition at all.

These findings indicate that IOA does not depend on a perfectly accurate knowledge hierarchy; instead, it is robust to reasonable structural noise. The Identifier's performance-based dependency scoring and the Organizer's mastery gating help stabilize the curriculum even when the underlying decomposition is perturbed. This confirms that IOA benefits from the hierarchical structure but is not overly sensitive to its exact form.

## K EXPLORATION ON COMBINING ON-POLICY DISTILLATION AND IOA

To explore whether IOA is complementary to on-policy distillation, we conduct a study combining IOA with GKD (Agarwal et al., 2024). Following the setting in the supplementary experiments of main text, we employ DeepSeek-R1-Distill-Qwen-14B and Qwen2.5-7B as teacher and student model respectively, to ensure the on-policy logit distillation feasible. As shown in Table 10, GKD alone underperforms IOA across all benchmarks, but incorporating GKD on top of IOA yields further improvements over IOA on nearly every evaluation task, including GSM8K ($63.95 \rightarrow 65.43$), MATH ($19.64 \rightarrow 20.78$), AIME2024 ($10.89 \rightarrow 11.62$), and HumanEval ($47.67 \rightarrow 49.25$). This suggests that IOA and on-policy distillation operate on different aspects of the language model distillation process. Specifically, IOA provides a structured, capacity-aware curriculum for what and when to learn, while GKD offers fine-grained corrective feedback on student rollouts from the objective side. Therefore, such two methods can be combined to produce complementary gains. This result confirms that IOA is compatible with and can further enhance on-policy distillation methods.

## L KNOWLEDGE DISTRIBUTION OF SEED AND SYNTHESIS DATASETS

Figure 6 compares the knowledge composition of the original seed set with the dataset synthesized by our IOA. Note that on the basis of the standard seed dataset in Appendix B, we supplement 500

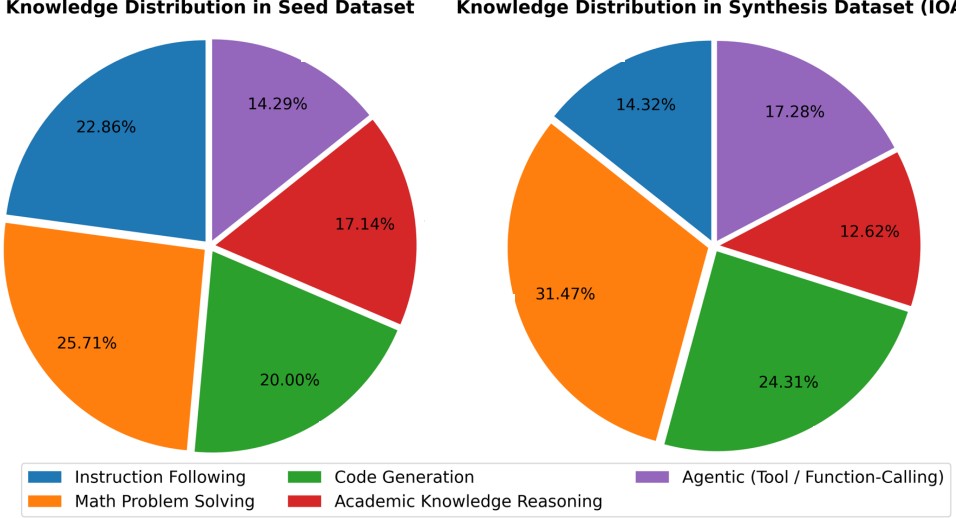

Figure 6: The knowledge distribution of original seed dataset and synthesis dataset by IOA.

items of tool using and function calling data corresponding to Appendix G. The seed set concentrates more on instruction following and math problem solving, with shares of 22.86% and 25.71%, followed by code generation (20.00%), academic knowledge reasoning (17.14%), and agentic/tool use (14.29%). IOA reshapes this mix toward capabilities where small LMs exhibit larger deficits: math increases to 31.47% (+5.76 pp), code to 24.31% (+4.31 pp), and agentic to 17.28% (+2.99 pp), while instruction following decreases to 14.32% (-8.54 pp) and academic knowledge to 12.62% ($-4.52$ pp). Overall, 13.06 percentage points are reallocated from generic instruction/encyclopedic QA to reasoning-, code-, and tool-centric data. This targeted yet diversified shift is produced by IOA's **diagnose-then-synthesize** pipeline—which favors domains with larger teacher–student disagreement and uncertainty (i.e., math, code, and agentic behaviors). Consistent with this redistribution, IOA delivers the strongest empirical gains on reasoning, coding, and agentic benchmarks (Table 1, 2, 6, 7, 8), indicating that the diagnosis module steers synthesis toward the student's most pronounced weaknesses while maintaining balanced coverage across knowledges.

## M    KNOWLEDGE HIERARCHY GENERATION PROMPTS

To facilitate the reproduction of hierarchical knowledge decomposition in Section 3.2, we provide the corresponding prompt as follows.

---

**Prompt 1: Knowledge Hierarchy Generation**

Please decompose the following learning domain into a hierarchical structure of fine-grained knowledge modules:
Domain: {domain}
Description: {description}
Please refer to above description and create 25-35 knowledge modules organized by category (not limited to above mentioned in the description), each with:
- Unique ID in format "category/module"
- Category name
- Descriptive module name
- Difficulty level
Return as a JSON array.  Example format:  [ {"id": "algebra/linear_equations", "category": "Algebra", "name": "Linear Equations", "difficulty": "introductory"}, {"id": "algebra/quadratic_equations", "category": "Algebra", "name": "Quadratic Equations", "difficulty": "intermediate"} ]

---

## N    Knowledge Representation Adaptation Prompts

To operationalize the five adaptation dimensions described in Section 3.4, we design explicit prompting templates for the teacher model. These templates ensure that the synthesized data is adapted to the cognitive capacity of student models.

---

**Prompt 2: Abstract Concept Concretization**

Explain the abstract concept of [CONCEPT] using a concrete analogy or real-world example (e.g., speed of a car for derivatives). Avoid formal definitions at first, and gradually transition to the symbolic or mathematical expression.

---

**Prompt 3: Complex Reasoning Decomposition**

Solve the problem step by step. Break down the reasoning into small sub-steps: (1) Extract relevant information; (2) Identify relationships; (3) Formulate equations; (4) Solve step by step; (5) Verify the solution. Provide each step explicitly.

---

**Prompt 4: Cognitive Load Management**

Reformulate the problem into a simpler version of the same type (e.g., start with a $2 \times 2$ system of equations before moving to larger systems). Ensure each sub-problem is self-contained and introduce incremental difficulty only after demonstrating mastery.

---

**Prompt 5: Representation Format Optimization**

Present the solution in a consistent, structured format using the following template: Step 1: [Action]; Step 2: [Action]; Step 3: [Action]. Use the same template across multiple examples to highlight structural patterns while varying the surface details.

---

**Prompt 6: Linguistic Complexity Reduction**

Rewrite the explanation of [PROBLEM] in simpler language. Use short, direct sentences. Replace advanced terms with simpler synonyms where possible. Use clear connectors such as "first", "next", "therefore". Ensure the reasoning remains correct but linguistically accessible.

---

These prompts are applied in practice to generate pedagogically adapted synthetic data for each curriculum stage, ensuring that the student model receives content that is both comprehensible and incrementally challenging.

## O    Practical Prompting Guidelines for Data Synthesis

### O.1    Relationship to Appendix H

Appendix O provides the **canonical, atomic** prompt templates for the five adaptation dimensions described in Section 3.4: (1) Abstract Concept Concretization, (2) Complex Reasoning Decomposition, (3) Cognitive Load Management, (4) Representation Format Optimization, and (5) Linguistic Complexity Reduction. These templates specify what kind of representation adaptation the teacher LLM should apply.

This section complements Appendix O by operationalizing these atomic templates into full prompting protocols that can be executed across curriculum stages. Concretely, we (i) integrate the five templates into a static **System Prompt** that encodes global pedagogical constraints, (ii) supply a stage-specific **User Prompt** carrying curriculum context and difficulty bounds, and (iii) enforce a machine-checkable **JSON output schema** for automatic filtering, verification, and training.

## O.2 PROMPTING FRAMEWORK OVERVIEW

We adopt a two-layer prompting stack to synthesize pedagogically adapted distillation data from teacher LLMs. The stack separates global, invariant constraints (System Prompt) from stage-specific context (User Prompt), and couples generation with a machine-checkable output contract.

- **System Prompt (static):** Encodes the five adaptation dimensions from Appendix N as hard constraints applicable to all synthesis calls. These constraints ensure consistent representation adaptation regardless of curriculum stage.

- **User Prompt (per stage):** Carries the current curriculum stage $s_i$ and its context: target knowledge units, prerequisite set, and the difficulty/size caps determined by the Organizer. It may further include the student's baseline ratio (relative to the teacher) and the requested number of items, enabling the teacher to calibrate instance complexity and coverage at this stage.

**Execution protocol.** For each curriculum stage $s_i$, we: (1) instantiate the static System Prompt once; (2) fill a stage-specific User Prompt with *(i)* `stage_id`, `knowledge_units`, `prereqs`, `difficulty/size_caps`, and `num` examples; and *(ii)* any stage-level guardrails such as "start from concrete analogy then transition to symbolic form," "explicit step-by-step reasoning," and "self-verification." (3) issue a synthesis call and collect candidates; (4) run automatic checks (schema validation, verification success, difficulty bounds, stage alignment); (5) discard and regenerate any failing items until the target count is met. We optionally apply semantic deduplication to maintain diversity across items while keeping template consistency.

**Output contract.** The teacher LLM must return a JSON array that conforms to the schema in Appendix O.5 (all required keys present, types correct). Each item must include: the targeted module and prereqs, the problem (analogy $\rightarrow$ formal), a stepwise solution with a canonical final answer, an independent verification routine, and adapter flags that document how the five dimensions were applied. Any item that fails verification, violates caps, or breaks the schema is discarded and regenerated. This contract enables downstream automatic filtering and reliable training set construction.

**Remedial and bridging hooks.** If mastery is not achieved for $s_i$, we trigger the Remedial prompt to generate simpler, tightly scoped items on the weak sub-skills; upon mastery, we trigger the Bridging prompt to gently increase complexity to the next difficulty notch while preserving the five adaptation requirements (see Appendix I.6). This keeps difficulty increments bounded and aligned with the Organizer's pacing.

**Reproducibility notes.** We log the exact System/User prompts and random seeds, enforce stable stepwise formatting, and reject items with missing verification, out-of-range difficulty, or stage misalignment (see Appendix O.8). These practices make the prompting framework robust and replicable across runs and stages.

## O.3 SYSTEM PROMPT TEMPLATE

Users can copy and paste the following contents as System Prompt:

> **Prompt 7: System Prompt Template**
>
> You are a teacher LLM generating pedagogically adapted synthetic data for a student model (SLM). Your goal is to align knowledge representation with the student's cognitive capacity.
>
> Strictly enforce the following adaptation requirements (see Appendix N for the canonical templates): 1) Abstract Concept Concretization: begin with concrete analogies before formalism. 2) Complex Reasoning Decomposition: present explicit, small-step reasoning. 3) Cognitive Load Management: start minimal and increase difficulty gradually. 4) Representation Format Optimization: use a consistent stepwise template. 5) Linguistic Complexity Reduction: prefer simple words, short sentences, and clear connectors (e.g., "first", "next", "therefore").
>
> If reasoning or verification fails, discard the example and regenerate. All outputs MUST follow the JSON schema provided by the user prompt.

## O.4 USER PROMPT TEMPLATE (PER CURRICULUM STAGE)

Users need to fill the placeholders below and then use it as the User Prompt at stage $s_i$:

> **Prompt 8: User Prompt Template (Per Curriculum Stage)**
>
> Target Domain: {DOMAIN}
> Curriculum Stage: {STAGE_ID}
> Knowledge Units: {K_MODULES}
> Prerequisites: {PREREQS}
> Student Baseline (relative to teacher): {BASELINE_RATIO}
>
> Please generate {NUM} new synthetic examples adapted to this stage.
>
> Requirements:
> - Obey the five adaptation dimensions (Appendix N).
> - Cognitive load constraints: max problem size {SIZE_CAP}, max symbolic/arith.
> complexity {COMPLEXITY_CAP}.
> - Provide a problem that transitions from concrete analogy to symbolic form.
> - Provide a full solution with explicit step-by-step reasoning and verification.
> - Output MUST conform to the JSON schema below.

## O.5 JSON OUTPUT SCHEMA

Listing 1: Schema (all keys required unless marked optional)

```
[
  {
    "module": "<knowledge unit, e.g., 'Algebra/Linear-Equations
        '>",
    "prereq": ["<prereq1>", "<prereq2>"],
    "difficulty_tag": "<introductory|interdiate|advanced>",
    "problem": "<text: concrete analogy -> symbolic formulation>",
    "solution": {
      "steps": ["Step 1: ...", "Step 2: ...", "..."],
      "final_answer": "<canonical answer>",
      "verification": "<independent check; describe or show test>"
    },
    "adapter_flags": {
      "concretization": true,
      "decomposition": true,
      "cognitive_load": {
        "scale": "<e.g., '2x2 system' or 'small input size'>",
        "notes": "<what was simplified and why>"
      },
      "format_template": "Stepwise-3",
      "simplified_language": true
    },
    "metadata": {
      "stage_id": "{STAGE_ID}",
      "seed_style_ref": "{SEED_SET_ID}"   // optional: link to
          seed style
    }
  }
]
```

## O.6   REMEDIAL AND BRIDGING PROMPTS

---

**Prompt 9: Remedial (when mastery not achieved)**

The student did NOT achieve mastery for {STAGE_ID}/{K_MODULES}. Generate {NUM} simplified remedial examples focusing ONLY on these weak sub-skills: {WEAK_SUBSKILLS_LIST}

Constraints:
- Reduce linguistic and structural complexity further.
- Keep instance size minimal; remove distracting details.
- Maintain explicit step decomposition and verification.
- Use the same JSON schema.

---

**Prompt 10: Bridging (after mastery)**

The student ACHIEVED mastery for {STAGE_ID}/{K_MODULES}. Generate {NUM} bridging examples with SLIGHTLY increased complexity (only one notch higher in scale/coefficients/constraints).

Constraints:
- Bounded difficulty increments; do NOT skip levels.
- Keep the five adaptation requirements.
- Use the same JSON schema.

---

## O.7   ILLUSTRATIVE MINI-TEMPLATES

**Mathematics (Quadratic Roots via Discriminant).**

- **Problem scaffold (analogy $\rightarrow$ formal):** Intuitive change/area analogy, then define $a, b, c$, compute $\Delta = b^2 - 4ac$, apply the quadratic formula, and check by substitution.

- **Step template:** Step 1 Identify coefficients $\rightarrow$ Step 2 Compute discriminant $\rightarrow$ Step 3 Apply formula $\rightarrow$ Step 4 Verify by substitution.

**Programming (String Processing).**

- **Problem scaffold:** "Assembly line" analogy for per-character processing $\rightarrow$ pseudocode $\rightarrow$ implementation.

- **Verification:** Minimal runnable unit tests (small, then slightly larger inputs).

## O.8   IMPLEMENTATION NOTES AND REPRODUCIBILITY CHECKLIST

- **Hard filtering:** Discard any item whose verification fails or is missing.

- **Schema validation:** Enforce the JSON schema keys/types; reject items with missing keys or out-of-range difficulty.

- **Stage alignment:** Ensure examples use only permitted knowledge units and prerequisites for stage $s_i$; apply gentle difficulty increments across stages.

- **Style stability:** Keep the same stepwise format across examples to facilitate pattern extraction by the student model.

- **Data logs:** Record the exact prompts (System/User) and random seeds to enable replication.

## P   CASE STUDIES

To illustrate how our pedagogically inspired synthesis framework operates in practice, we provide five representative synthetic data cases. Each example is instantiated in the unified JSON schema (Appendix O) and annotated with the adapter flags (Appendix N). Beyond presenting the raw cases, we also analyze how they reflect our methodological design choices: (i) grounding abstract knowledge

in concrete analogies, (ii) stepwise decomposition with verifiable outputs, (iii) explicit control of difficulty and cognitive load, and (iv) curricular progression across introductory/intermediate/advanced stages.

Listing 2: Algebra: solving a 2×2 linear system (introductory stage).

```
[
  {
    "module": "Algebra/Linear-Equations",
    "prereq": ["Arithmetic/Integers", "Algebra/Variable-
        Manipulation"],
    "difficulty_tag": "introductory",
    "problem": "You and your friend together have 7 apples, and
        you have 1 more than your friend. Translate this into
        equations (x + y = 7, x - y = 1) and solve for both x (
        your apples) and y (your friend's apples).",
    "solution": {
      "steps": [
        "Step 1: Translate the story into equations: x + y = 7 and
            x - y = 1.",
        "Step 2: Add the two equations to get 2x = 8, thus x = 4."
            ,
        "Step 3: Substitute back into x + y = 7, thus y = 3.",
        "Step 4: Verify: x - y = 4 - 3 = 1, which is correct."
      ],
      "final_answer": "x = 4, y = 3",
      "verification": "Both equations are satisfied with the
          solution."
    },
    "adapter_flags": {
      "concretization": true,
      "decomposition": true,
      "cognitive_load": { "scale": "2x2 system", "notes": "Small
          integer coefficients to reduce difficulty" },
      "format_template": "Stepwise-3",
      "simplified_language": true
    },
    "metadata": { "stage_id": "Math-S1", "seed_style_ref": "Seed-
        Math-001" }
  }
]
```

This case demonstrates our concretization principle: everyday scenarios (apples) act as a bridge to formal algebra. The solution is broken into atomic steps (translation → elimination → substitution → verification), ensuring decomposability and minimizing working memory burden. The cognitive load flag restricts complexity to a 2×2 system with integer coefficients, aligning with the Zone of Proximal Development by keeping the entry at an introductory level.

Listing 3: Calculus: derivative at a point (intermediate stage).

```
[
  {
    "module": "Calculus/Derivative-At-Point",
    "prereq": ["Algebra/Functions", "Limits/Intuition"],
    "difficulty_tag": "intermediate",
    "problem": "Connect the idea of a car's speedometer (
        instantaneous velocity) to the derivative of f(x) = 3x^2 -
        2x at x = 4. Provide both symbolic differentiation and a
        numerical approximation check.",
    "solution": {
```

```
 8          "steps": [
 9            "Step 1: Analogy to formal concept: instantaneous velocity
                 for derivative at a point.",
10            "Step 2: Compute derivative: f'(x) = 6x - 2, thus f'(4) =
                 24 - 2 = 22.",
11            "Step 3: Approximate numerically: with h = 0.001, (f(4+h)
                 - f(4))/h = 22."
12          ],
13          "final_answer": "22",
14          "verification": "Symbolic result and numerical approximation
                 are consistent."
15        },
16        "adapter_flags": {
17          "concretization": true,
18          "decomposition": true,
19          "cognitive_load": { "scale": "single derivative", "notes": "
                 Quadratic function only, avoids chain rule" },
20          "format_template": "Stepwise-3",
21          "simplified_language": true
22        },
23        "metadata": { "stage_id": "Math-S2", "seed_style_ref": "Seed-
                 Math-012" }
24    }
25  ]
```

Here we illustrate bridging: the analogy of speed directly scaffolds the abstract idea of a derivative. The dual representation (symbolic derivative and numerical approximation) provides a two-path validation mechanism, strengthening student confidence. The cognitive load is carefully capped (quadratic only, single point evaluation), so that the task constitutes a manageable step up from algebra.

Listing 4: Programming: string normalization (introductory stage).

```
 1  [
 2    {
 3        "module": "Programming/String-Processing/Normalize-Spaces",
 4        "prereq": ["Programming/Python/Basics"],
 5        "difficulty_tag": "introductory",
 6        "problem": "Implement a function normalize_spaces(s) that
                 removes leading/trailing spaces and ensures words are
                 separated by exactly one space.",
 7        "solution": {
 8          "steps": [
 9            "Step 1: Strip leading and trailing whitespace.",
10            "Step 2: Split string into words by whitespace.",
11            "Step 3: Join words with a single space.",
12            "Step 4: Write minimal unit tests for empty string, single
                 word, and multiple spaces."
13          ],
14          "final_answer": "def normalize_spaces(s: str) -> str:\n
                 parts = s.split()\n    return \" \".join(parts)",
15          "verification": "assert normalize_spaces('  a   b  ') == 'a
                 b'; assert normalize_spaces('') == ''; assert
                 normalize_spaces('x') == 'x'"
16        },
17        "adapter_flags": {
18          "concretization": true,
19          "decomposition": true,
20          "cognitive_load": { "scale": "small input size", "notes": "
                 No regex or advanced constructs" },
```

```
21        "format_template": "Stepwise-3",
22        "simplified_language": true
23      },
24      "metadata": { "stage_id": "Code-S1", "seed_style_ref": "Seed-
          Code-010" }
25    }
26  ]
```

This case embodies machine-verifiable decomposition: steps are specified as "strip → split → rejoin," each independently testable. By attaching unit tests inside the JSON, the case enforces explicit verifiability, a key design for reducing hallucination in distilled student models. Moreover, cognitive load is constrained by avoiding regex or edge-case-heavy logic, enabling an accessible intro programming entry.

Listing 5: Instruction Following: polite rescheduling email (intermediate stage).

```
1  [
2    {
3      "module": "Instruction/Email/Reschedule-Meeting",
4      "prereq": ["Instruction/Tone-Politeness", "Instruction/
          Constraint-Tracking"],
5      "difficulty_tag": "intermediate",
6      "problem": "Write a short, polite email in English to
          reschedule a 30-minute 1:1 meeting originally on Wednesday
           10:00. Suggest moving it to Thursday or Friday at the
          same time, explain that you have a medical appointment
          conflict, and include an apology and closing.",
7      "solution": {
8        "steps": [
9          "Step 1: Identify required elements: recipient, reason,
              apology, alternatives, closing.",
10         "Step 2: Draft the email under 120 words with polite tone.
              ",
11         "Step 3: Ensure all checklist items are present."
12        ],
13        "final_answer": "Subject: Request to Reschedule Our 1:1\n\
            nHi <Name>,\nI have a medical appointment that conflicts
             with our Wednesday 10:00 meeting. Could we reschedule
            our 30-minute 1:1 to Thursday 10:00 or Friday 10:00 this
             week? Sorry for the inconvenience, and I appreciate
            your flexibility.\n\nThanks,\n<Sender>",
14        "verification": "Checklist satisfied: {reason, apology, two
             alternatives, recipient, courteous closing}."
15      },
16      "adapter_flags": {
17        "concretization": true,
18        "decomposition": true,
19        "cognitive_load": { "scale": "short-form template", "notes":
              "Fixed structure reduces planning load" },
20        "format_template": "Stepwise-3",
21        "simplified_language": true
22      },
23      "metadata": { "stage_id": "IF-S2", "seed_style_ref": "Seed-IF
          -023" }
24    }
25  ]
```

This illustrates how format templates and constraint-based verification function within our framework. Instead of relying on vague stylistic evaluation, the output is judged against a structured checklist

(recipient present, apology included, alternatives given, courteous tone). The scaffold reduces ambiguity and fosters transferable patterns in instruction-following tasks. Difficulty is set to intermediate level: learners must combine multiple constraints but still within a short-form template.

Listing 6: Physics: Ohm's law in series circuit (advanced stage).

```
1  [
2    {
3      "module": "Physics/Electricity/Ohms-Law-Series",
4      "prereq": ["Physics/Units", "Algebra/Linear-Equations"],
5      "difficulty_tag": "advanced",
6      "problem": "Two resistors R1 = 2 ohms  and R2 = 3 ohms are in
           series with total voltage V = 10V. Compute the total
           current and the voltage drop across each resistor.",
7      "solution": {
8        "steps": [
9          "Step 1: Compute total resistance RT = R1 + R2 = 5 ohms.",
10          "Step 2: Apply Ohm's Law: I = V / RT = 10 / 5 = 2A.",
11          "Step 3: Voltage drops: V1 = I * R1 = 4V; V2 = I * R2 = 6V
               .",
12          "Step 4: Verify: V1 + V2 = 10V, consistent with supply."
13        ],
14        "final_answer": "Current = 2A; Voltage drops = (V1 = 4V, V2
             = 6V)",
15        "verification": "Equation balance and units are consistent."
16      },
17      "adapter_flags": {
18        "concretization": true,
19        "decomposition": true,
20        "cognitive_load": { "scale": "two resistors", "notes": "No
             parallel or AC cases" },
21        "format_template": "Stepwise-3",
22        "simplified_language": true
23      },
24      "metadata": { "stage_id": "Sci-S3", "seed_style_ref": "Seed-
           Sci-005" }
25    }
26  ]
```

This case highlights incremental difficulty escalation: after algebraic prerequisites and prior exposure to circuit basics, students are introduced to multi-component reasoning. The analogy (water flow) grounds the physics concept, while the verification step enforces dimensional analysis and conservation (V1 + V2 = V). This aligns with our curricular progression principle, as it sits at the advanced level but still keeps complexity bounded (two resistors only).

## Q  DISTINGUISHING INTRINSIC DIFFICULTY, PREREQUISITE FAILURE, AND COGNITIVE LOAD

Equ. 8 in the main paper uses the student performance $P_S(k)$ as a signal for controlling the difficulty increments across consecutive learning stages. We clarify that $P_S(k)$ is not interpreted as the *intrinsic difficulty* of a knowledge module. Instead, it represents the *student-perceived difficulty*, i.e., how challenging a module is for the current student model at its present mastery level. This interpretation aligns with Vygotsky's Zone of Proximal Development, where difficulty is defined relative to the learner's current capability rather than as an absolute property of content. By treating $P_S(k)$ as a proxy for developmental appropriateness, the difficulty constraint in Equ. 8 ensures that the progression of modules remains within the student's effective learning region.

This distinction is important because task difficulty and model performance are not equivalent in general: a model may fail either because the module is inherently challenging or because prerequisite knowledge is missing. Our IOA framework already separates these two cases. Missing foundational

knowledge is handled by the dependency graph extracted during the Identifier stage and enforced through the curriculum ordering; a module is never introduced until all of its prerequisites have been mastered (as shown in Equ. 7). Therefore, when Equ. 8 is evaluated, the student has already satisfied prerequisite knowledge, and a low $P_S(k)$ at that point reflects excessive cognitive load rather than missing dependencies.

## R    EXPLANATIONS FOR EXTREME HYPERPARAMETER SETTINGS

To further clarify the behavior of IOA under extreme hyperparameter choices, we provide here additional explanations for the two edge cases: $\tau_{\text{mastery}} = 1.0$ and $\tau_{\text{ZPD}} = 0$.

**Case 1: $\tau_{\textbf{mastery}} = 1.0$.** Requiring perfect mastery before advancing to the next module is generally infeasible in black-box distillation. Even small stochastic variation or inherent capacity gaps between the teacher and student may prevent the student from ever matching the teacher's performance exactly, causing GenerateRemedialData to be repeatedly invoked without convergence. Empirically, as also reflected in Figure 3, pushing $\tau_{\text{mastery}}$ too close to 1.0 yields diminishing performance gains while substantially increasing computational overhead due to additional remedial iterations. This aligns with Bloom's Mastery Learning principle, which advocates high but not absolute mastery thresholds (typically 80–95%). The default range used in IOA (0.85–0.95) follows this pedagogical recommendation and offers a favorable balance between stability and efficiency.

**Case 2: $\tau_{\textbf{ZPD}} = 0$.** In Equ. 8, $\tau_{\text{ZPD}}$ controls the maximum allowable increase in student-perceived difficulty between consecutive stages. Setting $\tau_{\text{ZPD}} = 0$ does not eliminate difficulty control; instead, it enforces a zero-increase constraint, requiring each new module to be no more difficult than the previous one. This results in a non-increasing or even reversed difficulty progression, directly contradicting Vygotsky's Zone of Proximal Development, which emphasizes gradual upward difficulty growth to maintain productive learning. Consistent with this interpretation, the second subfigure of Figure 3 shows that performance degrades significantly as $\tau_{\text{ZPD}} \to 0$, reflecting that the student is no longer exposed to appropriately challenging modules. Therefore, maintaining $\tau_{\text{ZPD}} > 0$ is essential for a pedagogically aligned and effective curriculum.

Overall, these observations confirm that extreme settings of $\tau_{\text{mastery}}$ or $\tau_{\text{ZPD}}$ break key pedagogical assumptions underlying IOA and lead to undesirable optimization behavior. This further validates the ranges adopted in our main experiments.

## S    ANALYSIS OF CATASTROPHIC FORGETTING IN MASTERY-BASED PROGRESSIVE LEARNING

To examine whether Mastery-Based Progressive Learning induces catastrophic forgetting, we monitor the stability of knowledge acquired in earlier curriculum stages. Specifically, we track the ratio $\frac{P_S(k')}{P_T(k')}$ for a subset of previously mastered modules $k' \notin s_i$ as training progresses into later stages. Empirically, we do not observe systematic degradation. The monitored ratios remain stable or exhibit slight upward trends, indicating that the student's competence on earlier modules is preserved rather than overwritten. This suggests that the progression mechanism in IOA does not cause the student to forget previously mastered knowledge. A key reason for this stability is the structure of the curriculum itself. Since later modules are sequenced according to prerequisite dependencies, the student naturally reuses earlier knowledges when learning more advanced modules. This repeated utilization acts as an implicit rehearsal mechanism, reinforcing foundational knowledge without the need for an explicit replay buffer. Furthermore, the mastery gating ensures that modules are not advanced prematurely, reducing gradient instability that commonly contributes to forgetting. In summary, across all experiments we evaluated, the Mastery-Based Progressive Learning procedure does not induce catastrophic forgetting, and the student's performance on previously mastered modules remains well-preserved throughout training.

## T    LIMITATION ANALYSIS

While our proposed IOA framework demonstrates consistent gains across instruction-following and reasoning benchmarks, several limitations remain.

**Evaluation scope.** Our experiments focus on small- to mid-sized models (e.g., 3B parameters in Qwen and LLaMA families) and a limited benchmark suite. This restricts claims of scalability to larger architectures, long-context reasoning, or safety-critical domains. Future work should broaden both task coverage and model diversity.

**Dependence on seed data.** The identifier relies on a compact seed set ($\sim$3,000 items across four domains). Although de-duplication and filtering are applied, the diagnostic ability may still be biased toward these domains and linguistic styles, limiting generalization.

**Teacher reliance.** IOA assumes access to strong teacher models (e.g., OpenAI o1, DeepSeek-R1). Mis-calibration or systematic biases from these teachers can propagate into the deficiency diagnosis and curriculum design. Reproducibility is also challenged by reliance on proprietary, black-box systems.

**Heuristic sensitivity.** Key thresholds (e.g., for deficiency, dependency, mastery) are set empirically. While effective in our setting, they may need re-tuning for new domains or students, and could introduce brittleness when score distributions shift.

**Compute overhead.** The full pipeline—diagnosis, synthesis, and iterative training—remains computationally intensive, requiring multi-GPU clusters. This may hinder adoption by smaller labs, motivating work on lighter-weight teachers or more efficient curriculum construction.

## U  DECLARATION OF LLM USAGE

During the preparation of this manuscript, large language models were employed exclusively as auxiliary tools for language-related purposes. Specifically, LLMs were used to (1) refine the clarity and fluency of sentences; (2) polish grammar and style for improved readability; and (3) suggest alternative phrasings without altering the original technical content. No parts of the research design, data collection, algorithm development, experiment execution, or result analysis relied on LLMs. All conceptualization of the methodology, implementation details, and interpretation of results were conducted independently by the authors. The authors confirm that LLM assistance did not contribute to the generation of novel scientific ideas, data, or claims, and that the core intellectual contributions of this work remain solely those of the authors.

