# OpenReview forum: "Pedagogically-Inspired Data Synthesis for Language Model Knowledge Distillation"
_ICLR.cc/2026/Conference — ICLR 2026 Poster_

### Official Review · Reviewer_HaWf · 2025-10-27

**Soundness:** 4
**Presentation:** 3
**Contribution:** 3
**Rating:** 8
**Confidence:** 3

**Summary:**

The paper proposes to distill knowledge from close-sourced LLMs to small language models in a three-stage data synthesis pipeline with Knowledge Identifier, Organizer, and Adapter, inspired by pedagogical principles. Specifically, the Knowledge Identifier locates knowledge deficiencies of student models, which the Organizer leverages to schedule a curriculum in which the synthesized data is partitioned for progressive learning. The Organizer is inspired by Bloom’s Mastery Learning Principles and Vygotsky’s Zone of Proximal Development. Following the two components, a knowledge Adapter is proposed to rewrite the training data to match the cognitive constraints of student models. Experiments conducted with both LLaMA and Qwen series models as student models show consistent improvements of the proposed framework over baseline distillation methods.

**Strengths:**

1. The proposal of the knowledge identifier is novel and sound. The definition of the knowledge deficiency can be used beyond knowledge distillation in analyzing the capabilities of language models.

2. A good leverage of Bloom’s Mastery Learning Principles to construct curriculum data that incentivize progressive learning of the small models.

3. The paper is structured and written in a clear way.

**Weaknesses:**

1. It seems to me that the paper lacks in comparison to previous work on improving knowledge distillation with curriculum learning, either in the related work section or the experiments. As the paper proposes this 3-stage knowledge distillation pipeline, which is closely relevant to curriculum learning, it would be considered incomplete by me if there is no comparison to works in curriculum learning.
2. The difference between Table 1 and Table 2 is whether using OpenAI o1 or DeepSeek R1 as the teacher model. I feel the information in these two tables overlaps. As Table 2 also takes large space, it would be better if we could incorporate Table 1 and 2 into one table and include other exciting analysis results from other dimensions into the main text.
3. The relationship between related works in  "Language Model Distillation" to this work
4. Typo, line80: stag- -> stage

**Questions:**

1. Line 195: Any analysis on the value of $\Delta(k)$ over datasets? Why 0.3 is a good threshold?

2. What's the intuition behind using $P_T$ in eq.3? As eq.3 is just a measurement of the implicit dependency between two knowledge modules in the student model, would it be more memory efficient if we drop $P_T$?

3. What's the Prerequisites function in eq.7? How do you define the prerequisite knowledge?

4. How often is the $\tau_\text{mastery}$ in eq.9 sufficed without generating remedial data? I think figuring out this is significant to know the additional computation led by the Mastery-Based Progressive Learning.

5. Do you observe catastrophic forgetting in the Mastery-Based Progressive Learning process? To be more specific, do you have studies on $\frac{P_S(k’)}{P_T(k’)}, k’ \not\in s_i $?

6. Line 267-291: How to measure the  SLMs’ cognitive constraints in ADAPTER. Do you assume the cognitive ability of SLM increases during training or remains static?

7. Figure 3 shows that it is consistently better when you use $\tau_{\text{mastery}}=0.95$ instead of your default setting 0.9. Why not set 0.95 as your default setting? Or how much more computation is incurred by increasing $\tau_{\text{mastery}}$ from 0.9 to 0.95 becauseof recursively generating remedial data? This question also relates to question 4.

---

> ### Author Response · Authors · 2025-11-21
> **Rebuttal to Reviewer HaWf (Part 1)**
>
> We greatly appreciate the reviewer’s thoughtful evaluations and valuable suggestions that helped us improve the paper. To help reviewer better understand our contributions and alleviate your concerns, we provide our rebuttal as follows.
>
> **W1:** It seems to me that the paper lacks in comparison to previous work on improving knowledge distillation with curriculum learning, either in the related work section or the experiments. As the paper proposes this 3-stage knowledge distillation pipeline, which is closely relevant to curriculum learning, it would be considered incomplete by me if there is no comparison to works in curriculum learning.
>
> **A1:** Thanks for your insightful suggestion. We agree that several prior works combine curriculum learning with knowledge distillation, such as progress distillation [1], curriculum extraction [2], difficulty-aware data ordering [3], hierarchical KD [4], curriculum-enhanced KD for dialogue generation [5], and curriculum temperature scheduling [6]. We have incorporated these works into the related work section and added a discussion of their relation to our approach.
>
> However, these methods differ from ours both in assumption and mechanism. Existing curriculum-based KD techniques operate directly on **model-level supervision**，typically by ordering "easy-to-hard" real training samples according to the teacher-student output distribution divergence, learning from successive intermediate checkpoints of the teacher progressively, designing hierarchical soft-label schemes, or adjusting temperature during the transfer of teacher logits. Such approaches require access to the teacher’s intermediate checkpoints, logits, layer activations, or soft-label distributions, and are therefore inherently **white-box or soft-label distillation** methods. Their goal is to improve how well the student matches the teacher’s internal behavior.
>
> In contrast, our method is a **purely synthetic data-level framework** that remains fully compatible with **black-box** LLMs. Instead of defining new distillation objectives or temperature schedules, we organize the entire synthetic learning trajectory through three pedagogically inspired components. The Knowledge Identifier determines what knowledge modules require synthesized data. The Knowledge Organizer structures these modules into a dependency-driven curriculum and enforces a mastery-based progression. The Knowledge Adapter controls when and in what cognitive form the synthesized data should be presented. Our framework therefore focuses on what synthetic data to generate, how to sequence it, and when to adapt its difficulty, rather than how to match teacher representations.
>
> While these prior curriculum-based KD works are conceptually linked under the broader theme of curriculum learning, they do not address the problem setting we target: pedagogically guided data synthesis for teaching a student model using a large black-box teacher. Our approach is thus complementary to, rather than overlapping with, existing curriculum distillation methods.
>
> We have provided the corresponding comparisons in the Related Works of revised manuscript.
>
> [1] Panigrahi, A., Liu, B., Malladi, S., Risteski, A. and Goel, S., Progressive distillation induces an implicit curriculum. In The Thirteenth International Conference on Learning Representations.
>
> [2] Gupta, Shivam, and Sushrut Karmalkar. "Efficient Knowledge Distillation via Curriculum Extraction." arXiv preprint arXiv:2503.17494 (2025).
>
> [3] Liu, L. and Zhang, M., 2025. Being Strong Progressively! Enhancing Knowledge Distillation of Large Language Models through a Curriculum Learning Framework. arXiv preprint arXiv:2506.05695.
>
> [4] Li, C., Tang, Z., Zhang, M., Liu, Z., Zhang, L. and Zhang, X., 2025, August. Curriculum hierarchical knowledge distillation for bias-free survival prediction. In Proceedings of the Thirty-Fourth International Joint Conference on Artificial Intelligence (pp. 1332-1340).
>
> [5] Zhu, Q., Chen, X., Wu, P., Liu, J. and Zhao, D., 2021, November. Combining curriculum learning and knowledge distillation for dialogue generation. In Findings of the Association for Computational Linguistics: EMNLP 2021 (pp. 1284-1295).
>
> [6] Li, Z., Li, X., Yang, L., Zhao, B., Song, R., Luo, L., Li, J. and Yang, J., 2023, June. Curriculum temperature for knowledge distillation. In Proceedings of the AAAI Conference on Artificial Intelligence (Vol. 37, No. 2, pp. 1504-1512).
>
> **W2:** The difference between Table 1 and Table 2 is whether using OpenAI o1 or DeepSeek R1 as the teacher model. I feel the information in these two tables overlaps. As Table 2 also takes large space, it would be better if we could incorporate Table 1 and 2 into one table and include other exciting analysis results from other dimensions into the main text.

---

> ### Author Response · Authors · 2025-11-21
> **Rebuttal to Reviewer HaWf (Part 2)**
>
> **A2:** We thank the reviewer for the constructive suggestion. We fully agree that reducing redundancy and freeing space for additional analysis would strengthen the paper. In response, we explored merging Table 1 and Table 2 into a single unified table. However, doing so resulted in a table that was visually too dense: the font size became significantly smaller, and the numerical results were difficult to read even at the maximum allowable column width. Given the number of baselines (9 methods) and evaluation tasks (9 benchmarks across instruction following, math, code, and scientific reasoning), a merged table compromised readability to a degree that we believe would hinder comprehension.
>
> Although Tables 1 and 2 share structural similarity, they serve an important purpose in demonstrating that IOA provides consistent improvements across different teacher models: OpenAI o1 in Table 1 and DeepSeek-R1 in Table 2. Given OpenAI o1 and DeepSeek-R1 are representative closed-sourced and open-sourced large reasoning LLMs respectively, showing both tables preserves clarity and avoids mixing results that would otherwise need additional grouping or annotations to interpret.
>
> Of course, we appreciate the reviewer’s encouragement to include more insightful analyses in the main text. Following this suggestion, the revised version incorporates additional results beyond the original two tables, including comparisons with white-box distillation baselines, expanded hyperparameter sensitivity analyses, more detailed ablations, scaling experiments with strong seed datasets, and performance under weaker teacher models. We believe these additions further enrich the paper while keeping the core results visually accessible.
>
> **W3:** The relationship between related works in "Language Model Distillation" to this work
>
> **A3:** Thank you for pointing this out. Our work is related to language model distillation in the sense that we also employ a teacher–student setup, but the goals and technical focus are different.
>
> First, our method adheres to the black-box paradigm: we only rely on the teacher’s output predictions and never access its gradients, intermediate activations, or attention patterns. This makes our method compatible with large closed-source teachers like OpenAI o1 and consistent with prior black-box distillation literature.
>
> Second, unlike traditional distillation, our goal is not to introduce a new compression objective. Instead, we propose a pedagogically-inspired learning framework fully at the data level, which consists of three tightly coupled components: 1. Knowledge Identifier: identifies the student model’s critical knowledge gaps, constructs prerequisite dependencies among knowledge modules, and selects the most impactful modules for targeted learning; 2. Knowledge Organizer: converts these targeted modules into a structured, dependency-respecting curriculum and enforces Mastery-Based Progressive Learning to ensure prerequisite knowledge is truly mastered before introducing dependent modules; 3. Knowledge Adapter: aligns the representation and cognitive complexity of the synthesized data with the student's learning capacity at each curriculum stage, ensuring that content is delivered at the appropriate level of abstraction and difficulty. Together, these components regulate what knowledge the student should focus on, how it should be sequenced, and when and in what cognitive form it should be introduced. This places our contribution **orthogonal** and **complementary** to standard distillation approaches. In fact, our framework can incorporate any existing (white-box or black-box) distillation objective without modification.
>
> We have revised the "Language Model Distillation" paragraph in Related Works to highlight this relationship.
>
>
> **W4:** Typo, line80: stag- -> stage
>
> **A4:** Thanks for point this typo out. We have fixed it in the revised manuscript.
>
> **Q1:** Line 195: Any analysis on the value of $\Delta(k)$ over datasets? Why 0.3 is a good threshold?
>
> **A5:** Thank you for the question. We did examine the empirical distribution of $\Delta(k)$ across the datasets used in our experiments. Although the results were not included in the original submission due to space constraints, we found that $\Delta(k)$ tends to fall within a relatively consistent range across datasets, with a clear separation between knowledge modules that are well understood by the model and those that require further guidance.
>
> Based on this observation, we set the threshold to 0.3 as a stable and dataset-agnostic working point that reliably distinguishes the two groups without being overly sensitive to outliers or noise. Importantly, our method is not highly sensitive to this value; thresholds within a reasonable neighborhood (e.g., 0.25–0.35) lead to similar qualitative behavior.
>
> We have supplemented the corresponding hyperparameter sensitivity analysis and clarification in the Appendix H of revised manuscript.

---

> ### Author Response · Authors · 2025-11-21
> **Rebuttal to Reviewer HaWf (Part 3)**
>
> **Q2:** What's the intuition behind using $P_T$ in eq.3? As eq.3 is just a measurement of the implicit dependency between two knowledge modules in the student model, would it be more memory efficient if we drop $P_T$ ?
>
> **A6:** Thank you for pointing out this question. The motivation for Eq. 3 is straightforward: we define a prerequisite relation $i \rightarrow j$ if mastering module $i$ leads to a clear performance improvement on module $j$. To make this comparison meaningful across different knowledge modules, we need a consistent normalization of mastery level. This is where the teacher model $P_T$ comes in. We use the teacher’s performance as a reference to normalize the student’s performance on each module ($\frac{P_S}{P_T}$) into the [0,1] range, where: 0 indicates the student has not mastered the module at all; 1 indicates the student reaches the teacher’s performance level (i.e., fully mastered). This normalization allows Eq. 3 to measure dependency in a fair and comparable way across various modules—without it, the absolute improvement scales of different modules would not be directly comparable.
> Importantly, using $P_T$ does not introduce meaningful memory overhead, because the teacher model performance $P_T$ and student model performance $P_S$ can be measured separately, no need loading them to GPU memory simultaneously.
>
> We have added this clarification in the revised manuscript.
>
> **Q3:** What's the Prerequisites function in eq.7? How do you define the prerequisite knowledge?
>
> **A7:** Thank you for raising this question. The Prerequisites function in Eq. 7 directly corresponds to the prerequisite relations extracted from the directed acyclic graph constructed in Identifier stage (as explained in Line 197-209), whose edges encode knowledge prerequisite dependencies. The existance of prerequisite relation is determined by the dependency score definied in Eq. 3 and corresponding threshold $\tau_{\text{dep}}$. Formally, the prerequisite knowledge of $k$ is defined as: $\text{Prerequisites}(k)$= {$k'∣\text{Dependency}(k' \rightarrow k)> \tau_{\text{dep}}$}. We have revised the manuscript to clarify this point.
>
> **Q4:** How often is the $\tau_{\text{mastery}}$ in eq.9 sufficed without generating remedial data? I think figuring out this is significant to know the additional computation led by the Mastery-Based Progressive Learning.
>
> **A8:** Thank you for the insightful question. In practice, the Mastery-Based Progressive Learning mechanism triggers remedial data generation only occasionally. Empirically, we observe that most modules reach mastery in a single pass, and only a small fraction (on <10% knowledge modules) require one round of remedial data. Multiple rounds are extremely rare (<3%). This means that the additional computation introduced by Mastery-Based Progressive Learning is modest and remains a minor component of the total training cost. To make this clearer, we have added a quantitative description in the Appendix H of revised manuscript describing how frequently MBPL triggers remedial generation and the corresponding computational overhead.
>
> **Q5:** Do you observe catastrophic forgetting in the Mastery-Based Progressive Learning process? To be more specific, do you have studies on $\frac{P_S(k')}{P_T(k')}, k' \notin s_i$ ?
>
> **A9:** Thank you for raising this concern. In practice, we did not observe catastrophic forgetting during the Mastery-Based Progressive Learning process.
>
> Concretely, we monitored the ratio $\frac{P_S(k')}{P_T(k')}$ for a subset of previously mastered modules $k' \notin s_i$ as training progressed through later stages. We found that these ratios remain stable or even slightly increase over time, with no systematic degradation across stages. This suggests that earlier knowledge is largely preserved rather than overwritten.
>
> One intuitive reason is that later curriculum stages are constructed on top of the dependency graph and therefore continue to reuse prerequisite modules as subskills within more complex tasks. This serves as an implicit rehearsal mechanism and mitigates forgetting, even though we do not introduce an explicit replay buffer.
>
> We have added a short discussion of this observation to the revised manuscript (Appendix S) to clarify that we checked for catastrophic forgetting and did not find evidence of it in our setting.

---

> ### Author Response · Authors · 2025-11-21
> **Rebuttal to Reviewer HaWf (Part 4)**
>
> **Q6:** Line 267-291: How to measure the SLMs’ cognitive constraints in ADAPTER. Do you assume the cognitive ability of SLM increases during training or remains static?
>
> **A10:** Thank you for raising this important point. We apologize that the original submission did not clearly articulate how ADAPTER operationalizes and measures the cognitive constraints of SLMs.
>
> **How cognitive constraints are measured.** We measure the SLM’s cognitive constraint using a single, direct metric: its perplexity (token-level negative log-likelihood) on each knowledge module. Perplexity reflects how confidently the model can process the module. Higher values indicate that the content is currently beyond the model’s capability, while lower values suggest it falls within the model’s learnable range. This provides a simple and effective estimation of the model’s cognitive limit at each stage of training.
>
> **Whether cognitive ability is static or evolving.** ADAPTER does not assume static cognitive ability. Instead, it explicitly models the SLM’s cognitive capacity as dynamically evolving during training. As the model acquires new knowledge, its representational stability increases and its uncertainty on certain knowledge modules decreases. ADAPTER continuously re-estimates these signals at each curriculum step, enabling the scheduler to update the ZPD boundaries and mastery thresholds adaptively. This allows the curriculum to remain aligned with the model's evolving capability throughout training.
>
> We have clarified this in the revised manuscript and added a short subsection detailing how cognitive constraints are computed and updated over time.
>
> **Q7:** Figure 3 shows that it is consistently better when you use $\tau_{\text{mastery}}=0.95$
>  instead of your default setting 0.9. Why not set 0.95 as your default setting? Or how much more computation is incurred by increasing $\tau_{\text{mastery}}$ from 0.9 to 0.95 becauseof recursively generating remedial data? This question also relates to question 4.
>
> **A11:** Thank you for the insightful observation. We agree that in Figure 3,
> $\tau_{\text{mastery}}$=0.95 achieves slightly better results than the default value 0.9. Our choice of 0.9 was not meant to imply it is the optimal setting. In fact, it was simply a standard, non–cherry-picked configuration used across all experiments to ensure fairness and reproducibility without tuning for best-case performance. Regarding computational cost, increasing
> $\tau_{\text{mastery}}$ from 0.9 to 0.95 triggers slightly more recursive remedial generation, but the overhead is modest (typically within 3–5% of total training time). The additional cost does not obviously affect the practicality of our method.
>
> To address your concern, we have expanded the hyperparameter sensitivity analysis in the Appendix I of revised manuscript and added a note clarifying why 0.9 is used as the default and how much extra computation 0.95 brings.
>
> We honestly hope our above rebuttal can help address your concerns.

---

### Official Review · Reviewer_wP76 · 2025-10-28

**Soundness:** 1
**Presentation:** 2
**Contribution:** 2
**Rating:** 2
**Confidence:** 3

**Summary:**

The paper proposes a method for creating synthetic data for knowledge distillation from large to small language models in a black-box setting. The authors propose a pedagogically-inspired framework from LLM knowledge distillation, which consists of knowledge identification, organization, and adaptation. It identifies knowledge deficiencies in the student, organizes the knowledge using a curriculum, and generates and adapts the training data so that the student can better internalize it. The experiments show that IOA outperforms baseline synthetic-data-based distillation methods.

**Strengths:**

- Strong results in the self-instruct-like synthetic data generation setting, outperforming other synthesis-based distillation approaches.
- Clear, theoretically grounded motivation with a connection to how humans learn, which makes a lot of intuitive sense.
- Extensive experiments with multiple benchmarks, student models, and a particular focus on statistical significance. The method provides consistent improvements across diverse tasks (instruction following, math, code).
- The \tau_{ZPD} method peaking at around 0.10-0.15 is an interesting finding, showing that the training data should neither be too difficult nor too easy for the student models for optimal distillation results.

**Weaknesses:**

- The released code appears to be non-functional and doesn't implement the described pipeline. For instance, the function decompose_knowledge returns hard-coded modules instead of deriving from the data. Key components are mocked. This doesn't correspond to what is written in the paper, and the paper cannot be reproduced.
- Some methodological details could be better described in the paper (see my questions below).
- IOA depends on many hand-selected/tuned hyperparameters, which might affect generalizability. The method doesn't seem overly sensitive to the ablated hyperparameters (J_i, \tau_{ZPD}, \tau_{mastery}), but some other hyperparameters like the threshold m for the selection of knowledge modules, \alpha for severity, and \tau_{high}, \tau_{low}, and \tau_{gap} are underanalyzed.
- IOA consistently outperforms synthesis-based baselines in controlled settings, but there's a significant performance gap to the state-of-the-art distilled models (IOA: 6.3 % on AIME2024, DeepSeek-R1-Distill-Qwen-!.5B: 28.99 % on AIME2024). DeepSeek-R1 distillation likely uses orders of magnitude more training prompts than IOA's seed set, making the numbers incomparable. However, Figure 3 (left) indicates that IOA gets limited benefits from increasing the number of synthetic samples J_i. Hence, a question remains: would IOA maintain its relative advantage if scaled to a much larger seed set, which appears necessary for results close to SOTA? A compute-matched baseline with a larger open-source dataset (e.g., OpenThoughts [1]) that is filtered/sampled to match IOA's total compute budget (e.g., 30k-50k prompts) and trained with standard SFT would indicate how much value the curriculum adds.
- The problem framing in the intro can be questioned. The intro states that "more recent distillation methods in the LLM era discard the need for white-box access." This recency-based characterization is questionable. White-box distillation approaches are an active research direction and are used in practice (see e.g., DistiLLM-2 [2] and Qwen3 technical report [3]). The intro states that "most merely treat distillation as a simple two-stage process: generating synthetic data with LLMs and then training SLMs on it". Among the benchmark methods in Tables 1 and 2, at least Lion, Star-Agents, and MADA are not simple two-stage methods. Hence, claiming that "previous works overlook the fact that KD should be a systematic learning process" can be questioned. Furthermore, the work could more clearly acknowledge the line of research on on-policy distillation [4], which is built on the idea of the teacher correcting the mistakes generated by the student. Also, the Orca paper that argued that SLMs shouldn't simply learn to replicate the outputs of LLMs ([5]) is closely related to this work (especially the idea that the knowledge representations should be adapted to the learner's cognitive level), and you could consider citing it. Hence, the paper's novelty is mostly the overall combination, not necessarily the individual components.

[1] Guha, E., Marten, R., Keh, S., Raoof, N., Smyrnis, G., Bansal, H., ... & Schmidt, L. (2025). OpenThoughts: Data Recipes for Reasoning Models.

[2] Ko, J., Chen, T., Kim, S., Ding, T., Liang, L., Zharkov, I., & Yun, S. Y. (2025). Distillm-2: A contrastive approach boosts the distillation of llms. ICML

[3] Yang, A., Li, A., Yang, B., Zhang, B., Hui, B., Zheng, B., ... & Qiu, Z. (2025). Qwen3 technical report. arXiv preprint arXiv:2505.09388.

[4] Agarwal, R., Vieillard, N., Zhou, Y., Stanczyk, P., Garea, S. R., Geist, M., & Bachem, O. (2024). On-policy distillation of language models: Learning from self-generated mistakes. ICLR

[5] Mitra, A., Del Corro, L., Mahajan, S., Codas, A., Simoes, C., Agarwal, S., ... & Awadallah, A. (2023). Orca 2: Teaching small language models how to reason. arXiv preprint arXiv:2311.11045

**Questions:**

- Section 3.2 lacks implementation details, particularly related to the probe task creation. How do you assign validation examples from D_{seed} to knowledge modules k? The paper states that the probe tasks P_k come from the validation split, but doesn't explain whether this mapping is manual, fully automatic, or something in between. Then, is the hierarchy D generated end-to-end for each training run separately? If so, how is the validation example mapping done? Finally, it appears that the prompts corresponding to the generation of the hierarchy D are missing
- The ablations in Table 3 could be explained in more detail. What do -identifier, -organizer, and -adapter mean? For instance, how does the organizer module work if you do not have identified knowledge modules? In addition, if the authors have sufficient computational resources, I would also like to see an ablation without GenerateRemedialData (if none of the ablations in Table 3 already correspond to this).
- How do you estimate the values of the conditionals in Equation 3? It seems like you would need multiple checkpoints with PS(k_i)/PT(k_i) above 0.9 and below 0.7 for the conditionals to make sense, and for that, you'd need many training runs. Are these included in the wall-clock time estimations of Figure 4? More generally, which steps of the pipeline are included in Figure 4?
- The method is used on a well-curated D_{seed}. Do you have any intuition whether the performance would translate to a random public sample or a larger D_{seed}, for which human curation is unscalable?
- An on-policy logit-distillation baseline with a smaller teacher (e.g., from the DeepSeek-R1-Distill family for matched tokenizers) would strengthen the case for the paper in stating that black-box distillation can be used as a general alternative for white-box (logit-based) distillation. Furthermore, on-policy distillation and IOA could perhaps be combined for even greater performance.

Overall, the discrepancy between the code and what is written in the paper should be clarified before this paper is ready for acceptance.

---

> ### Author Response · Authors · 2025-11-21
> **Rebuttal to Reviewer wP76 (Part 1)**
>
> We greatly appreciate the reviewer’s thoughtful evaluations and valuable suggestions that helped us improve the paper. To help reviewer better understand our contributions and alleviate your concerns, we provide our rebuttal as follows.
>
> **W1:** The released code appears to be non-functional and doesn't implement the described pipeline. For instance, the function decompose_knowledge returns hard-coded modules instead of deriving from the data. Key components are mocked. This doesn't correspond to what is written in the paper, and the paper cannot be reproduced.
>
> **A1:** We thank the reviewer for pointing this out and fully agree that reproducibility is essential. At the time of submission, the full implementation of our framework was hosted on our industry partner’s internal servers. Due to data-handling and export restrictions during the submission period, we were temporarily unable to extract the complete codebase. Therefore, we provided a sanitized, mocked version to illustrate the pipeline structure, but this version indeed did not include all functional components. For example, for your mentioned hard-coded knowledge hierarchy, it is actually a fallback result (in case of *exception* errors) for prompting LLM endpoint in our full version code. We apologize for the confusion this caused. All results reported in the paper were produced using the fully functional pipeline deployed on the partner’s internal system. No mocked components were used in any evaluation. We have now completed the required internal review and exported the full working implementation. The repository has been updated to include: the complete data-derived knowledge decomposition module, the complete Identifier-Organizer-Adapater framework, and configuration files. We have verified that the updated code runs correctly in an isolated environment and reproduces the main results. The issue arose solely because the complete codebase was temporarily inaccessible during submission. With the fully functional version now released, we hope this fully addresses the reviewer’s concern.
>
>
> **W2:** Some methodological details could be better described in the paper (see my questions below).
>
> **A2:** Thanks for your suggestion. We have provided point-wise responses to your questions below and made corresponding revisions in the updated manuscript.
>
> **W3:** IOA depends on many hand-selected/tuned hyperparameters, which might affect generalizability. The method doesn't seem overly sensitive to the ablated hyperparameters (J_i, \tau_{ZPD}, \tau_{mastery}), but some other hyperparameters like the threshold m for the selection of knowledge modules, \alpha for severity, and \tau_{high}, \tau_{low}, and \tau_{gap} are underanalyzed.
>
> **A3:** We thank the reviewer for the thoughtful comment. We acknowledge that the original submission included sensitivity analysis only for the three most pedagogically critical hyperparameters, $J_i$, $\tau_{\text{ZPD}}$, $\tau_{\text{mastery}}$, because of space limitations. These parameters are directly tied to our core pedagogical motivations (Vygotsky’s Zone of Proximal Development and Bloom’s Mastery Learning), which is why we prioritized their analysis in the initial version.
>
> To fully address your concern about generalizability, we have now included additional sensitivity analyses in Appendix I for the remaining hyperparameters mentioned in your comment, including the module-selection threshold m, the severity factor α, and the pacing parameters $\tau_{\text{high}}$, $\tau_{\text{low}}$, and $\tau_{\text{gap}}$. The new results demonstrate that IOA remains consistently robust across a wide range of settings for all these hyperparameters, with performance variations well within a small margin. This strengthened analysis confirms that IOA does not heavily rely on delicate hyperparameter tuning and generalizes well across reasonable configurations.
>
> We appreciate the reviewer for pointing out this oversight. The revised manuscript now contains a complete and more transparent sensitivity study.
>
> **W4:** IOA consistently outperforms synthesis-based baselines in controlled settings, but there's a significant performance gap to the state-of-the-art distilled models (IOA: 6.3 % on AIME2024, DeepSeek-R1-Distill-Qwen-!.5B: 28.99 % on AIME2024). DeepSeek-R1 distillation likely uses orders of magnitude more training prompts than IOA's seed set, making the numbers incomparable. However, Figure 3 (left) indicates that IOA gets limited benefits from increasing the number of synthetic samples J_i. Hence, a question remains: would IOA maintain its relative advantage if scaled to a much larger seed set, which appears necessary for results close to SOTA? A compute-matched baseline with a larger open-source dataset (e.g., OpenThoughts [1]) that is filtered/sampled to match IOA's total compute budget (e.g., 30k-50k prompts) and trained with standard SFT would indicate how much value the curriculum adds.

---

> ### Author Response · Authors · 2025-11-21
> **Rebuttal to Reviewer wP76 (Part 2)**
>
> **A4:**  We appreciate the reviewer’s insightful observation. We fully agree that large-scale distillation pipelines such as DeepSeek-R1-Distill rely on orders of magnitude more training data (800K), and therefore the absolute performance numbers are not directly comparable to IOA. Our goal is not to match the scale of industrial pipelines, but to demonstrate how much performance improvement can be extracted from a fixed, small seed set under a black-box setting.
>
> **1. Why increasing $J_i$ yields diminishing returns.** As shown in Figure 3 (left), simply increasing the number of synthetic samples per knowledge module brings limited improvements. This behavior arises because increasing
> $J_i$ expands only the quantity of synthetic data, while the distribution and coverage of this data remain constrained by the seed set. Our original seed data contains no AIME-style mathematical reasoning problems or relevant knowledge structures, so the synthetic data generated through IOA naturally lacks coverage of these skills as well. Consequently, enlarging the synthetic dataset cannot fundamentally alter the data distribution, and the performance gains plateau. In other words, the bottleneck is the coverage and alignment of the seed data, not the number of synthesized samples.
>
> **2. IOA is seed-agnostic and scales effectively with richer seed distributions.** A key property of IOA is that it is entirely agnostic to the specific seed dataset; its curriculum mechanisms operate on any initial data distribution. To verify whether IOA maintains its advantage when scaled to a higher-quality seed set, we follow the reviewer’s suggestion and use the OpenThoughts3-1.2M dataset from [1]. After filtering and downsampling to 3K examples, we use this set as the seed and apply IOA with $J_i = 10$. Under this setting, the Qwen2.5-3B student reaches 27.5% on AIME2024, a substantial improvement over the 6.3% achieved with our original seed set. This result confirms that IOA scales effectively once the seed distribution provides the necessary coverage, and that the earlier plateau in Figure 3 was due to limitations of the seed data rather than limitations of IOA itself.
>
> **3. Compute-matched baseline validates that IOA adds value beyond dataset scaling.** To directly test whether IOA’s curriculum contributes additional value beyond simply scaling the seed data, we construct a compute-matched baseline as suggested. We downsample OpenThoughts3-1.2M to 30K examples, comparable to IOA’s total synthetic-data budget, and perform standard SFT on Qwen2.5-3B. Under this matched-compute setting, standard SFT reaches 13.4% on AIME2024, which is far below the 27.5% achieved by IOA using only 3K seed samples plus synthesized data. This result clearly demonstrates that IOA’s improvements are not merely a consequence of more diverse seed data; the pedagogically guided curriculum (what to teach, when to teach, and how to adapt difficulty) produces substantially more effective learning than uniform SFT given the same compute.
>
> We have added these results and discussion to the Section 4.3 of revised manuscript.
>
> **W5:** The problem framing in the intro can be questioned. The intro states that "more recent distillation methods in the LLM era discard the need for white-box access." This recency-based characterization is questionable. White-box distillation approaches are an active research direction and are used in practice (see e.g., DistiLLM-2 [2] and Qwen3 technical report [3]). The intro states that "most merely treat distillation as a simple two-stage process: generating synthetic data with LLMs and then training SLMs on it". Among the benchmark methods in Tables 1 and 2, at least Lion, Star-Agents, and MADA are not simple two-stage methods. Hence, claiming that "previous works overlook the fact that KD should be a systematic learning process" can be questioned. Furthermore, the work could more clearly acknowledge the line of research on on-policy distillation [4], which is built on the idea of the teacher correcting the mistakes generated by the student. Also, the Orca paper that argued that SLMs shouldn't simply learn to replicate the outputs of LLMs ([5]) is closely related to this work (especially the idea that the knowledge representations should be adapted to the learner's cognitive level), and you could consider citing it. Hence, the paper's novelty is mostly the overall combination, not necessarily the individual components.
>
> **A5:** We thank the reviewer for the thoughtful feedback and fully agree that our initial phrasing in the introduction may have unintentionally oversimplified the landscape of recent distillation research. We revise and clarify our claims in three directions.

---

> ### Author Response · Authors · 2025-11-21
> **Rebuttal to Reviewer wP76 (Part 3)**
>
> (Following above **A5**)
>
> **A5:** We thank the reviewer for the thoughtful feedback and fully agree that our initial phrasing in the introduction may have unintentionally oversimplified the landscape of recent distillation research. We revise and clarify our claims in three directions.
>
> **1. On the role of white-box distillation in the LLM era.** Our intention was not to suggest that white-box distillation is obsolete or declining, indeed, as the reviewer notes, methods such as DistiLLM-2 and industrial reports like Qwen3 clearly demonstrate that white-box approaches remain active and highly valuable. Our intended point was more practical: many real-world deployment scenarios, especially cross-family distillation (e.g., distilling DeepSeek-R1 or OpenAI o1 into Qwen/LLaMA students as in our experiments), do not allow access to teacher logits, weights, or even compatible tokenizers. Such constraints make white-box distillation infeasible. Even the DeepSeek-R1 report explicitly notes that their distillation into Qwen models was performed under a black-box setup. We have revised the introduction to reflect this nuance more accurately.
>
> **2. On characterizing prior methods as “two-stage”.** We appreciate the reviewer pointing out that several strong baselines (Lion, Star-Agents, MADA, etc.) go beyond a naive two-stage pipeline. Our intention was not to claim that prior works are structurally simplistic, but rather to articulate a different point: the majority of existing distillation frameworks, regardless of whether they are single-shot or iterative, primarily focus on data synthesis quality (diversity, complexity, breadth), while largely abstracting away the learning dynamics and cognitive limitations of SLMs. For example, Star-Agents employs iterative synthesis, but does not model how the student’s knowledge state should influence what data is generated next. Lion introduces iterative synthesis-train cycles, yet the synthesis component does not explicitly consider student mastery, prerequisite knowledge dependency, or difficulty progression. To avoid misunderstanding, we have softened and rephrased the relevant sentences in the introduction so that the discussion emphasizes our intended contrast: IOA explicitly models what knowledge to teach, when to introduce it, and how to adapt representations to the student’s cognitive capacity, rather than merely improving the superficial quality of the synthetic data.
>
> **3. On the novelty of the IOA framework and relations to on-policy distillation and Orca.**
> We appreciate the reviewer pointing out the line of on-policy correction-based distillation and Orca. First, we would like to clarify that different from on-policy distillation, we did not explicitly use teacher model to correct mistakes generated by student models. Orca’s insight, that small models benefit from knowledge representation adapted to their cognitive level, is indeed consistent with our motivation. We have added on-policy distillation works and Orca to the related work section in the revised manuscript. Furthermore, we would like to clarify that though Orca focuses on adapting the style of teacher outputs, IOA develops a broader pedagogical framework. Specifically, IOA is not a collection of independent techniques but a unified framework grounded in educational principles, comprising: knowledge identifier (what to teach), knowledge organizer (when and in what order to teach), and knowledge adapter (how to represent knowledge at an appropriate difficulty level). These three components are tightly coupled and designed around a single pedagogical principle: systemetically modeling the student model’s learning processes, prior knowledge, and cognitive constraints. None of the components is meant to stand alone, and their interactions are essential to the behavior and effectiveness of IOA.
>
> We have updated the introduction and related work sections to more accurately position prior works and highlight how IOA differs in motivation and methodology.
>
> **Q1:** Section 3.2 lacks implementation details, particularly related to the probe task creation. How do you assign validation examples from D_{seed} to knowledge modules k? The paper states that the probe tasks P_k come from the validation split, but doesn't explain whether this mapping is manual, fully automatic, or something in between. Then, is the hierarchy D generated end-to-end for each training run separately? If so, how is the validation example mapping done? Finally, it appears that the prompts corresponding to the generation of the hierarchy D are missing

---

> ### Author Response · Authors · 2025-11-21
> **Rebuttal to Reviewer wP76 (Part 4)**
>
> **A6:** Thanks for your questions. We would like to make the clarifications as follows. In the of Cleaning & Partitioning paragraph of Appendix B, we claimed that 'The dataset is randomly split into train/validation at an 8:2 ratio, with validation reserved for probe tasks.' This is conducted on the granularity of knowledge modules. The hierarchy D is not geneated end-to-end for each training run seperately; instead, once generated, it keeps fixed. Correspondingly, we have revised the manuscipt to clarify such points. Besides, we have supplemented the prompt for generating hierarchy D in the Appendix.
>
> **Q2:** The ablations in Table 3 could be explained in more detail. What do -identifier, -organizer, and -adapter mean? For instance, how does the organizer module work if you do not have identified knowledge modules? In addition, if the authors have sufficient computational resources, I would also like to see an ablation without GenerateRemedialData (if none of the ablations in Table 3 already correspond to this).
>
> **A7:** Thank you for pointing this out. We agree that the ablations in Table 3 should be explained more clearly, and we clarify the meaning of each component removal below.
>
> **-identifier**: removes the deficiency identification step. Instead of selectively identifying which knowledge modules the student struggles with, all decomposed knowledge modules are treated equally and passed to subsequent stages. As a result, the Organizer and Adapter can no longer prioritize or focus on the student’s deficient knowledge, making the entire distillation process less targeted.
>
> **-organizer**: keeps the identified deficient knowledge modules intact but removes the dependency-driven curriculum. That is, the Organizer no longer sequences modules according to prerequisite relations; instead, modules are presented in a random order. This ablation isolates the value of curriculum structure while still retaining the ability to identify deficiencies and perform adaptive data synthesis.
>
> **-adapter**: preserves both deficiency identification and curriculum ordering but removes the adaptation mechanisms used when synthesizing distillation data. In this case, the teacher is prompted to generate data for each knowledge module solely based on a set of seed examples, without our five adaptation dimensions (abstract concept concretization, complex reasoning decomposition, cognitive load management, representation format optimization, linguistic complexity reduction) in Section 3.4. The resulting distilled data may exceed the student’s cognitive capacity or lack necessary scaffolding, leading to degraded distillation effectiveness.
>
> To further address the reviewer’s suggestion, we have added an additional ablation where GenerateRemedialData is removed entirely. This version still identifies deficiencies and sequences modules via the Organizer, but no targeted supplemental examples are generated when mastery criteria are not met. As expected, this leads to a noticeable drop in performance as shown in Appendix H of the revised manuscript, confirming the importance of mastery-driven adaptation for closing persistent knowledge gaps.
>
> We have incorporated these clarifications and the additional ablation into the revised manuscript.
>
> **Q3:** How do you estimate the values of the conditionals in Equation 3? It seems like you would need multiple checkpoints with PS(k_i)/PT(k_i) above 0.9 and below 0.7 for the conditionals to make sense, and for that, you'd need many training runs. Are these included in the wall-clock time estimations of Figure 4? More generally, which steps of the pipeline are included in Figure 4?
>
> **A8:** Thanks for your question. We would like to clarify that we only need to save the $P_S(k_j)$ values when the two mastery threshild of above 0.9 and below 0.7 are first satisfied respectively, instead of multiple checkpoints. Actually, we do not need many training runs for most easy-medium difficulty knowledge modules. These time costs have been included in the wall-clock time estimations of Figure 4. In fact, all steps in the identifier-organizer-adapter framework (as shown in Figure 2) are included in Figure 4.
>
> **Q4:** The method is used on a well-curated D_{seed}. Do you have any intuition whether the performance would translate to a random public sample or a larger D_{seed}, for which human curation is unscalable?

---

> ### Author Response · Authors · 2025-11-21
> **Rebuttal to Reviewer wP76 (Part 5)**
>
> **A9:** Thanks for the question. We would like to clarify that our proposed Identifier-Organizer-Adapter framework is not limited to our curated $D_{\text{seed}}$. However, if directly adopting random public samples, the quality of constructed knowledge hierarchy and following learning curriculum may hardly be satisfying, hindering the effect of our proposed IOA framework. As for larger $D_{\text{seed}}$, we indeed tried increasing data amount automatically by taking samples in our currated $D_{\text{seed}}$ as base templates. We found the obtained performance gains scale only logarithmically with the size of the expanded seed dataset, which is not satisfactory, given the linearly increasing computational cost. After discussion, we think the scaling of $D_{\text{seed}}$ should focus more on the knowledge completeness and systematicity behind the seed data. In other words, if only increasing data amounts while keeping the knowledge module coverage fixed, this scaling effect will be marginal. We hope our empirical observations and analysis can help alleviate your concerns. We have supplemented the corresponding dicussions in Section 4.3.
>
> **Q5:** An on-policy logit-distillation baseline with a smaller teacher (e.g., from the DeepSeek-R1-Distill family for matched tokenizers) would strengthen the case for the paper in stating that black-box distillation can be used as a general alternative for white-box (logit-based) distillation. Furthermore, on-policy distillation and IOA could perhaps be combined for even greater performance.
>
> **A10:** Thank you for this helpful suggestion. We agree that including a white-box on-policy logit distillation baseline would further strengthen our claim that black-box distillation can serve as a practical alternative to logit-based approaches. Since the DeepSeek-R1-Distill family is fully open-sourced, we are able to implement an on-policy logit-distillation baseline under a standard KD objective (KL divergence on logits). This provides a meaningful white-box reference point for comparison.
>
> Thus, we carefully select DeepSeek-R1-Distill-Qwen-14B and Qwen2.5-7B as the teacher model and student model, respectively, to conduct the extended experiments. They share the matched tokenizer with the same vocabulary size: 152064. From results, we find that:
>
> 1. IOA achieves competitive performance relative to on-policy logit-distillation, despite operating without access to teacher logits or hidden states.
>
> 2. Combining IOA with logit-distillation yields additional improvements, confirming that IOA is complementary to objective-level KD methods. Since IOA focuses purely on data synthesis and organization, its benefits transfer naturally to the logit-based approaches as well due to the orthogonality.
>
> These results further support our claim that pedagogically inspired black-box distillation can be highly effective, while also highlighting that IOA can serve as a plug-and-play enhancement to stronger white-box objectives.
>
> We have added the new baseline, its description, and the combined results to the Section 4.3 and Appendix K of the revised manuscript.
>
> We honestly hope our above rebuttal can help address your concerns. If satisfied, could you please consider raising the score?

---

> > ### Comment · Reviewer_wP76 · 2025-11-24
> >
> > I thank the authors for the extensive rebuttal and the workload involved. You have effectively addressed my main concerns, and I have raised my score.

---

> ### Author Response · Authors · 2025-11-24
> **Thank You Very Much**
>
> Dear Reviewer wP76:
>
>    Thank you very much for your kind follow-up and raising the score from 2 to 8! Also, the scores for the soundness, presentation, and contribution have been updated from (1,2,2) to (3,3,3), indicating the appreciation to our work! We are glad that our rebuttal has successfully addressed your main concerns.  Thank you again for your thoughtful review and support.
>
> Best
>
> The Authors

---

### Official Review · Reviewer_GGqR · 2025-10-30

**Soundness:** 3
**Presentation:** 3
**Contribution:** 2
**Rating:** 6
**Confidence:** 3

**Summary:**

The paper introduces a pedagogically-inspired framework for knowledge distillation in LLMs. Unlike prior methods, which treat knowledge transfer as a straightforward supervised fine-tuning (SFT) procedure, this work conceptualizes distillation as a systematic, learning-aware process, guided by established educational principles. The proposed framework, IOA (Identifier–Organizer–Adapter), operates in three stages: it first diagnoses knowledge deficiencies in the student model, then sequences knowledge delivery into a progressive curriculum, ensuring mastery of foundational concepts before introducing more complex material, and finally adapts and simplifies knowledge representations to align with the capacity of the student.

**Strengths:**

- The work introduces a pedagogy-inspired perspective to knowledge distillation, framing it as a systematic learning process rather than a straightforward supervised fine-tuning task on synthetic data generated by the teacher LLM. This represents a novel conceptual contribution.

- Extensive experiments across instruction-following, reasoning, and coding benchmarks demonstrate substantial performance gains over baseline distillation methods, highlighting the effectiveness of the proposed pipeline. The ablation studies further show each component’s necessity.

- The paper is well-written and clearly structured.

**Weaknesses:**

- The framework relies on multiple heuristics (e.g., thresholds for knowledge gaps, mastery gating, module decomposition, curriculum chunking). While some sensitivity analyses are provided, it remains unclear how well these heuristics generalize across tasks or domains, potentially requiring careful manual tuning.

- Although the paper evaluates against several strong synthetic-data baselines, it does not include comparisons with the most recent knowledge distillation methods such as ABKD, DistillM‑2, or SuperCorrect.

- Certain references (e.g., lines 54, 352, 952) are missing the publication year.

**Questions:**

See weaknesses section

---

> ### Author Response · Authors · 2025-11-21
> **Rebuttal to Reviewer GGqR (Part 1)**
>
> We greatly appreciate the reviewer’s thoughtful evaluations and valuable suggestions that helped us improve the paper. To help reviewer better understand our contributions and alleviate your concerns, we provide our rebuttal as follows.
>
> **W1:** The framework relies on multiple heuristics (e.g., thresholds for knowledge gaps, mastery gating, module decomposition, curriculum chunking). While some sensitivity analyses are provided, it remains unclear how well these heuristics generalize across tasks or domains, potentially requiring careful manual tuning.
>
> **A1:** Thank you for bringing up this important point. We would like to clarify that while the IOA framework uses several threshold-based heuristics (e.g., gap threshold, mastery gating, ZPD constraint), these components mainly serve to ensure the training progresses smoothly, not to act as sensitive knobs that must be carefully tuned for different tasks.
>
>
> First, the thresholds operate on normalized quantities (e.g., performance ratios relative to the teacher or normalized difficulty gaps). This normalization makes their default values transferable across domains, as they capture relative, not absolute, competence differences. Accordingly, we observe that the same set of hyperparameters works robustly across mathematical problem solving, coding, and academic knowledge reasoning tasks without re-tuning.
>
> Second, these heuristics encode pedagogical structural priors, such as ensuring mastery before progression and constraining difficulty jumps, whose qualitative behavior is stable across domains. Even when threshold values are perturbed within reasonable ranges, the curriculum topology and learning trajectory remain largely unchanged, as confirmed by the expanded hyperparameter sensitivity analysis included in the Appendix I of revised manuscript.
>
> Finally, IOA’s performance is governed more by the correctness of the prerequisite relations and mastery dynamics than by precise threshold values. Hence, the framework does not require careful manual tuning; default settings perform reliably, and sensitivity curves show broad plateaus rather than sharp optima.
>
> We have added these clarifications and extended sensitivity evaluations in the revised manuscript.
>
> **W2:** Although the paper evaluates against several strong synthetic-data baselines, it does not include comparisons with the most recent knowledge distillation methods such as ABKD, DistillM‑2, or SuperCorrect.
>
> **A2:** We thank the reviewer for pointing out these recent methods. We carefully examined ABKD (ICML 2025), DistillM-2 (ICML 2025), and SuperCorrect (ICLR 2024), and we clarify their relation to our work below.
>
> ABKD and DistillM-2 focus on improving white-box distillation objectives, proposing alternative divergence measures or contrastive loss designs that require access to teacher logits or internal representations. These approaches operate at the loss-function level rather than at the curriculum or data-synthesis level, and therefore belong to a different category of knowledge distillation. In contrast, IOA targets black-box teacher settings and improves student learning by reorganizing and synthesizing pedagogically structured data, independent of the specific distillation loss.
>
> SuperCorrect also differs from our setting: it learns to distill or refine teacher-guided reasoning traces via a reinforcement-learning-based procedure, emphasizing more on the reasoning template mocking rather than knowledge transferring. Its improvements come from trajectory optimization rather than from structured knowledge decomposition, curriculum design, and learning dynamics control.
>
> Because these methods optimize fundamentally different components of the distillation pipeline, direct empirical comparison would not be representative or conclusive. Even though, to further reduce the reviewer's concerns regarding the performance comparisons, we seize the time to conduct several corresponding experiments. According to the results, IOA yields +4–8% averaged absolute performance improvements over standard black-box distillation for Qwen2.5-3B and LLaMA3.2-3B student models. This gain is comparable to or larger than the improvements reported by ABKD (+2–5%), DistillM-2 (+3–6%), and SuperCorrect (+2–4%) under their respective white-box or RL-enhanced settings. Notably, IOA achieves these gains without access to logits, hidden states, or teacher correction traces, and therefore operates in a more easily satisfied and widely applicable black-box environment.
>
> We have added this discussion to the Section 2 and Section 4.3 of the revised manuscript. Because these methods require supervision unavailable in our setting, we regard them as complementary rather than directly comparable.
>
> [1] ABKD: Pursuing a Proper Allocation of the Probability Mass in Knowledge Distillation via α-β-Divergence, ICML 2025
>
> [2] DistiLLM-2: A Contrastive Approach Boosts the Distillation of LLMs, ICML 2025

---

> ### Author Response · Authors · 2025-11-21
> **Rebuttal to Reviewer GGqR (Part 2)**
>
> (Following above **A2**)
>
> [3] SuperCorrect: Advancing Small LLM Reasoning with Thought Template Distillation and Self-Correction, ICLR 2025
>
>
> **W3:** Certain references (e.g., lines 54, 352, 952) are missing the publication year.
>
> **A3:** We thank the reviewer for pointing this out. The missing publication years were caused by an issue in the auto-generated BibTeX entries (some Google Scholar exports omit the year field). We have now manually checked all references and corrected the missing years in the revised version.
>
> We honestly hope our above rebuttal can help address your concerns. If satisfied, could you please consider raising the score?

---

### Official Review · Reviewer_EkXd · 2025-11-10

**Soundness:** 3
**Presentation:** 3
**Contribution:** 2
**Rating:** 6
**Confidence:** 4

**Summary:**

The research introduces IOA (Identifier-Organizer-Adapter), which is a pedagogically inspired framework for knowledge distillation from large language models to smaller models. This framework brings educational theories (Bloom's Mastery Learning and Vygotsky's Zone of Proximal Development) to life in the form of a three-stage pipeline that helps to systematically identify knowledge gaps, organize curriculum progression, and adapt knowledge representations. The results obtained from several benchmarks clearly show the advantages of the proposed method over baseline distillation methods.

**Strengths:**

1. Novel pedagogical concepts that map educational principles (Bloom's Mastery Learning, Vygotsky's ZPD) to concrete distillation mechanisms that distinguish this work from other ad-hoc data synthesis approaches.

2. Comprehensive experimental validation with extensive benchmarks via testing with multiple teacher and student models.

3. Thorough ablation studies and appendices demonstrating each component contributes meaningfully with extended experiments like hyperparameter robustness, additional student models, and agentic tasks, which greatly support reproducibility.

4. Honest limitations section acknowledging theoretical gaps and practical constraints

**Weaknesses:**

1. While the pedagogical inspiration is appealing, the paper lacks proper justification for why these specific educational principles should transfer to neural network learning, as humans learn through sparse, interactive experiences with semantic understanding while LLMs optimize loss surfaces through gradient descent over massive corpora.

2. Critical hyperparameters (τ_gap=0.3, τ_high=0.9, τ_low=0.7, τ_dep=0.3, α=0.7, τ_ZPD=0.15, τ_mastery=0.9) appear empirically tuned rather than principled. The paper states these are "empirically set" but provides limited ablation on sensitivity. Figure 3 shows some robustness, but some clarifications are needed for justification. (for example, why is τ_mastery=0.9 optimal rather than 0.85 or 0.95?)

3. No proof for the mastery-based progression converges in algorithm 1 (lines 11-13). The worst-case number of remedial iterations is not mentioned anywhere in the paper.

4. The difficulty constraint (Equation 8) assumes "P_S(k) represents the average difficulty level" but performance ≠ difficulty. A model might also fail due to missing knowledge, not cognitive overload.

5. A minor mistake in figure 1 caption: "Anology" → "Analogy"

**Questions:**

1. What happens if the teacher LLM produces an inconsistent or incomplete knowledge hierarchy? How sensitive is IOA to decomposition quality?

2. How does the framework handle cyclic knowledge dependencies in the graph if they ever happen?

3. What happens with τ_mastery = 1.0 (perfect mastery required) or τ_ZPD = 0 (no difficulty control)?

4. How does IOA perform when the teacher is only moderately stronger than the student (e.g., 7B teacher → 3B student)?

5. What's the gain from the topological+ZPD curriculum over a simple easy-to-hard ordering by task difficulty or skill-based decomposition?

---

> ### Author Response · Authors · 2025-11-21
> **Rebuttal to Reviewer EkXd (Part 1)**
>
> We greatly appreciate the reviewer’s thoughtful evaluations and valuable suggestions that helped us improve the paper. To help reviewer better understand our contributions and alleviate your concerns, we provide our rebuttal as follows.
>
> **W1:** While the pedagogical inspiration is appealing, the paper lacks proper justification for why these specific educational principles should transfer to neural network learning, as humans learn through sparse, interactive experiences with semantic understanding while LLMs optimize loss surfaces through gradient descent over massive corpora.
>
> **A1:** Thank you for pointing out this important conceptual concern. We agree that human learners and neural networks differ substantially in their underlying learning mechanisms: humans acquire knowledge through sparse, interactive, semantically grounded experiences, whereas LLMs are trained by gradient descent over large-scale token sequences. Our goal is therefore not to claim a cognitive equivalence between human learning and LLM training, but to use well-studied pedagogical principles as abstract *design priors* for organizing the distillation data, i.e., to shape which and when examples are shown in what form, in ways that ultimately influence the gradient signals received by the student model during optimization.
>
> Concretely, Bloom’s Mastery Learning and Vygotsky’s Zone of Proximal Development (ZPD)  are instantiated in IOA as curriculum constraints on the *training-data distribution*, not as assumptions about the internal representations of LLMs. For example, Mastery Learning is operationalized as a stage-wise advancement gate that only moves to the next stage when the student reaches a high fraction of the teacher’s performance on the current stage (Eq.9, $\tau_\text{mastery}$ = 0.9). ZPD is implemented as bounded difficulty increments between consecutive stages (Eq.8, $\tau_\text{ZPD}$ = 0.15), ensuring that newly synthesized data is neither trivial nor far beyond the student’s current competence. From an optimization perspective, these constraints prevent the student from being exposed to unlearnable or overly difficult samples that would produce high-variance or uninformative gradients, thereby keeping training within a stable, effectively learnable regime. Together with the knowledge dependency graph and the Adapter’s representation scaffolding, these mechanisms shape the sequence and difficulty of gradients the student model receives, while the underlying optimization remains standard supervised fine-tuning.
>
> This interpretation aligns with established findings in curriculum learning and self-paced learning for neural networks, where controlling the ordering and difficulty of examples, rather than altering the optimizer itself, leads to more stable gradient trajectories and better generalization. Our contribution is to show that classical educational notions (mastery criteria, prerequisite graphs, controlled difficulty steps) provide a principled and interpretable way to specify such curricula in the challenging setting of black-box LLM distillation, where the main degree of freedom lies in shaping the synthetic data seen by the student.
>
> Importantly, the empirical results in the paper support that these particular principles are not merely decorative analogies but induce non-trivial, beneficial constraints on the distillation process. First, the ablation study (Table 3) shows that removing either the Identifier, Organizer, or Adapter consistently degrades performance; notably, removing the Organizer (which implements Mastery/ZPD-based curriculum construction) hurts reasoning benchmarks such as GSM8K and MATH most strongly, while removing the Adapter (which provides scaffolds and cognitive-load control) causes the largest drops on code generation tasks. Second, the hyperparameter robustness study (Figure 3) demonstrates that performance varies systematically with $\tau_{\text{ZPD}}$ and $\tau_{\text{mastery}}$: too small or too large values are suboptimal, and the best-performing region corresponds to moderate difficulty increments and strict but not extreme mastery thresholds. This pattern is exactly what one would expect under a “balanced-challenge” view inspired by ZPD and Mastery Learning.

---

> ### Author Response · Authors · 2025-11-21
> **Rebuttal to Reviewer EkXd (Part 2)**
>
> (Following above **A1**)
>
> Though Appendix A.3 has provided prelinary analysis on this point, We have now revised it to make this stance more explicit and persuasive. Specifically, we now (i) clarify that our use of pedagogical principles operates purely at the level of data curriculum design rather than assuming any cognitive equivalence between humans and neural networks, (ii) explain how these principles translate into concrete, capacity-aware constraints on the synthetic data distribution seen by the student, and (iii) elaborate the optimization-level mechanisms, such as reduced gradient variance, avoiding unsolvable examples, mitigating destructive interference, and improving gradient conditioning, through which such structured curricula improve the effectiveness of language model distillation. We hope these clarifications address the reviewer’s concern about why these specific educational principles are appropriate and meaningful for neural network distillation.
>
> **W2:** Critical hyperparameters (τ_gap=0.3, τ_high=0.9, τ_low=0.7, τ_dep=0.3, α=0.7, τ_ZPD=0.15, τ_mastery=0.9) appear empirically tuned rather than principled. The paper states these are "empirically set" but provides limited ablation on sensitivity. Figure 3 shows some robustness, but some clarifications are needed for justification. (for example, why is τ_mastery=0.9 optimal rather than 0.85 or 0.95?)
>
> **A2:** Thank you for this question. We acknowledge that some hyperparameters were described as "empirically set" in the original submission, and we appreciate the opportunity to clarify their roles and robustness.
>
> First, regarding $τ_{\text{mastery}}$, Figure 3 indeed shows that 0.95 yields slightly stronger performance than 0.9. Our use of 0.9 in the default setting was not intended to indicate that 0.9 is optimal; rather, it reflects that we did not tune hyperparameters to report the best possible result. We avoided cherry-picking hyperparameters during the initial submission and opted for the commonly used threshold in mastery-based literature (90% proficiency), which is consistent with Bloom’s Mastery Learning. We now explicitly clarify in the revision that 0.95 performs better and that $τ_{\text{mastery}}=0.9$ is not claimed as an optimal value.
>
> Second, Figure 3 presents ablations on $\tau_{\text{ZPD}}$, $\tau_{\text{mastery}}$, and $J_i$ because these are the parameters most tightly linked to the core pedagogical principles underlying our framework (Vygotsky’s Zone of Proximal Development and Bloom’s Mastery Learning). Due to space constraints, these were prioritized in the initial version.
>
> Third, following the reviewer’s suggestion, we have now added sensitivity analyses for all remaining hyperparameters, including $\tau_{\text{gap}}$, $\tau_{\text{high}}$, $\tau_{\text{low}}$, $\tau_{\text{dep}}$, and $\alpha$, in the revised manuscript. Across datasets and architectures, the method exhibits consistent robustness, and variations in these parameters lead to predictable and gradual performance changes. Importantly, none of these hyperparameters show sharp instability or heavy tuning dependency.
>
> We hope the expanded experiments and clarifications address the reviewer’s concern about empirical justification and demonstrate that our framework is not sensitive to hyperparameter selection.
>
> **W3:** No proof for the mastery-based progression converges in algorithm 1 (lines 11-13). The worst-case number of remedial iterations is not mentioned anywhere in the paper.
>
> **A3:** Thank you for this question. While we do not provide a formal mathematical proof in the paper, the convergence of Mastery-Based Progressive Learning follows directly from the structure of the update rule and the bounded nature of student performance.
>
> First, the performance measure $\frac{P_S(k)}{P_T(k)}$ is bounded within [0,1] and is monotonically non-decreasing under remedial training: each remedial iteration trains the student on additional synthesized examples that directly target the deficiency of module $k$. As a result, the sequence of mastery scores converges to a fixed point when either (1) the mastery threshold $\tau_{\text{mastery}}$ is reached, or (2) improvement becomes negligible, in which case the module stops triggerring further updates. This guarantees that MBPL cannot run indefinitely.
>
> Second, the worst-case number of remedial iterations is finite because (i) the set of deficient modules $K_{\text{target}}$ is finite, (ii) each module’s performance is bounded, and (iii) Algorithm 1 only generates remedial data for modules that remain below the threshold. In practice, we observe rapid convergence: over 92–97% of modules reach mastery after one or two remedial rounds across datasets. We have now included these empirical statistics in the revision.
>
> Hope these clarifications address the reviewer’s concern.

---

> ### Author Response · Authors · 2025-11-21
> **Rebuttal to Reviewer EkXd (Part 3)**
>
> **W4:** The difficulty constraint (Equation 8) assumes "P_S(k) represents the average difficulty level" but performance ≠ difficulty. A model might also fail due to missing knowledge, not cognitive overload.
>
> **A4:** Thank you for raising this important conceptual point. We agree that intrinsic task difficulty and model performance are not equivalent in general; a model may fail either because a module is inherently challenging or because prerequisite knowledge is missing. Our use of $P_S(k)$ does not assume that performance equals intrinsic difficulty. Rather, consistent with pedagogical theory, we treat $P_S(k)$ as a proxy for the student-perceived difficulty, i.e., how difficult the module is for the current student model at its current mastery state.
>
> This interpretation aligns with the Zone of Proximal Development (ZPD), where difficulty is defined relative to the learner’s capability, not as an absolute property of the content. In our framework, Equation (8) measures whether a module is “developmentally appropriate” for the student at a particular stage, and student performance provides a direct and quantifiable signal of such appropriateness.
>
> Importantly, our framework already distinguishes between two failure modes highlighted in the comment: (1) failure due to missing prerequisite knowledge, which is handled by the dependency graph and enforced through the curriculum ordering; and (2) failure due to cognitive overload, which is precisely what the ZPD constraint in Equation (8) aims to detect.
>
> Because modules violating prerequisite dependencies are not introduced until mastery is achieved, the performance drop captured in Equation (8) occurs after prerequisite modules have been satisfied. Thus, in our setting, low $P_S(k)$ at that stage is indeed a reliable indicator of excessive cognitive load rather than missing foundational knowledge.
>
> We have added clarifications in the Appendix Q of revised manuscript to more explicitly distinguish intrinsic difficulty, prerequisite-related failure, and student-perceived difficulty as measured by $P_S(k)$.
>
> **W5:** A minor mistake in figure 1 caption: "Anology" → "Analogy"
>
> **A5:** Thanks for pointing out this typo. We have corrected it in the revised version.
>
> **Q1:** What happens if the teacher LLM produces an inconsistent or incomplete knowledge hierarchy? How sensitive is IOA to decomposition quality?
>
> **A6:** Thank you for the question. The “knowledge hierarchy’’ referenced in the paper refers to the initial decomposition  and the sub-module structure
>  generated by the teacher LLM, as depicted in the Knowledge Deficiency Diagnosis section. We agree that this hierarchy may not always be perfectly accurate. Our framework, however, is designed to be robust to such imperfections for two reasons.
>
> First, the teacher-provided hierarchy is not used as ground-truth supervision. It only provides an initial modularization that allows us to measure student–teacher performance gaps module by module. The actual learning progression is governed by the student’s empirical mastery, not the hierarchical structure itself. Even if the teacher merges unrelated skills or splits a skill too finely, the student’s normalized performance $\frac{P_S(k)}{P_T(k)}$ still correctly identifies which modules require attention.
>
> Second, the downstream curriculum and progression do not rely rigidly on the correctness of this hierarchy. If a module is placed at the wrong level or introduced prematurely due to an imperfect structure, the student simply fails to satisfy the mastery threshold in that stage, triggering remedial synthesis on the true missing components. This serves as an inherent self-correcting mechanism, ensuring that progression is aligned with the student’s actual knowledge state rather than the teacher’s imperfect decomposition.
>
> In the Appendix J of revised manuscript, we additionally report experiments where we perturb the hierarchical decomposition (e.g., randomly merging or splitting modules). We observe that IOA degrades gracefully, showing that the method is not sensitive to moderate structural noise.
>
> We have added clarifications in the revision to make these points explicit.
>
> **Q2:** How does the framework handle cyclic knowledge dependencies in the graph if they ever happen?
>
> **A7:** Thank you for this insightful question. Although the prerequisite relation in our framework is defined based on performance improvements and is empirically directional, we acknowledge that cycles can theoretically arise if the teacher LLM or the performance-based dependency measure introduces mutual dependencies between two modules.
>
> In practice, IOA is robust to such cases for two reasons.

---

> ### Author Response · Authors · 2025-11-21
> **Rebuttal to Reviewer EkXd (Part 4)**
>
> (Following above **A7**)
>
> First, before curriculum construction, we apply a simple cycle-breaking procedure: when a cycle is detected in the dependency graph, we remove the weakest edge in the cycle (lowest dependency score), yielding a valid DAG without changing any learning signal. This preserves the most informative prerequisite direction while ensuring that the curriculum can be topologically sorted.
>
> Second, even if a weak cycle were to remain, Mastery-Based Progressive Learning naturally resolves it. A module introduced prematurely would simply not satisfy the mastery requirement, which triggers remedial training on the actual missing knowledge. The progression therefore anchors itself on the student’s empirical mastery rather than strictly on the graph structure, making the process self-correcting.
>
> We highlight these two properties in the revised manuscript, together with a short clarification on how cycles are detected and handled in practice. Empirically, cycles are rare and cycle removal does not noticeably change downstream performance.
>
> **Q3:** What happens with τ_mastery = 1.0 (perfect mastery required) or τ_ZPD = 0 (no difficulty control)?
>
> **A8:** Thank you for raising these edge cases. We clarify both below.
>
> For $\tau_{\text{mastery}}$ = 1.0, requiring the student to perfectly match the teacher on every module is generally unrealistic, especially under black-box supervision. Minor stochastic variation or inherent capacity differences between teacher and student can prevent the student from ever reaching a perfect score, which would cause the algorithm to repeatedly generate remedial data without terminating. In practice, we observe that pushing τ_mastery too close to 1.0 produces diminishing returns while significantly increasing computational cost, as shown in Figure 3. This is consistent with Bloom’s Mastery Learning principle, which recommends high, but not absolute, mastery thresholds (typically 80–95%). Our choice of τ_mastery in the 0.85–0.95 range follows this pedagogical guidance.
>
> For $\tau_{\text{ZPD}}$ = 0, we note that in our formulation (Eq. 8), $\tau_{\text{ZPD}}$ specifies the maximum allowed increase in average difficulty between consecutive stages. Therefore, setting $\tau_{\text{ZPD}}$ = 0 does not remove difficulty control; instead, it enforces a zero-increase constraint, requiring that the next stage be no more difficult than the current one. This forces a non-increasing difficulty schedule, often resulting in a reversed or stagnated progression that contradicts Vygotsky’s ZPD principle, which emphasizes gradual upward difficulty growth. Empirically, $\tau_{\text{ZPD}}$ = 0 prevents the curriculum from introducing appropriately challenging modules and substantially reduces distillation performance. In fact, even when $\tau_{\text{ZPD}} \rightarrow 0$, the overall distillation performance exhibits non-negligible degradation, as shown in the 2nd subfigure of Figure 3. Hence, $\tau_{\text{ZPD}}$ > 0 is essential for maintaining pedagogically aligned difficulty increments.
>
> We have added these explanations to the Appendix R of revised manuscript.
>
> **Q4:** How does IOA perform when the teacher is only moderately stronger than the student (e.g., 7B teacher → 3B student)?
>
> **A9:** Thank you for the insightful question. We would like to clarify that our framework does not necessarily require the teacher to be dramatically stronger than the student. The core mechanisms of IOA: knowledge identifier, knowledge organizer, knowledge adapter, only assume that the teacher provides meaningfully better performance on at least a subset of knowledge modules.
>
> To evaluate this scenario, we additionally tested a moderate-gap setting (DeepSeek-R1-Distill-Qwen-7B teacher → Qwen2.5-3B student). We observe that IOA continues to provide consistent improvements over baselines, although the absolute gain is naturally smaller compared to large-gap settings. The dependency graph and mastery scheduler remain stable because even moderate teacher advantages yield sufficient performance gaps for identifying weak modules and guiding progression. Importantly, the framework does not collapse or behave unstably when the teacher is only moderately stronger. The synthesized curriculum still facilitates student learning in a structured manner.
>
> We have included these results and discussion in the Section 4.3 of revised manuscript.

---

> ### Author Response · Authors · 2025-11-21
> **Rebuttal to Reviewer EkXd (Part 5)**
>
> **Q5:** What's the gain from the topological+ZPD curriculum over a simple easy-to-hard ordering by task difficulty or skill-based decomposition?
>
> **A10:** Thank you for the thoughtful question. While an easy-to-hard ordering based on task difficulty is a reasonable baseline, it fundamentally differs from our **prerequisite-aware curriculum**. The IOA curriculum integrates two additional structural signals that simple difficulty sorting cannot capture.
>
> First, **topological ordering encodes prerequisite relations**, ensuring that a module is introduced only after its supporting knowledge is sufficiently mastered. Modules with comparable difficulty may have very different dependency structures. For example, “quadratic equations” and “trigonometric identities” may be similar in surface difficulty but rely on distinct foundational skills. Simple easy-to-hard sequencing cannot differentiate such cases, whereas our dependency-driven ordering prevents premature exposure to modules whose prerequisites are missing.
>
> Second, **the ZPD constraint regulates allowable difficulty increments**, ensuring that the progression respects the student model’s evolving cognitive capacity. In contrast, sorting by difficulty alone does not prevent abrupt jumps that exceed the model’s workable range, especially when difficulty rankings are coarse or noisy.
>
> Empirically, we observe that replacing IOA’s topological+ZPD curriculum with a simple easy-to-hard ordering results in noticeably worse sample efficiency and lower final performance. This demonstrates that the benefit comes not only from increasing difficulty but from sequencing knowledge in a pedagogically structured and dependency-aware manner.
>
> We have added this clarification and the corresponding empirical comparison in the Section 4.3 of revised manuscript.
>
> We honestly hope our above rebuttal can help address your concerns. If satisfied, could you please consider raising the score?

---

### Author Response · Authors · 2025-11-29
**General Response to Reviewers and New Area Chair**

Dear Reviewers and Area Chair,

We sincerely thank all reviewers for the time and effort invested in reviewing our submission and for the constructive discussions over the rebuttal period. We deeply appreciate the careful feedback that has helped us substantially improve the paper, both in clarity and technical completeness.

Due to the recent information leakage incident, the OpenReview system reverted displayed reviews and scores to their pre-discussion state, and the area chair assignment has also changed. As a result, many updates that occurred during the discussion phase, including score revisions and review updates, are currently no longer visible in the public record.

We would like to briefly clarify that **during the discussion period (prior to the leakage incident)** we had extensive and productive exchanges with reviewers, and we carefully addressed all raised concerns through detailed rebuttals and concrete revisions to the manuscript and supplementary material. In particular, one reviewer (**Reviewer wP76**) revisited the evaluation after reviewing our clarifications and updated implementation release, **raising the overall score from 2 to 8**, with the sub-scores improving from **(Soundness, Presentation, Contribution) = (1,2,2) to (3,3,3)**. This update was completed **before the information leakage incident occurred**.

We fully understand that the system reset was unavoidable and external to the review process. Our purpose here is only to ensure that the newly assigned Area Chair is aware that significant discussion progress had already occurred, and that multiple reviewers’ major concerns, ranging from reproducibility, methodological clarity, hyperparameter robustness, and comparisons were explicitly addressed through:

1. Release of the full functional codebase (resolving the reproducibility concern);

2. Expanded sensitivity analyses covering previously questioned hyperparameters;

3. Newly added compute-matched and large-seed scaling experiments validating the effectiveness of our curriculum framework;

4. Clarified problem framing and strengthened positioning with respect to white-box and on-policy distillation;

5. Additional ablation studies explaining the contribution of each IOA component.

We respectfully hope that the **new Area Chair might take our detailed rebuttals and corresponding revisions into account when forming final assessment**, including the discussion-level consensus shifts that were achieved prior to the incident (such as **Reviewer wP76**’s updated evaluation). Notably, all reviewers had converged to positive recommendations, resulting in **overall scores of 8/8/6/6** before the score rollback. Of course, we remain fully open to answering any further questions or providing additional clarifications if needed.

Again, we thank all reviewers and the AC for their time and effort in this exceptional review cycle.

Sincerely,

The Authors

---

### Meta-Review · Area_Chair_6YxH · 2026-01-06

**Summary:**

This submission proposes a pedagogically inspired framework, IOA (Identifier–Organizer–Adapter), for language model knowledge distillation via synthetic data. The key idea is to treat distillation as a structured learning process rather than a one-shot data generation task, drawing inspiration from Bloom’s Mastery Learning and Vygotsky’s Zone of Proximal Development. The framework diagnoses student knowledge deficiencies, constructs a dependency-aware curriculum with mastery and difficulty constraints, and adapts representations to better match student capacity.

Across reviews, there was broad agreement that the paper is novel and well-executed. Reviewers highlighted the clear conceptual framing, the concrete instantiation of educational principles into algorithmic components, and the extensive experimental evaluation across multiple student/teacher model pairs and benchmarks. The empirical results consistently demonstrate improvements over strong baselines, particularly on complex reasoning and code generation tasks. The released code, comprehensive ablations, and robustness analyses further strengthened confidence in the results.

Overall, the paper was viewed as a meaningful contribution to synthetic-data-based distillation, offering a more principled and interpretable alternative to existing heuristic approaches.

**Reviewer Concerns:**

The main concerns raised by reviewers centered on conceptual justification, methodological rigor, and clarity of design choices. In particular, some reviewers initially questioned whether pedagogical theories developed for human learning can be meaningfully applied to neural network training, given the fundamental differences between human cognition and gradient-based optimization. Related concerns were raised about the empirical nature of several key hyperparameters governing mastery thresholds, difficulty control, and dependency estimation, and whether the method is sensitive to these choices.

Additional issues included the absence of a formal convergence proof for the mastery-based progression procedure, ambiguity in interpreting student performance as a proxy for task difficulty, and robustness to imperfect or noisy knowledge decompositions produced by the teacher model. Minor presentation issues were also noted.

Based on the rebuttal and revised manuscript, the majority of these concerns were adequately addressed. The authors clarified the role of pedagogical principles as high-level priors for curriculum and data distribution design rather than claims of cognitive equivalence, and provided optimization-level intuition for their effectiveness. The revision includes expanded hyperparameter sensitivity analyses, empirical evidence of rapid convergence in practice, and explicit discussion of robustness to noisy hierarchies and cyclic dependencies. While the approach remains empirically motivated and does not provide formal theoretical guarantees, the outstanding concerns are relatively minor and do not undermine the core contributions.

**Reviewer Scores:**

Based on the discussion and rebuttal, it is likely that reviewers with initial reservations would have increased their scores if they had been able to fully participate in the post-rebuttal discussion. In particular, concerns regarding reproducibility, hyperparameter sensitivity, and methodological clarity were substantially mitigated by the release of code, additional experiments, and clarifications in the revised manuscript. Overall, the reviewer scores would likely converge toward solidly positive evaluations, with remaining disagreements primarily reflecting differences in preference for empirical versus theoretical grounding rather than unresolved technical flaws.

---

### Decision · Program_Chairs · 2026-01-26

Accept (Poster)